# Global Convergence of Sampling-Based Nonconvex Optimization through Diffusion-Style Smoothing

## Abstract

Sampling-based optimization (SBO), like cross-entropy method and evolutionary algorithms, has achieved many successes in solving non-convex problems without gradients, yet its convergence is poorly understood. In this paper, we establish a non-asymptotic convergence analysis for SBO through the lens of *smoothing*. Specifically, we recast SBO as gradient descent on a smoothed objective, mirroring noise-conditioned score ascent in diffusion models. Our first contribution is a landscape analysis of the smoothed objective, demonstrating how smoothing helps escape local minima and uncovering a fundamental *coverage–displacement tradeoff*: smoothing renders the landscape more benign by enlarging the locally convex region around the global minimizer, but at the cost of introducing optimizer displacement. Building on this insight, we establish non-asymptotic convergence guarantees for SBO algorithms to a neighborhood of the global minimizer. Furthermore, we propose an annealed SBO algorithm, Diffusion-Inspired Dual-Annealing (DIDA), which is provably convergent to the global optimum. We conduct extensive numerical experiments to verify our landscape results and also demonstrate the compelling performance of DIDA compared to other gradient-free optimization methods. Lastly, we discuss implications of our results for diffusion models.

## 1 Introduction

Many real-world optimization problems are highly nonconvex or even discontinuous, e.g., in optimal control problems in contact-rich robotics (Graesdal et al., 2024; Li et al., 2024), computer vision (Brox & Malik, 2011; Hruby et al., 2021) and machine learning (Bengio, 2009; Jain & Kar, 2017; Chaudhari et al., 2017; Gargiani et al., 2019). While significant research in optimization has yielded methods like interior-point algorithms (Freund & Mizuno, 2000), sequential convex programming (Dinh & Diehl, 2010), and sum-of-squares techniques (Powell, 1965) to tackle nonconvexity, they may be computationally intensive, only guarantee convergence to local minima, or struggle with non-smoothness or discontinuities.

Recently, Sampling-based Optimization (SBO) Ernst et al. (2007); Ma et al. (2019); Hansen (2023); Williams et al. (2016) has gained considerable popularity as a promising alternative. Their appeal stems from the ability to handle highly nonconvex or even discontinuous objective functions effectively without requiring explicit gradients. Furthermore, they evaluate many candidate solutions in parallel, allowing for massive parallelization on Graphics Processing Units (GPUs). As a result, SBO techniques, such as Model Predictive Path Integral Control (MPPI) (Williams et al., 2017) and cross-entropy method (CEM) (Ernst et al., 2007), have found successful

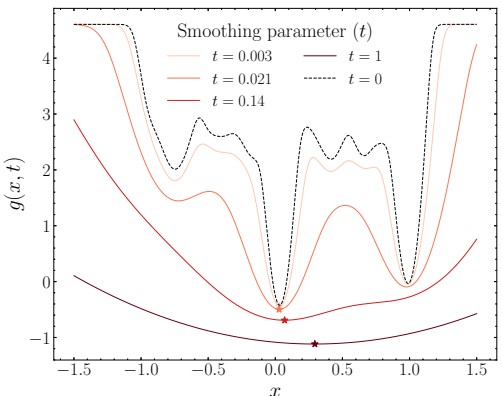

Figure 1: Illustration of smoothing over different $t$ on a bounded-domain GMM-shaped landscape $g(x, t) = -\log \rho_t(x)$. As $t$ increases, the landscape is smoothed and the minimizer $x_t^*$ of the smoothed objective shifts away from the reference minimizer $x^*$.

applications in path planning (Li et al., 2024; Pan et al., 2024), image processing (Joseph & Bhatnagar, 2018) and black-box optimization (De Boer et al., 2005).

Despite this growing empirical success and adoption, several critical gaps remain in the understanding and application of SBO. The theoretical side, particularly concerning convergence properties (e.g., guarantees of convergence to global or local optima, convergence rates), are still underdeveloped compared to more traditional optimization methods. Additionally, there is a lack of a systematic framework for designing SBO algorithms, including principled methods for hyperparameter tuning, adaptive sampling strategies, and variance reduction. In light of the gap, we answer the following:

*Why can SBO effectively find global optima in highly non-convex landscapes? How can hyperparameters be designed to achieve optimal convergence?*

To answer this question, we leverage a recent observation (Mobahi & Iii, 2015; Pan et al., 2024): many SBO methods find a high-quality optimizer of the objective function not by optimizing directly on $f(x)$, but rather by optimizing a smoothed version $g(x;t)$ (defined in Equation (4)). The parameter $t$ controls this smoothing and corresponds to the sampling variance in SBO. When $t = 0$, $g(x;t) = f(x)$ and when $t$ increases, $g(x;t)$ becomes smoothed as demonstrated in Figure 1. Therefore, the convergence of SBO boils down to the properties of the function $g(x;t)$.

**Contribution.** Our first contribution is that we show the landscape of $g(x;t)$ demonstrates a *"coverage–displacement"* tradeoff: suppose the global optimizer $x^*$ lies in a locally convex region of $f(x)$, then as $t$ increases, the convex region expands, covering a larger region that allows for easier convergence; meanwhile, a larger $t$ also increases the optimizer displacement as the optimizer of $g(x;t)$ shifts away from the true optimizer $x^*$. Based on this landscape property of $g(x;t)$, we show conditional non-asymptotic convergence of a class of SBO algorithms to a neighborhood of the global minimizer. Lastly, the tradeoff also suggests an annealing strategy for the parameter $t$, based on which we propose a new algorithm: DIDA that schedules the smoothing parameter and converges to the exact global minimizer under the stated conditions. In experiments, we verify the coverage–displacement tradeoff, and also show that the proposed DIDA algorithm achieves state-of-the-art over benchmarks. Lastly, we show that the SBO has deep connections to the ODE method in diffusion model. Specifically, our results have implications for guidance's role in improving sample quality.

## 2 Related Work

**Zeroth-order gradient estimation.** A large class of gradient-free optimization algorithms uses function evaluations to estimate the gradient via one-point/multi-point estimators and conducts stochastic gradient descent (SGD) Liu et al. (2018); Balasubramanian & Ghadimi (2019); Wang et al. (2018). It has been studied in bandit online optimization settings (Flaxman et al., 2004; Bach & Perchet, 2016) for convex functions, showing its dimension-dependent global convergence rate (Nesterov & Spokoiny, 2017; Shamir, 2015; Duchi et al., 2014). Nesterov & Spokoiny (2017) establish complexity bounds for random Gaussian search directions, and Duchi et al. (2014) sharpen minimax rates for two-point zeroth-order estimators. These results focus on oracle complexity and direct optimization of the original objective, mostly in convex settings or stationarity-style nonconvex guarantees. For non-convex functions, related algorithms have also been proposed in (Belloni et al., 2015) with SGD-style updates. However, these works mainly show local convergence in non-convex settings. Related noise-injection analyses view stochastic or Gaussian perturbations as a smoothing mechanism. Harshvardhan & Stich (2021) analyze first-order perturbed SGD for objectives $f = g + h$, assuming the smoothed stochastic-gradient noise and the smoothed-gradient bias are controlled relative to a regular component $g$ that satisfies a PL or strong-convexity condition. Mishchenko & Stich (2023) study Gaussian noise injection for first-order SGD under a decomposition $f = \hat{f} + \omega$, where $\hat{f}$ is smooth/PL and the high-frequency perturbation $\omega$ is bounded with bounded Clarke subgradients. These works explain how injected noise can reduce high-frequency nonconvexity, but they are first-order analyses that assume a hidden regular component. In contrast, our work studies sampling-based zeroth-order updates through the landscape of the Boltzmann-smoothed objective $g(x;t)$, including the coverage–optimizer-displacement tradeoff that enables our conditional convergence analysis.

**(Cross-entropy-style) sampling-based optimization.** Our work is closely related to a class of methods that cast optimization as a sampling problem from a target distribution (cf. (3)). For example, De Boer et al. (2005); Rubinstein & Kroese (2004) propose a gradient-free method that minimizes the KL divergence between the target distribution and the sampling distribution. It has been applied to planning (Pinneri et al., 2021; Chua et al., 2018), nonlinear programming (Kothari & Kroese, 2009), and power systems (Ernst et al., 2007) and has been proved to asymptotically converge to the target distribution (Margolin, 2005; Joseph & Bhatnagar, 2018). Monte Carlo Markov Chain (MCMC) methods (Metropolis et al., 1953; Hastings, 1970) like Metropolis-Hastings (Green & Han, 1992), Simulated Annealing (Bertsimas & Tsitsiklis, 1993), and CMA-ES (Akimoto et al., 2012) are proposed to draw samples from the target distribution by constructing a Markov chain that converges to the target distribution. While sharing the same proposal distribution and update rules, our work targets the original optimization problem rather than KL minimization, and contributes a landscape analysis for the smoothed distribution (cf. (4)) that enables conditional non-asymptotic convergence of SBO, which to the best of our knowledge is not present in this literature.

**Homotopy optimization.** Our results on the varying smoothing parameter (Section 4.4) are closely related to homotopy optimization Blake & Zisserman (1987); Yuille (1989); Allower & Georg (1990); Allgower & Georg (1990); Dunlavy & O'Leary (2005). With wide applications Yuille (1989); Terzopoulos (1988); Gold et al. (1994); Brox & Malik (2011); Hruby et al. (2021); Bengio (2009); Jain & Kar (2017); Chaudhari et al. (2017); Gargiani et al. (2019); Pan et al. (2019); Bergman & Axehill (2017); Pan et al. (2024); Xue et al. (2024), homo-

| Algorithm | Parameters Algorithm 1 | Solution Algorithm 1 |
|---|---|---|
| MPPI (Williams et al., 2018) | $t$, $\lambda$ | Softmax |
| SA (Coleman et al., 1993) | $t$, $\lambda$ | Softmax |
| CEM (Rubinstein & Kroese, 2004) | $t$, $\lambda$ | Top-k |
| CMA-ES (Akimoto et al., 2012) | $t^{\mathrm{cov}}$, $\lambda$ | Softmax |
| MBD (Pan et al., 2024) | $t$, $\lambda$ | Softmax |
| DIDA (Ours) | $t$, $\lambda$ | Softmax |

Table 1: Comparison of different SBO algorithms with Gaussian proposal distributions parameterized by smoothing parameter $t$ and temperature parameter $\lambda$.[2]

topy optimization works by solving a sequence of smoothed surrogate problems that gradually transform from an easy-to-solve objective to the original complex objective (Lin et al., 2023), and the aim is to find (near) global optima for the original objective. Among various homotopy methods, the Gaussian smoothing (or continuation) method (Loog et al., 2001; Mobahi & Fisher, 2015) has gained significant attention due to its simplicity and effectiveness. Extensive theoretical analyses have been conducted for Gaussian smoothing: Mobahi & Fisher (2015) relate Gaussian smoothing to convex-envelope approximations, and Mobahi & Iii (2015) bound the endpoint of Gaussian continuation through an optimization-complexity measure. Regarding convergence, Hazan et al. (2015) prove a double-loop graduated-optimization guarantee under the global $\sigma$-nice condition, including stochastic and zeroth-order variants, while single-loop convergence is established in (Iwakiri et al., 2022; Lin et al., 2023). These continuation analyses assume global structural properties of the smoothed path or bound the endpoint of a continuation trajectory. Geometric schedules have also been studied for stochastic optimization; for example, Ge et al. (2019) show that step-decay learning rates improve the final iterate of SGD in streaming least-squares regression under noise-covariance and fourth-moment assumptions. Our geometric schedule instead anneals the smoothing level $t_m$, coupled with the temperature $\lambda_m$, so that the iterates remain in the moving smoothed convex region while the optimizer displacement decreases. Overall, existing homotopy theory relies on strong assumptions on the smoothed landscape; our work instead characterizes locally benign properties of the Boltzmann-smoothed landscape and connects them to finite-sample sampling-based updates.

## 3 Problem Formulation and Preliminaries

We focus on a general unconstrained optimization problem:

$$\min_{x \in \mathbb{R}^d} f(x), \tag{1}$$

---

[2]Blue indicates fixed parameters, Red indicates annealed parameters. $t^{\mathrm{cov}}$ for CMA-ES denotes $t$ adapted via covariance matrix.

where the objective function $f : \mathbb{R}^d \to \mathbb{R}$ may be non-convex, and we denote a global optimizer as $x^*$ with minimum $f^* = f(x^*)$. We consider a broad class of *sampling-based optimization methods* (SBO). SBO (Algorithm 1) is composed of three steps at each iteration $m$: (a) Update parameters (Algorithm 1): update the sampling parameters $\theta_m$ based on the previous iterate $x_m$ and the sampling parameters. (b) Sample (Algorithm 1): at each step $m$, propose $N$ candidate points $\{y_i\}_{i=1}^N$ from a sampling distribution $P(y|x_m, \theta_m)$ and evaluate the corresponding function values $\{f_i = f(y_i)\}_{i=1}^N$. One common choice of $P(y|x, \theta)$, and the focus of this paper, is the Gaussian proposal distribution $P_{t|0}(\cdot \mid x) = \mathcal{N}(x, tI_d)$. We write its density as $p_{t|0}(y \mid x) = k(y - x; t)$, where

$$k(z; t) = (2\pi t)^{-d/2} \exp\left(-\frac{\|z\|^2}{2t}\right)$$

is the density of $\mathcal{N}(0, tI_d)$. (c) Update solution (Algorithm 1): update the current solution estimate $x$ following certain update rules. Common practices include the softmax update rule:

$$x_{m+1} = \sum_{i=1}^N w_i y_i \quad \text{where} \quad w_i = \frac{e^{-\frac{1}{\lambda} f(y_i)}}{\sum_{j=1}^N e^{-\frac{1}{\lambda} f(y_j)}} \tag{2}$$

and the top-$k$ update rule: $x_{m+1} = \frac{1}{k} \sum_{j=1}^k y_{i_j}$, where $\{i_j\}_{j=1}^k$ index the $k$ lowest-cost samples. Crucially, the updates rely solely on the function $f(y_i)$ evaluation without requiring direct computation of gradients $\nabla f$, making it a *zeroth-order optimization* method. Based on how the sampling distribution $P(y|x, \theta)$ is updated and the update rules they use, Table 1 summarizes the most popular SBO algorithms.

---

**Algorithm 1:** Generic Sampling-Based Optimization

---

**Input:** initial guess $x_0$, sampling parameters $\theta_0$, sample size $N$, iteration number $K$, objective $f$, proposal distribution $P(\cdot \mid x, \theta)$, parameter update rule $U_P$, solution update rule $U_S$

1 **for** $m = 0$ *to* $K - 1$ **do**
2      Draw samples $\{y_i\}_{i=1}^N \overset{\text{i.i.d.}}{\sim} P(\cdot \mid x_m, \theta_m)$ ;          // Generate candidates
3      Evaluate function values $\{f_i = f(y_i)\}_{i=1}^N$;
4      Update solution $x_{m+1} = U_S(x_m, \{y_i\}_{i=1}^N, \{f_i\}_{i=1}^N, \theta_m)$ ;          // Solution update
5      Update parameters $\theta_{m+1} = U_P(\theta_m, m, x_m)$ ;          // Parameter update
6 **return** $x_K$

---

This work proposes that the success of sampling based optimization lies in an implicit connection to gradient-based optimization, but on a modified landscape. Specifically, we show that sampling-based methods effectively perform gradient descent on a *smoothed version* of the original objective function. This implicit smoothing mitigates the challenges posed by non-convexity, facilitating convergence towards better solutions, potentially global minima. In the subsequent subsection, we formally define the smoothed objective function and show its connection to sampling-based optimization.

**Roadmap for the rest of the paper.** Section 3.1 defines the smoothed objective $g(x; t)$ and connects it to SBO. Then, in Section 4, we use this smoothing framework to analyze the convergence of sampling-based optimization methods. Specifically, Section 4.2 studies the landscape of $g(x; t)$, and Section 4.3 analyzes the convergence of SBO. Our analysis also leads to algorithmic insights on the design of SBO algorithms, discussed in Section 4.4. Finally, Section 4.5 discusses implications of our results for diffusion models.

### 3.1 Preliminaries: Sampling-based optimization is ZO-GD on smoothed objective

**Smoothed objective.** We consider the Gibbs-Boltzmann density

$$p_0(x) \propto e^{-f(x)/\lambda}, \tag{3}$$

where $\lambda > 0$ is a temperature parameter. Minimizing $f(x)$ is equivalent to finding the mode of the distribution with density $p_0(x)$. This distribution is widely used as in Langevin dynamics based algorithms for optimization (Ma et al., 2019; Wibisono, 2018; Xu et al., 2018). As discussed earlier, the effectiveness of

these methods is deeply connected to an implicit **landscape smoothing** effect, i.e, their update mechanisms effectively approximate gradient steps on a smoothed version of the original landscape (Pan et al., 2024). To define this smoothed landscape, we use the Gaussian density $k(\cdot; t)$ defined above and set

$$p(x; t) = (p_0(\cdot) * k(\cdot; t))(x) = \int_{\mathbb{R}^d} p_0(y) k(x - y; t) dy, \quad g(x; t) = -\lambda \log p(x; t). \tag{4}$$

The parameter $t$ controls the degree of smoothing: as $t \to 0$, $g(x; t)$ approaches $f(x)$ (up to constants), while for larger $t$, $g(x; t)$ becomes smoother, potentially convexifying the landscape even if $f(x)$ is highly non-convex. Furthermore, as $t$ increases, the smoothed density $p(\cdot; t) = p_0 * k(\cdot; t)$ becomes progressively more regular, and the corresponding smoothed objective $g(\cdot; t) = -\lambda \log p(\cdot; t)$ inherits a smoother landscape than $f$. In Theorem 4.3 we show that, under Assumptions 4.1 and 4.2, $g(\cdot; t)$ is strongly convex in a neighborhood $\mathcal{R}_{SC}(t)$ with curvature on the order of $\Omega\big(\lambda/(t + \lambda/\alpha)\big)$.

**Connection between SBO and $g(x; t)$.** SBO's update step (Algorithm 1) with softmax update rules (Equation (2)) can be interpreted as performing *zeroth-order gradient descent* on this smoothed objective $g(x; t)$.

**Proposition 3.1** (Adapted from (Mobahi & Iii, 2015; Pan et al., 2024)). *Let $P_{t|0}(\cdot \mid x) = \mathcal{N}(x, tI_d)$, with density $p_{t|0}(y \mid x) = k(y - x; t)$, and recall $p_0(y) \propto \exp(-f(y)/\lambda)$ and $g(x; t) = -\lambda \log\big((p_0 * k(\cdot; t))(x)\big)$. Then the softmax update Equation (2) can be written as one step of estimated gradient descent on $g(\cdot; t)$:*

$$x^+ = x - \frac{t}{\lambda} \nabla_x g^{(0)}(x; t), \tag{5}$$

$$\nabla_x g^{(0)}(x; t) := \frac{\lambda}{t} \left( x - \frac{\sum_{i=1}^N w_i y_i}{\sum_{i=1}^N w_i} \right), \qquad \{y_i\}_{i=1}^N \overset{i.i.d.}{\sim} P_{t|0}(\cdot \mid x), \tag{6}$$

*where $w_i := \exp(-f(y_i)/\lambda)$ (equivalently, $w_i \propto p_0(y_i)$; the normalizing constant of $p_0$ cancels). Moreover, $\nabla_x g^{(0)}(x; t)$ is a zeroth-order (Monte Carlo) estimator of $\nabla_x g(x; t)$ in the sense that, as $N \to \infty$,*

$$\nabla_x g^{(0)}(x; t) \to \nabla_x g(x; t) \quad in \ probability.$$

Proposition 3.1 shows that SBO updates perform zeroth-order gradient descent on the smoothed objective $g(x; t)$, rather than directly on $f$. Intuitively, smoothing via convolution yields a more benign landscape for optimization. In the next section, we formalize this intuition by analyzing the geometry of $g(x; t)$ (e.g., local strong convexity) and deriving convergence guarantees for SBO.

**Connection between Smoothing and Diffusion.** Gaussian kernel smoothing is also standard in the diffusion (score-based) literature, where one samples from a potentially complex target distribution $p_0(x)$ by progressively corrupting it with Gaussian noise and then approximately reversing this noising process. For concreteness, we adopt the additive Brownian-noise forward process (Karras et al., 2024)[3]:

$$dx_t = dw_t, \quad t \in [0, T], \quad x_0 \sim p_0(x)$$

where $w_t$ is a standard Brownian motion. In short, the target density $p_0(x)$ is corrupted with Gaussian noise with density $k(x; t)$ to generate a smoothed density $p(x; t) = p_0(x) * k(x; t)$. The sampling procedure is done via the backward process, where the sample is guided back following the following ODE that goes in reverse time Song et al. (2020):

$$dx_t = -\frac{1}{2} \nabla \log p(x_t; t) dt, \quad t \in [T, 0]$$

In the above equation, note that $-\log p(x; t)$ is exactly the smoothed objective $g(x; t)$ we defined in Equation (4) up to the multiplicative factor $\frac{1}{\lambda}$. Therefore, the above ODE is equivalent to the gradient flow on $g(x; t)$ with time-varying $t$, which is similar to SBO except in diffusion model, the quantity $\nabla \log p(x; t)$ is typically estimated by a neural network. In light of this connection between SBO and diffusion model, we will discuss the implications of our result for diffusion model in Section 4.5.

---

[3]Many works instead use a *variance-preserving* parameterization (e.g., an Ornstein–Uhlenbeck/VP-SDE). These formulations are closely related to additive Gaussian smoothing through a rescaling of the state and a reparameterization of the noise level; see Lu & Song (2024) for a unified view.

## 4 Main results

As established in Section 3.1, sampling-based optimization methods can be interpreted as zeroth-order gradient descent on a smoothed objective function $g(x; t)$. Building on this foundation, this section presents our core theoretical contributions, where we rigorously analyze the properties of this smoothed landscape and the convergence behavior of SBO that navigates it. Our analysis begins with the key assumptions (Section 4.1). In Section 4.2, we detail our main findings on the landscape of $g(x; t)$, particularly the *coverage–displacement tradeoff* inherent in the smoothing parameter $t$. This tradeoff governs the balance between achieving a more favorable landscape structure and the optimizer displacement introduced by the smoothing process. Subsequently, we explore the implications of this landscape for sampling-based optimization, showing a conditional convergence analysis for SBO to an approximate optimizer (Section 4.3). Finally, the coverage–displacement tradeoff also suggests a way to jointly anneal the smoothing parameter $t$ and the temperature parameter $\lambda$ to improve convergence, which we show in (Section 4.4) and we demonstrate their effective convergence to the true optimum. To the best of our knowledge, the necessity of co-annealing the temperature parameter $\lambda$ in conjunction with $t$ is not known in the literature and hence we name our new algorithm as Diffusion-Inspired Dual Annealing(DIDA).

### 4.1 Assumptions

To analyze the behavior of our optimization approach, we impose the following assumptions on the problem structure. Our first assumption is on the tail bound of the distribution $p_0$.

**Assumption 4.1** (Global Sub-Gaussian Assumption). *There exists a constant $D_\tau > 0$ such that the tail probability deviating from the optimal solution $x^*$ is bounded: $P_0(\|x - x^*\| \geq a) \leq \exp\left(-\frac{a^2}{2\tau^2}\right), \quad \forall a \geq D_\tau$, where $P_0$ is the probability measure induced by the density $p_0$. We also denote the total probability of being outside the $D_\tau$-ball around $x^*$ as $P_0(\|x - x^*\| \geq D_\tau) = P_{out}$.*

The sub-Gaussian assumption (Assumption 4.1) describes the target distribution $p_0(x)$'s concentration around the global optimum $x^*$. Weaker than log-concavity (Ma et al., 2019), it constrains only tail behavior, accommodating a broader class of **non-convex** functions. A concrete control example is given in Section B.2: a positive-definite quadratic LQR core is perturbed by finitely many bounded Gaussian/RBF state penalties, a standard smooth-obstacle modeling device in potential-field and MPC-style motion planning (Khatib, 1986; Cho et al., 2018; Pae et al., 2021; Wang et al., 2022). The bounded perturbation preserves Gaussian-type Boltzmann tails, while the localized penalties create compact negative-curvature regions, so the induced density is not log-concave. This illustrates the intended scope of Assumptions 4.1 and 4.2: confining tails and local regularity near the selected minimizer, without global convexity. This condition is also met in many practical problems, like regularized machine learning losses (Jain & Kar, 2017), or LQR-style control (Pan et al., 2019). Adopting it allows our analysis to cover more non-convex functions, ensuring sufficient probability mass near the optimum. $\tau$ sets the concentration width, and $P_{\text{out}}$ quantifies tail probability.

Our second assumption is on the behavior of $f$ around $x^*$. For notational convenience, we define the following regions: $B_\tau := \{x \mid \|x - x^*\| < D_\tau\}$, a ball of radius $D_\tau$ around $x^*$.

**Assumption 4.2** (Local Convexity and Smoothness). *We assume that the objective function $f(x)$ is strongly convex and smooth within the ball $B_\tau$ centered at the optimum $x^*$. Specifically, there exists constants $\alpha > 0$ and $\beta > 0$ such that for all $x, y \in B_\tau$, the function satisfies $\alpha$-strong convexity: $f(y) \geq f(x) + \nabla f(x)^\top (y - x) + \frac{\alpha}{2}\|y - x\|^2$, and $\beta$-smoothness: $\|\nabla f(x) - \nabla f(y)\| \leq \beta\|x - y\|$. The ratio $\kappa_0 = \beta/\alpha$ defines the local condition number of $f(x)$ within $B_\tau$.*

Assumptions 4.1 and 4.2 provide the necessary structure for our analysis. Specifically, Assumption 4.2 ensures local regularity (strong convexity and smoothness) near the optimum $x^*$, while Assumption 4.1 controls the global tail decay of the target distribution $p_0$. This combined structure enables a rigorous analysis of the convergence behavior of sampling-based optimization methods within the proposed framework, which we introduce in the subsequent subsections.

### 4.2 Coverage and Optimizer-Displacement Tradeoff: The Role of Smoothing

Here we show that smoothing the original objective $f(x)$ into $g(x;t)$ via the parameter $t$ induces a fundamental tradeoff. This tradeoff balances the *coverage* of desirable landscape properties (such as convexity) with the optimizer displacement between the minimizer $x_t^*$ of $g(x;t)$ and the true global minimizer $x^*$ resulting from the smoothing process. This displacement is different from a function-value optimality gap, and we use this terminology consistently below. Our main results demonstrate that as $t$ increases, the convex region of $g(x;t)$ expands. Concurrently, however, the optimizer displacement widens, as the minimizer of $g(x;t)$ deviates further from the true optimum of $f(x)$. We detail these findings below.

**Coverage: Expansion of convex region.** We now state our first main result, showing that as $t$ increases, the convex region around $x^*$ expands at least on the order of $\sqrt{t}$.

**Theorem 4.3** (Strongly Convex Bound). *Under Assumption 4.1 and Assumption 4.2, let d denote the dimension of the space and define $\kappa_0 := \beta/\alpha$. Fix any $C_\alpha \in (0,1)$ and assume $\lambda \le \lambda_{\max}$ for some fixed $\lambda_{\max} > 0$. Then, for any $t > 0$, the function $g(x;t)$ is $\frac{C_\alpha \lambda}{t + \frac{\lambda}{\alpha}}$-strongly convex within the region*

$$\mathcal{R}_{SC}(t) := \left\{ x \in \mathbb{R}^d \;\middle|\; \|x - x^*\| \le C_E \min\left( \sqrt{\frac{t + \frac{\lambda}{\beta}}{\frac{\lambda}{\beta}}} \,,\, \sqrt{\frac{t + \tau^2}{\tau^2}} \right) D_\tau \right\} \tag{7}$$

*i.e., for all $x \in \mathcal{R}_{SC}(t)$ we have $\nabla^2 g(x;t) \succeq \frac{C_\alpha \lambda}{t + \frac{\lambda}{\alpha}} I$, provided the sufficient parameter conditions in Equation (20) hold. Here $D_\tau$, $\tau$, and $P_{out}$ are the sub-Gaussian/tail parameters from Assumption 4.1, and $C_E$ is the expansion-rate factor defining $\mathcal{R}_{SC}(t)$.*

As shown in Theorem 4.3, increasing the smoothing parameter $t$ generally improves the landscape's structure for optimization. Larger values of $t$ tend to expand the region $\mathcal{R}_{SC}(t)$ within which the smoothed function $g(x;t)$ exhibits strong convexity, a phenomenon illustrated in Figure 2. The sufficient condition Equation (20) makes explicit a concrete high-dimensional parameter regime in which this strong convexity guarantee holds; in particular it enforces the scalings $D_\tau^2 = \Theta(d)$, $P_{out} = O(1/d)$, and $C_E^2 = \Theta((\log d)/d)$, along with an explicit upper bound on $\tau$ in terms of $(\kappa_0, C_\alpha)$.

A larger convex region provides a wider basin of attraction, making it easier for optimization algorithms (including sampling-based ones that implicitly perform gradient descent on $g(x;t)$) to navigate towards $x_t^*$, which is defined as $x_t^* := \arg\min_x g(x;t)$ and avoid getting trapped in potentially complex local structures inherited from $f(x)$. In essence, large $t$ increases the "coverage" of the well-behaved, convex-like landscape. We caution that coverage is distinct from conditioning: the strong-convexity constant $\alpha_t = \frac{C_\alpha \lambda}{t + \lambda/\alpha}$ shrinks as $t$ grows, so the basin enlarges while local contraction slows. Next, we introduce the other side of the tradeoff, that is, a larger $t$ can introduce greater optimizer displacement.

**Optimizer displacement.** Despite the larger coverage, smoothing the objective function introduces displacement: the minimizer $x_t^*$ of $g(x;t)$ generally differs from the original function's global minimizer $x^*$. Understanding and bounding this displacement is crucial, as a large deviation would mean that optimizing $g(x;t)$ does not yield a sufficiently accurate solution to the original problem (1). The subsequent theorem bounds this displacement, relating it to the smoothing parameter $t$ and the properties of $f$.

**Theorem 4.4** (Optimizer displacement). *Let Assumption 4.1 and Assumption 4.2 hold. For any smoothing parameter $t > 0$, let $x_t^*$ denote the unique minimizer of the smoothed function $g(x;t)$ within the strongly convex region $\mathcal{R}_{SC}(t)$, as established in Theorem 4.3. If $x_t^*$ also lies within the region of local strong convexity and smoothness for the original function $f(x)$, i.e., $x_t^* \in B_\tau = \{x \mid \|x - x^*\| < D_\tau\}$, then the optimizer displacement $\|x_t^* - x^*\|$ is bounded by:*

$$\|x_t^* - x^*\| \le \min\left\{ \frac{(1 - C_\alpha)t}{C_\alpha} \frac{1}{4 D_\tau}, (D_\tau + \tau) \frac{t + \frac{\lambda}{\alpha}}{C_\alpha t} \right\} + \frac{\kappa_0 - 1}{2 C_\alpha} \sqrt{\frac{1}{2\pi(\frac{1}{t} + \frac{\alpha}{\lambda})}}. \tag{8}$$

*where $C_\alpha \in (0,1)$ is the strong convexity parameter from Theorem 4.3, and $D_\tau$, $\tau$ are the parameters from Assumption 4.1.*

Theorem 4.4 shows a larger $t$ increases the optimizer displacement $\|x_t^* - x^*\|$. The bound in (8) indicates that the displacement arises from two primary sources. The first, captured by both $\frac{(1-C_\alpha)t}{C_\alpha}\frac{1}{4D_\tau}$ and $(D_\tau + \tau)\frac{t+\frac{\lambda}{\alpha}}{C_\alpha t}$, is largely influenced by the smoothing parameter $t$ and the sub-Gaussian tail. The second term, $\frac{\kappa_0 - 1}{2C_\alpha}\sqrt{\frac{1}{2\pi\left(\frac{1}{t}+\frac{\alpha}{\lambda}\right)}}$, captures *local asymmetry* around $x^*$: when the local condition number $\kappa_0 = \beta/\alpha$ is close to 1, this contribution is small, and it grows with increasing anisotropy/curvature mismatch (larger $\kappa_0$). Notably, both contributions vanish as $t \to 0$, consistent with $g(\cdot; t) \to f$ (up to constants). Moreover, the bound remains finite as $t \to \infty$, matching the empirical behavior reported in Section 5.

**Discussion.** The landscape coverage (Theorem 4.3) versus optimizer-displacement (Theorem 4.4) tradeoff formally underpins successful SBO methods like MPPI (Williams et al., 2017) and diffusion-inspired approaches like Model-Based Diffusion (Pan et al., 2024), which often implicitly use smoothing (via sample averaging or explicit diffusion) to navigate complex, non-convex landscapes. We quantify this fundamental smoothing tradeoff here: larger $t$ creates a more benign, larger convex region but increases the optimizer displacement.

The coverage–displacement tradeoff developed here forms the foundation for analyzing the convergence of SBO algorithms, which we present in Section 4.3. Beyond the fixed $t$ regime, the inherent tradeoff in $t$ motivates the use of an annealing schedule for $t$. A large fixed smoothing level may be useful for entering the strongly convex region, but it is not the final optimization target because it can leave a large displacement $\|x_t^* - x^*\|$. Instead, we start with a relatively large $t_0$ to leverage the enlarged convex region and then decrease $t_m$ geometrically while annealing $\lambda_m$. The multi-stage analysis in Section 4.4 is designed to keep the iterate $x_m$ inside the moving region $\mathcal{R}_{SC}(t_m)$ as the region contracts, while the displacement of $x_{t_m}^*$ shrinks with $t_m$. This is the mechanism by which the algorithm approaches the true global minimum $x^*$.

**Multiple Convex Regions.** While our current analysis focuses on landscapes with a dominant global minimum characterized by local convexity (Assumptions 4.1 and 4.2), our analysis techniques could potentially be extended to multi-convex-region (multi-modal) scenarios. Larger $t$ might merge distinct basins of attraction, while annealing could help distinguish them later. Further investigation is needed to formalize the behavior in landscapes with multiple convex regions.

## 4.3 Convergence Analysis with Fixed Smoothing Parameter $t$

**From landscape analysis to convergence guarantees.** The landscape results of Section 4.2 supply two key ingredients for convergence analysis. First, the strong-convexity bound (Theorem 4.3) guarantees that $g(x; t)$ is $\alpha_t$-strongly convex with $\alpha_t = \frac{C_\alpha \lambda}{t + \lambda/\alpha}$ inside an expanding region $\mathcal{R}_{SC}(t)$, providing a basin in which gradient-based updates contract. Second, the optimizer-displacement bound (Theorem 4.4) quantifies the bias $\|x_t^* - x^*\|$ introduced by smoothing. Together, these determine the three-term error decomposition that underlies both the single-stage and multi-stage results below: *contraction* (governed by the $t$-dependent condition number $\kappa_t = \frac{\beta_t}{\alpha_t}$), *estimation noise* (captured by the gradient-estimator bias $K_t$ and variance $\sigma_t^2$ from Theorem C.1), and *optimizer displacement* (from Theorem 4.4).

For a fixed smoothing parameter $t$, Theorem 4.5 combines these ingredients to establish a non-asymptotic convergence rate to a neighborhood of $x^*$ whose size is controlled by the optimizer displacement and estimator error. The coverage–displacement tradeoff then motivates annealing $t$ (and $\lambda$) across stages: each stage $m$ inherits the landscape guarantees at its own noise level $t_m$ with stage-indexed parameters $\alpha_m, \beta_m, \kappa_m$, progressively tightening the optimizer displacement while remaining inside the contracting convex region, culminating in the conditional convergence guarantee of Theorem 4.6. See Table 3 for notation.

**Zeroth-order gradient estimator bounds.** Understanding the zeroth-order gradient estimator (Equation (6)) is crucial for SBO's non-asymptotic convergence. Theorem C.1 details its bias and variance bounds: bias bounded by $\frac{\lambda}{N} \cdot \mathcal{P}_B(t^{-1/2}, t^{1/2})$ and variance by $\frac{\lambda^2}{N} \cdot \mathcal{P}_V(t^{-1}, t)$. The polynomials $\mathcal{P}_B, \mathcal{P}_V$ (e.g., $\mathcal{P}_B = c_1 x + c_0 + c_2 y$) have coefficients that does not depend on $N$. This ensures estimator consistency, as bias/variance vanish for large $N$.

**Convergence of SBO under fixed $t$.** Combining Theorem 4.3 and the gradient estimator bounds Theorem C.1, we can derive the non-asymptotic convergence rate of the SBO in Equation (6).

**Theorem 4.5** (Convergence of Sampling without Annealing over $t$). *For any fixed smoothing level $t > 0$, the function $g(x; t)$ is $\alpha_t$-strongly convex, $\beta_t$-smooth, $L_t$-Lipschitz, and has condition number $\kappa_t = \frac{\beta_t}{\alpha_t}$. The step size is $\eta = \frac{\alpha_t}{4\beta_t^2}$. If initial iterate lies in original convex region: $x_0 \in \mathcal{R}_{SC}(t)$ (Theorem 4.3), with probability at least $(1 - \delta)$, the error satisfies:*

$$\|x_k - x^*\|^2 \leq \|x_t^* - x^*\|^2 + (1 - \frac{1}{4\kappa_t^2})^k \|x_0 - x^*\|^2 + \frac{4(K_t^2 + \sigma_t^2)}{\delta \, \alpha_t}(\frac{t}{2\lambda} + \frac{1}{\alpha_t})$$

*where $\|x_t^* - x^*\|^2$ follows Theorem 4.4, $K_t = \frac{\lambda}{N}\left(M_{-\frac{1}{2}}t^{-\frac{1}{2}} + M_0 \frac{L_t}{\lambda} + M_{\frac{1}{2}}\left(\frac{L_t}{\lambda}\right)^2 t^{\frac{1}{2}}\right)$ and $\sigma_t^2 = \frac{\lambda^2}{N}\left(V_{-1}t^{-1} + V_0\left(\frac{L_t}{\lambda}\right)^2 + V_1\left(\frac{L_t}{\lambda}\right)^4 t\right)$, where $M_{-1/2}, M_0, M_{1/2}$ and $V_{-1}, V_0, V_1$ are the dimensional-independent parameters come from former gradient estimator bounds in Theorem C.1.*

Theorem 4.5 indicates that the SBO update in (6) enjoys a *linear convergence rate* to an approximate optimizer, with residual error controlled by the smoothing level and estimator error. In practice, we choose step size: $\eta = \frac{1}{4\kappa_0 t}$ when $t \gg D_\tau$, and $\eta = \frac{1}{4\beta_0}$ otherwise. Different from previous asymptotic analysis (Iwakiri et al., 2022), Theorem 4.5 provides a *conditional non-asymptotic convergence* result for SBO on the smoothed nonconvex landscape, which reveals several practical insights in algorithm design and hyperparameter selection: Firstly, *greater sample size $N$* reduces both the bias and variance of the gradient estimator and thus the final residual error. Secondly, according to gradient estimator bounds Theorem C.1, there exists an *optimal temperature $\lambda^* = \Theta(\beta\sqrt{t})$* where the final residual error is minimized, which inspires a dual annealing strategy later in Algorithm 2. Note that Theorem 4.5 requires initial point start at $x_0 \in \mathcal{R}_{SC}(t)$ which could be small when $t$ is small. In the following section, we aim to release this restriction by starting from a larger kernel $t$ and anneal down it to control optimality gap.

### 4.4 Convergence Analysis with Varying Smoothing Parameter $t$

As stated in Section 4.2, a varying smoothing parameter $t$ balances the coverage of the smoothed convex region and the optimizer displacement of SBO. Extending the fixed-$t$ analysis requires a schedule that keeps the iterates inside the convex region. Theorem 4.6 shows that a diffusion-style *geometric time schedule* with $\lambda$-annealing maintains this invariant with a reduced sample size. Because both the residual error (Theorem 4.5) and optimizer displacement (Theorem 4.4) shrink with $t$, this yields convergence to $x^*$.

**Theorem 4.6** (Conditional Global Convergence of Dual-Level Annealing Algorithm). *Given initial iterate lies in original convex region: $x_{t_0} \in \mathcal{R}_{SC}(t_0)$ (Theorem 4.3), consider the update rule from time $t_0$ to $t_F$ with the following geometric time schedule:*

$$t_{m+1} = \begin{cases} \gamma t_m, & m < M_0, \\ t_F, & m \geq M_0, \end{cases} \tag{9}$$

$$x_{t_{m+1}} = x_{t_m} - \eta \nabla \hat{g}(x_{t_m}; t_m) \tag{10}$$

*where $t_F$ is the final sampling kernel, $M_0 = \lceil \log_\gamma(\frac{t_0}{t_F}) \rceil$ is the iteration at which the annealing stops, the sampling temperature is set adaptively as $\lambda_m = \beta\sqrt{t_m}$, and step size is set as $\eta = \frac{\alpha_m}{4\beta_m^2}$. With adaptive annealing $\lambda_m = \beta\sqrt{t_m}$, the required sample size $N$ is bounded by:*

$$N = \max\left\{ \frac{3\beta^2 M d}{E_0 C_E^2 D_\tau^2 \beta_1^2 \delta}(V_{-1} + V_0 + V_1), \frac{2M}{\delta d}\frac{\left(M_{-\frac{1}{2}} + M_0 \frac{L}{\beta} + M_{\frac{1}{2}}\frac{L^2}{\beta^2}\right)^2}{V_{-1} + V_0\frac{L^2}{\beta^2} + V_1\frac{L^4}{\beta^4}} \right\} \tag{11}$$

*where $V_{-1}, V_0, V_1$ and $M_{-1/2}, M_0, M_{1/2}$ are dimension-independent constants in the gradient estimator bounds from Theorem C.1. Without adaptive annealing $\lambda_m = \lambda_0$, let $\chi_0 := L/\lambda_0$. The required sample size $N$ is bounded by:*

$$N = \max\left\{ N_{acc}(t_0), N_{acc}(t_c), \max_m N_{bias}(t_m, \lambda_0) \right\}. \tag{12}$$

*Here* $N_{acc}(t) = \frac{3\lambda_0^2 M d}{E_0 C_E^2 D_\tau^2 \beta_1^2 \delta}(V_{-1}t^{-1} + V_0\chi_0^2 + V_1\chi_0^4 t)$ *and* $N_{bias}(t) = \frac{2M}{\delta d}(M_{-1/2}t^{-1/2} + M_0\chi_0 + M_{1/2}\chi_0^2 t^{1/2})^2(V_{-1}t^{-1} + V_0\chi_0^2 + V_1\chi_0^4 t)^{-1}$. *Then with probability at least* $1 - \delta$, *the dual-level annealing algorithm converges to*

$$\|x_M - x^*\|^2 \le \|x_F^* - x^*\|^2 + \left(1 - \frac{1}{4\kappa_F^2}\right)^{M-M_0}(C_E^2 D_\tau^2 + k_g t_F) + \frac{4\left(\frac{2M}{\delta}K_F^2 + \sigma_F^2\right)}{\delta\,\alpha_F}\left(\frac{t_F}{2\lambda_F} + \frac{1}{\alpha_F}\right) \quad (13)$$

*where* $k_g = C_E^2 \min\{\frac{\beta^2}{\lambda^2}, \frac{1}{\tau^4}\}$.

Theorem 4.6 provides a conditional non-asymptotic convergence result for SBO toward the global minimizer under the stated annealing and parameter assumptions. Compared to Theorem 4.5, it permits a wider range of initial points by choosing a large initial smoothing level $t_0$ and then tracking the iterate through the shrinking regions $\mathcal{R}_{SC}(t_m)$. Here $x_F^*$ denotes the minimizer of the final smoothed objective $g(\cdot; t_F)$ in the certified region. In the early stage of iteration, annealing mainly tracks the iterate inside a sequence of shrinking smoothed convex regions $\mathcal{R}_{SC}(t_m)$; once $t_m$ reaches a user-chosen threshold $t_F$ at which the optimizer displacement $\|x_F^* - x^*\|$ is acceptable, the algorithm switches to fixed-$t_F$ local search and the linear contraction term $(1 - \frac{1}{4\kappa_F^2})^{M-M_0}$ takes over. The choice of $t_F$ is therefore a precision hyperparameter: smaller $t_F$ tightens the displacement but requires a longer tracking phase. We do not claim the schedule is optimal in $t_F$; an optimal selection is left to future work. Algorithm-wise, Theorem 4.6 unifies two annealing strategies in the literature: simulated annealing (Bertsimas & Tsitsiklis, 1993) for temperature $\lambda$ and diffusion annealing (Pan et al., 2024) for noise level $t$. Our results suggest the temperature $\lambda$ and noise level $t$ should be jointly scheduled to achieve the best convergence rate: in the high $t$ regime, larger temperature $\lambda$ is preferred to make the distribution less concentrated to encourage exploration; in the low $t$ regime, smaller temperature $\lambda$ is preferred to make the distribution more concentrated to encourage convergence to the exact minimizer.

**Algorithm Design.** Inspired by Theorem 4.6, we propose Diffusion-Inspired Dual Annealing (DIDA), featuring the dual annealing strategy detailed in Algorithm 2: *smoothing annealing* over $t$ (Algorithm 2) and *temperature annealing* over $\lambda$ (Algorithm 2). For *smoothing annealing* over $t$, we approximate $t_m = \gamma^m t_0$, where $\gamma \in (0, 1)$ is a hyper-parameter. For *temperature annealing* over $\lambda$, we follows $\lambda_m = \beta\sqrt{t_m}$ with an approximated Lipschitz constant $\beta^2 \approx \frac{\mathrm{Var}[f(x_t)]}{\mathrm{Var}[x_t]} = \frac{\mathrm{Var}[f(x_t)]}{t_m}$, where $\mathrm{Var}[f(x_t)]$ is the variance of the sampled function values; this keeps $\chi \approx \frac{L}{\beta}$ constant across stages. The approximated Lipschitz leads to a simplified temperature schedule $\lambda_m = \sqrt{\mathrm{Var}[f(x_t)]}$.

---

**Algorithm 2:** Diffusion-Inspired Dual Annealing for Zeroth-Order Optimization

**Input:** initial noise $T$, final noise $t_F$, initial guess $x_T$, sample size $N$, iteration number $M$, annealing rate $\gamma$

1 Initialize $x_0 \leftarrow x_T$, $t_0 \leftarrow T$;
2 **for** $m = 1$ *to* $M$ **do**
3    $t_m = \gamma \cdot t_{m-1}$ ;            // Smoothing annealing
4    Draw samples $\{y_i\}_{i=1}^N \sim P_{t_m|0}(\cdot \mid x_m)$;
5    $\lambda_m = \sqrt{\mathrm{Var}[f(y_i)]}$ ;        // Temperature annealing
6    Estimate $\nabla_x g^{(0)}(x; t)$ with Equation (6);
7    Update: $x_{m+1} \leftarrow x_m - \frac{1}{4}t_m \cdot \nabla_x g^{(0)}(x_m; t_m)$;
8 **return** $x_M$

---

**Practical hyperparameters under zeroth-order access.** Although Theorem 4.6 is stated in terms of $\alpha, \beta, L$ and the certified switching threshold $t_F$, Algorithm 2 does not require estimating any of these constants. The schedule is fully specified by the user-chosen geometric ratio $\gamma$, total iteration number $M$, final smoothing $t_F$ and initial smoothing $t_0$. The temperature $\lambda_m = \sqrt{\mathrm{Var}[f(y_i)]}$ is computed from the current sample batch. The theorem constants $\alpha, \beta, L$ are therefore analytical objects used only to certify convergence; they are not inputs to the algorithm.

## 4.5 Implications for Diffusion Models

As shown in Section 3, the ODE form of the reverse process in diffusion can be viewed as a gradient flow on log density $-\log p(x; t)$, which is exactly $g(x, t)$ up to a constant. Therefore, the diffusion model can be viewed as descent of learned gradients on the smoothed objective, whose convexity is improved over $t$ as stated in Section 4.2. This optimization perspective offers a conceptual lens for interpreting noise-conditioned

diffusion sampling as descent on a $t$-dependent landscape; we present it as an implication of our theory rather than a complete empirical characterization of diffusion-model behavior.

**Guidance makes convex region more dominant.** Classifier-free guidance has been widely adopted to improve the sample quality of diffusion models (Jeon et al., 2025; Ho & Salimans, 2022). Theorem 4.3 offers a theoretical insight for diffusion models: the more concentrated the initial distribution (i.e. the smaller sub-Gaussian parameter), the faster the convergence. In classifier-free guidance, the original multi-modal distribution is concentrated to a single mode by conditioning on a class, where tuning the weight of the guidance can trade off the sample quality and sample diversity. From optimization perspective, the more smoothed the landscape is, the larger the contraction rate would be, leading to faster convergence and better tolerance to the noise in score estimation.

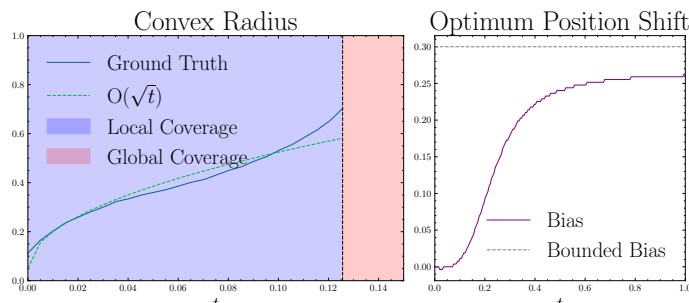

Figure 2: Coverage and optimizer displacement for the bounded-domain GMM-shaped visualization in Figure 1, where $g(x,t) = -\log \rho_t(x)$ for an unnormalized mixture $\rho_t$. The coverage expansion and bounded optimizer shift match Theorems 4.3 and 4.4.

## 5 Experimental Results

**Abalating the role of dual-annealing.** In these tables, "cost" denotes the optimized objective value after the evaluation budget, averaged over independent runs with standard deviations reported; lower values are better. For black-box benchmarks, each method evaluates candidates $y_{m,i}$ over optimization steps $m$ and samples $i$, and we report $\text{cost} = \min_{m,i} f(y_{m,i})$; equivalently, the implementation maximizes $J = -f$ and

| Task/Env. | DIDA(ours) | CEM (Rubinstein & Kroese, 2004) | CMA-ES (Akimoto et al., 2012) |
|---|---|---|---|
| *Blackbox optimization* | | | |
| Ackley (d=200) | **3.1 ±0.1** | 14.3 ±0.1 | 14.2 ±0.1 |
| Ackley (d=400) | **4.4 ±0.2** | 14.7 ±0.1 | 14.6 ±0.0 |
| Ackley (d=800) | **6.0 ±0.1** | 14.9 ±0.0 | 14.8 ±0.0 |
| Levy (d=200) | **11.8 ±2.0** | 744.3 ±23.7 | 744.3 ±23.7 |
| Levy (d=400) | **53.6 ±5.0** | 1567.4 ±28.2 | 1567.4 ±28.2 |
| Levy (d=800) | **202.5 ±11.3** | 3212.5 ±24.8 | 3212.5 ±24.8 |
| Rastrigin (d=200) | **1703.3 ±65.0** | 3644.2 ±32.0 | 3648.8 ±43.4 |
| Rastrigin (d=400) | **3782.1 ±80.3** | 7478.4 ±76.0 | 7478.4 ±76.0 |
| Rastrigin (d=800) | **8337.7 ±132.9** | 15231.6 ±116.9 | 15231.6 ±116.9 |
| *Trajectory optimization* | | | |
| ant | **0.032 ±0.080** | 0.649 ±0.101 | 0.879 ±0.177 |
| halfcheetah | **0.414 ±0.042** | 0.998 ±0.008 | 0.995 ±0.006 |
| hopper | **0.623 ±0.007** | 0.861 ±0.006 | 0.929 ±0.010 |
| humanoidrun | **0.298 ±0.059** | 0.973 ±0.008 | 0.989 ±0.017 |
| humanoidstandup | **0.781 ±0.025** | 0.876 ±0.000 | 0.876 ±0.000 |
| humanoidtrack | **0.845 ±0.009** | 1.015 ±0.002 | 1.022 ±0.008 |
| pushT | **0.834 ±0.034** | 0.960 ±0.032 | 1.028 ±0.031 |
| walker2d | **0.352 ±0.062** | 0.850 ±0.001 | 0.848 ±0.001 |

Table 2: Summary of optimized cost comparison for DIDA, CEM, and CMA-ES. Full comparison can be found in the Appendix (Table 4).

reports $-\max_{m,i} J(y_{m,i})$. For trajectory-optimization tasks, cost is the task-specific normalized trajectory objective returned by the benchmark evaluator.

To validate the coverage–displacement tradeoff in Theorems 4.3 and 4.4, we visualize the landscape and the coverage/optimizer displacement for a 1-d Gaussian Mixture model in Figures 1 and 2 and for the checkerboard function in Figure 3. The coverage expansion rate and optimizer displacement are within the bound of our theory. The controlled GMM visualization is designed to illustrate the theorem-covered mechanism, while the checkerboard, high-dimensional black-box functions, and trajectory-optimization tasks are empirical stress tests beyond the strict sufficient assumptions. For the 1D GMM visualization, we use an unnormalized GMM-shaped density $\rho_t(x) = \sum_{i=1}^{128} s_i \exp\left(-\frac{(x-\mu_i)^2}{2(t+3\cdot10^{-3})}\right) + 10^{-2}$ and plot $g(x,t) = -\log \rho_t(x)$ on a bounded domain; normalization is unnecessary because the displayed objective is defined up to an additive constant. For the checkerboard visualization, we use $\rho_t(x) = \sum_i s_i \exp\left(-\frac{\|x-\mu_i\|^2}{2t}\right) + 10^{-2}$ and $g(x,t) = -\log \rho_t(x)$, with centers selected from alternating cells of a 2D grid.

In Table 2, we evaluate these baselines and DIDA on high-dimensional black-box optimization and contact-rich trajectory optimization tasks. DIDA outperforms all baselines with a clear margin thanks to its dual annealing strategy, demonstrating the effectiveness of our theoretical prediction.

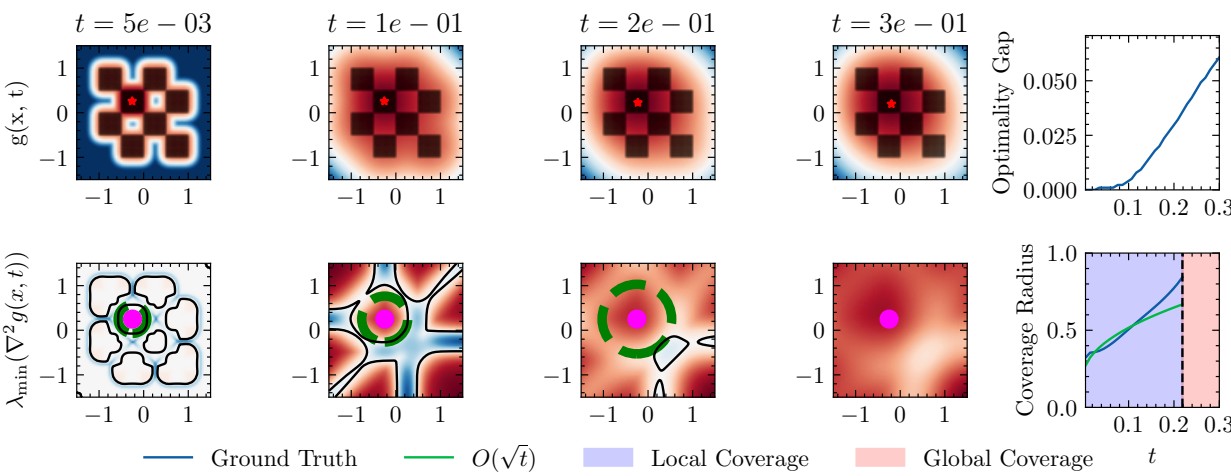

Figure 3: The smoothed checkerboard landscape and its numerical coverage/optimizer-displacement estimate over different $t$. The top row visualizes $g(x,t) = -\log\rho_t(x)$ and the bottom row visualizes the Hessian-based local convexity certificate. The purple dot denotes the reference optimizer $x^* = (-0.25, 0.25)$, the red star denotes the grid minimizer $x^*_t$, and the green dashed circle is the largest ball centered at $x^*$ before the estimated zero-level contour of $\lambda_{\min}(\nabla^2 g(x,t))$. The coverage radius is computed on a $2048 \times 2048$ grid by evaluating Hessian eigenvalues and measuring the distance from $x^*$ to the nearest zero-level contour; if no zero crossing is found in the plotted domain, the domain is marked as globally covered.

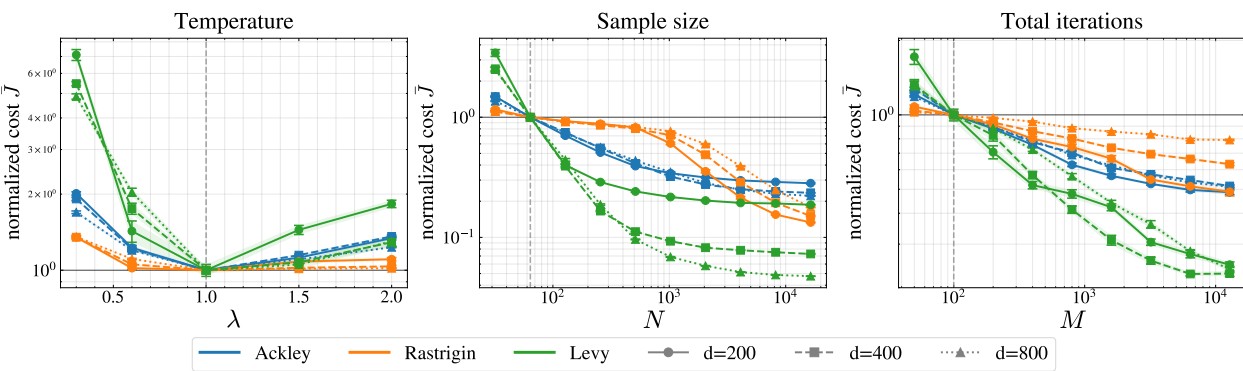

Figure 4: Sensitivity of DIDA to $\lambda$, $N$, and $M$ on the black-box benchmarks. We plot normalized cost to our default parameter choice. $\lambda = 1$ is robust across functions, and cost plateaus at $N \gtrsim 4096$ and $M \gtrsim 6400$.

**Sensitivity Study.** We sweep the temperature $\lambda$, sample size $N \in \{32, \dots, 16384\}$, and iteration count $M \in \{50, \dots, 12800\}$ on the black-box benchmarks (Figure 4). $\lambda = 1$ is robust across all functions, balancing the SNIS variance against the optimizer-displacement bias of Theorem 4.4. Cost decreases near-monotonically with $N$ and $M$, consistent with the $\frac{1}{\sqrt{N}}$ SNIS rate (Theorem C.1) and the per-stage contraction (Theorem 4.6); both knobs reach a plateau, e.g. Ackley $d = 800$ drops from 6.06 to 1.34 (4.5×) as $N$ grows from 64 to 16384. The default $(N, M) = (64, 100)$ sits near the knee and beats every baseline in Table 2.

## 6  Conclusion and Future Work

This paper conducts a comprehensive study on the non-asymptotic convergence behavior of SBO algorithms through the lens of diffusion-style smoothing. Based on our bias-coverage tradeoff analysis, we propose a new SBO algorithm, DIDA, demonstrating strong empirical performance. Future work includes extending our

theoretical analysis to function with multiple optima and applying our landscape analysis to diffusion models to improve sampling efficiency.

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

# Appendix

## A  Notation

| Symbol | Meaning | Definition |
|--------|---------|------------|
| $x$ | decision variable | $x \in \mathbb{R}^d$ |
| $f(x)$ | original objective function | $f : \mathbb{R}^d \to \mathbb{R}$ |
| $x^*$ | global minimizer of $f(x)$ | $f(x^*) = \min_x f(x)$ |
| $p_0(x)$ | target distribution density | $p_0(x) = \frac{e^{-f(x)/\lambda}}{\int e^{-f(x)/\lambda}dx}$ |
| $P_0(\cdot)$ | target distribution measure | Associated with density $p_0(x)$ |
| $k(\cdot; t)$ | kernel function | $k(\cdot; t) = (2\pi t)^{-d/2} e^{-\|\cdot\|^2/2t}$ |
| $p(x; t)$ | smoothed distribution | $p(x; t) = (p_0(\cdot) * k(\cdot; t))(x)$ |
| $g(x; t)$ | smoothed objective function | $g(x; t) = -\lambda \log p(x; t)$ |
| $x_t^*$ | minimizer of $g(x; t)$ | $g(x_t^*; t) = \min_x g(x; t)$ |
| $p_{t\mid0}(x \mid y)$ | forward distribution | $p_{t\mid0}(x \mid y) = k(x - y; t)$ |
| $p_{0\mid t}(y \mid x)$ | backward distribution | $p_{0\mid t}(y \mid x) = \frac{p_0(y)p_{t\mid0}(x\mid y)}{p(x;t)}$ |
| $\lambda$ | temperature parameter | $\lambda > 0$ |
| $\alpha$ | local strong convexity of $f(x)$ | Constant $\alpha > 0$ |
| $\beta$ | local smoothness of $f(x)$ | Constant $\beta > 0$ |
| $\kappa_0$ | local condition number of $f(x)$ | $\kappa_0 = \beta/\alpha$ |
| $\nabla_x g(x; t)$ | gradient of smoothed objective function | $\frac{\lambda}{t}\left(x - \mathbb{E}_{p_{0\mid t}}[y \mid x]\right)$ |
| $\nabla_x g^{(0)}(x; t)$ | zeroth-order gradient estimator | $\frac{\lambda}{t}\left(x - \frac{\sum_i w_i y_i}{\sum_i w_i}\right),\ w_i = e^{-f(y_i)/\lambda}$ |
| $\mathcal{N}(\mu, \Sigma)$ | Gaussian distribution | density $\propto \exp\left(-\frac{1}{2}(x-\mu)^\top \Sigma^{-1}(x-\mu)\right)$ |
| $\mathcal{SG}(\mu, \sigma^2)$ | scalar sub-Gaussian tail proxy | $\sigma^2 > 0$ is scalar; $P(\|X - \mu\| \geq a) \leq \exp\left(-\frac{a^2}{2\sigma^2}\right)$ for $a \geq 0$ |
| $D_s$ | determined by sub-Gaussian tail parameters | $D_s = \tau\sqrt{\log(1/P_{\text{out}})}$ |

Table 3: Notation used in the main text and appendix.

### A.1  Important Equations

**Proposition A.1** (Smoothed Function Properties). *The smoothed function $g(x; t)$ is defined as:*

$$g(x; t) = -\lambda \log\left(p_0(\cdot) * k(\cdot; t)\right)(x)$$

*Its gradient and Hessian have the following forms:*

***Gradient Equations:***

$$\nabla_x g(x; t) = \mathbb{E}_{y \sim p_{0\mid t}}\left[\nabla f(y) \mid x\right] = \int \nabla f(y) p_{0\mid t}(y \mid x)dy \tag{14}$$

$$\nabla_x g(x; t) = -\lambda \frac{\nabla_x p(x; t)}{p(x; t)} = \frac{\lambda}{t}\left(x - \mathbb{E}_{y \sim p_{0\mid t}}[y \mid x]\right) \tag{15}$$

$$\tag{16}$$

***Hessian Equations:***

$$\nabla_x^2 g(x; t) = \mathbb{E}_{y \sim p_{0\mid t}}\left[\nabla_x^2 f(y) \mid x\right] - \frac{1}{\lambda} Cov_{y \sim p_{0\mid t}}\left[\nabla_x f(y) \mid x\right] \tag{17}$$

$$\nabla_x^2 g(x; t) = \frac{\lambda}{t^2}\left(x\mathbb{E}_{y \sim p_{0\mid t}}[\nabla f(x) \mid y] - \mathbb{E}_{y \sim p_{0\mid t}}[x\nabla f(x) \mid y]\right) \tag{18}$$

$$\nabla_x^2 g(x; t) = \frac{\lambda}{t^2}\left(tI - Cov_{y \sim p_{0\mid t}}[y \mid x]\right) \tag{19}$$

**Lower Bound via Brascamp-Lieb Inequality**

Recall the *Matrix Brascamp-Lieb inequality* Bakry et al. (2014):

**Lemma A.2.** *For a probability measure $d\mu = e^{-W(y)}dy$ on $\mathbb{R}^n$ with strictly convex potential $W : \mathbb{R}^n \to \mathbb{R}$, we have for any smooth vector-valued function $h : \mathbb{R}^n \to \mathbb{R}^n$:*

$$\mathrm{Cov}_\mu(h) \preceq \int_{\mathbb{R}^n} (\nabla^2 W(y))^{-1} \nabla h(y) \nabla h(y)^\top \, d\mu(y).$$

# B  Landscape Analysis

## B.1  Additional Corollaries for Theorem 4.3

Here we first recap the main result about the landscape.

**Theorem 4.3** (Strongly Convex Bound). *Under Assumption 4.1 and Assumption 4.2, let d denote the dimension of the space and define $\kappa_0 := \beta/\alpha$. Fix any $C_\alpha \in (0,1)$ and assume $\lambda \le \lambda_{\max}$ for some fixed $\lambda_{\max} > 0$. Then, for any $t > 0$, the function $g(x;t)$ is $\frac{C_\alpha \lambda}{t + \frac{\lambda}{\alpha}}$-strongly convex within the region*

$$\mathcal{R}_{SC}(t) := \left\{ x \in \mathbb{R}^d \ \middle| \ \|x - x^*\| \le C_E \min\left( \sqrt{\frac{t + \frac{\lambda}{\beta}}{\frac{\lambda}{\beta}}} \, , \, \sqrt{\frac{t + \tau^2}{\tau^2}} \right) D_\tau \right\} \tag{7}$$

*i.e., for all $x \in \mathcal{R}_{SC}(t)$ we have $\nabla^2 g(x;t) \succeq \frac{C_\alpha \lambda}{t + \frac{\lambda}{\alpha}} I$, provided the sufficient parameter conditions in Equation (20) hold. Here $D_\tau$, $\tau$, and $P_{out}$ are the sub-Gaussian/tail parameters from Assumption 4.1, and $C_E$ is the expansion-rate factor defining $\mathcal{R}_{SC}(t)$.*

We next record the explicit sufficient parameter conditions referenced in Theorem 4.3:

$$
\begin{aligned}
&\text{(i)} \quad D_\tau^2 \ge bd && \text{for some constant } b \ge \max\left\{ \frac{9}{4}\frac{\lambda_{\max}}{\beta}, \, \frac{1 - C_\alpha}{18}\kappa_0^2, \, \frac{81}{8e}\kappa_0^4 \right\}, \\
&\text{(ii)} \quad P_{\text{out}} \le \frac{1}{d}, \\
&\text{(iii)} \quad \tau \le \tau_{\max} := \frac{2}{3\sqrt{3}}\sqrt{1 - C_\alpha}\,\kappa_0, \\
&\text{(iv)} \quad C_E^2 \le \frac{\beta}{16\lambda_{\max}b}\frac{\log d}{d}, \\
&\text{(v)} \quad \lambda \le 2\alpha
\end{aligned}
\tag{20}
$$

*Remark* B.1 (Restrictions in the sufficient regime). Equation (20) is a sufficient, restrictive high-dimensional regime for Theorem 4.3; it is not claimed to be necessary. In particular, since $D_\tau^2 \ge bd$ and

$$b \ge \max\left\{ \frac{9}{4}\frac{\lambda_{\max}}{\beta}, \frac{1 - C_\alpha}{18}\kappa_0^2, \frac{81}{8e}\kappa_0^4 \right\},$$

the theorem requires

$$D_\tau^2 \ge \frac{9}{4}\frac{\lambda_{\max}}{\beta}d, \qquad D_\tau^2 \ge \frac{1 - C_\alpha}{18}\kappa_0^2 d, \qquad D_\tau^2 \ge \frac{81}{8e}\kappa_0^4 d.$$

The regime also requires

$$P_{\text{out}} \le \frac{1}{d}, \qquad \tau \le \frac{2}{3\sqrt{3}}\sqrt{1 - C_\alpha}\kappa_0, \qquad C_E^2 \le \frac{\beta}{16\lambda_{\max}b}\frac{\log d}{d}, \qquad \lambda \le \min\{\lambda_{\max}, 2\alpha\}.$$

These conditions collect the proof requirements that the tail mass is small, the local convex neighborhood is large enough relative to dimension and condition number, and the expansion factor $C_E$ is small enough. The proof uses these bounds for sufficiently large $d$, for example to ensure $C_E \le 1/3$, $P_{\text{out}} \le P_{\text{in}}$, $D_s < D_\tau$, and the Gaussian lower-tail bound used in (73).

### B.2 Control Examples Satisfying or Motivating the Assumptions

**Example 1: linear control with localized Gaussian state penalties.** We give a concrete two-dimensional example showing that Assumptions 4.1 and 4.2 can hold while the objective is nonconvex. The example is a scalar finite-horizon linear-control problem whose decision variable is the control sequence $u = (u_0, u_1)$. Let

$$x_0 = 0, \qquad x_1 = ax_0 + bu_0, \qquad x_2 = ax_1 + bu_1,$$

and choose $a = 0.2$ and $b = 1$, so $x_1 = u_0$ and $x_2 = 0.2u_0 + u_1$. Define

$$J(u_0, u_1) = \frac{q}{2}(x_1^2 + x_2^2) + \frac{r}{2}(u_0^2 + u_1^2) + \sum_{j=1}^{3} \epsilon_j \|z(u)\|^2 \exp\left(-\frac{\|z(u) - z_j\|^2}{2\sigma_j^2}\right),$$

where $z(u) = (x_1, x_2)$, $q = 0.1$, $r = 1$,

$$z_1 = (0.9, 0.75), \quad z_2 = (-0.9, 0.70), \quad z_3 = (0.75, -0.85),$$

and

$$(\epsilon_1, \epsilon_2, \epsilon_3) = (0.14, 0.126, 0.119), \qquad (\sigma_1, \sigma_2, \sigma_3) = (0.14, 0.14, 0.126).$$

This is a quadratic LQR core plus bounded localized Gaussian/RBF state penalties. Such exponential potential terms are standard smooth soft-obstacle or keep-out costs in artificial-potential-field and MPC-style motion planning (Khatib, 1986; Cho et al., 2018; Pae et al., 2021; Wang et al., 2022); multiple terms simply model multiple obstacles, terrain features, or contact regions.

The quadratic core has Hessian

$$H_{\text{LQR}} = \begin{bmatrix} r + q(1 + a^2) & qa \\ qa & r + q \end{bmatrix},$$

whose eigenvalues are approximately 1.082 and 1.122, so the base landscape is close to circular in the $u_0, u_1$ plane. The localized penalty is exponentially small around the optimizer, and direct evaluation gives a positive minimum Hessian eigenvalue throughout $\{u : \|u - u^*\| < D_\tau\}$ with $D_\tau = 0.25$, verifying Assumption 4.2. The same objective is nevertheless nonconvex. The centers $z_1, z_2, z_3$ correspond to $(0.9, 0.57)$, $(-0.9, 0.88)$, and $(0.75, -1.0)$ in control space, where the minimum Hessian eigenvalues are approximately $-10.5$, $-8.8$, and $-10.4$, certifying compact negative-curvature patches.

Finally, the localized penalty $\psi(u) = \sum_{j=1}^{3} \epsilon_j \|z(u)\|^2 \exp(-\|z(u) - z_j\|^2/(2\sigma_j^2))$ is bounded:

$$0 \le \psi(u) \le C_\psi < \infty.$$

Writing the quadratic LQR core as $Q_{\text{LQR}}(u)$, we have

$$e^{-C_\psi/\lambda} e^{-Q_{\text{LQR}}(u)/\lambda} \le e^{-J(u)/\lambda} \le e^{-Q_{\text{LQR}}(u)/\lambda}.$$

Thus the Boltzmann density $p_0(u) \propto \exp(-J(u)/\lambda)$ is sandwiched between constant multiples of the Gaussian-like density induced by the positive-definite quadratic LQR core. Its tails are therefore sub-Gaussian for a suitable tail parameter $\tau$, verifying Assumption 4.1. Since $\nabla^2 J$ has negative eigenvalues in the three localized patches, the same density is not log-concave; this is precisely the setting where global log-concavity is too restrictive but the paper's tail-plus-local-regularity assumptions still apply.

**Example 2: nonlinear residual control landscape from prior work.** As a separate control example from the literature, Qu et al. (2020) consider the scalar dynamics

$$x_{t+1} = 0.5x_t + u_t + f(x_t), \qquad f(x) = \frac{0.01x}{1 + 0.9\sin x},$$

with $x_0 = 50$, $Q = 10$, $R = 1$, and a linear feedback policy $u_t = -Kx_t$. Their Figure 1 shows that this small nonlinear residual can create a policy-cost landscape $C(K)$ with many local minima, even though the nominal

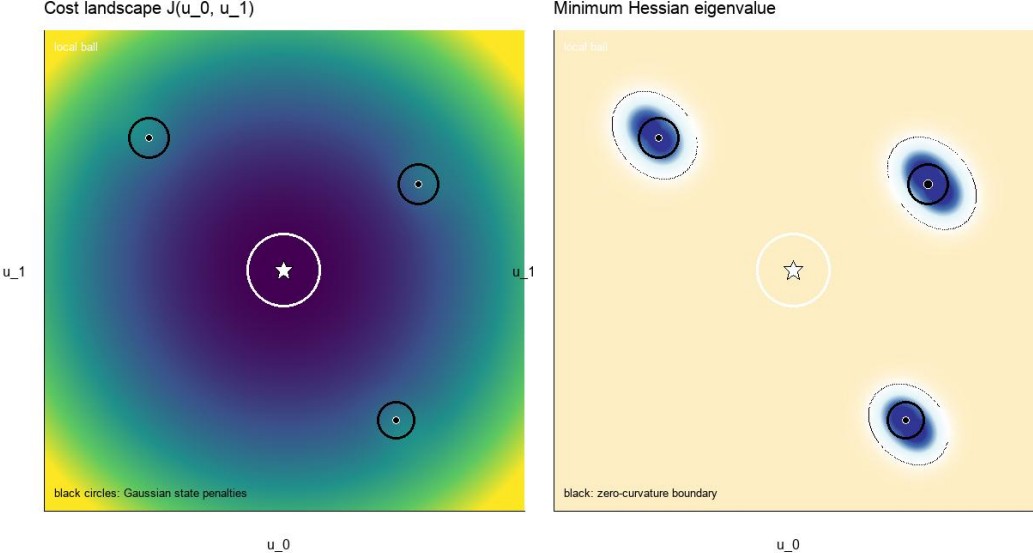

Figure 5: A scalar linear-control objective satisfying Assumptions 4.1 and 4.2 while remaining nonconvex in the control variables $u_0, u_1$. Left: contours of the finite-horizon cost $J(u_0, u_1)$, with black circles marking localized Gaussian/RBF state penalties. Right: the minimum Hessian eigenvalue is positive near $u^*$ but negative in compact patches around the soft obstacles. The positive-definite quadratic LQR core preserves Gaussian-type tails of the Boltzmann density.

linear part is a one-dimensional LQR system. For the finite-horizon cost $C_H(K) = \sum_{t=0}^{H-1}(Qx_t^2 + Ru_t^2)$ used in our recreation, the first control term already gives

$$C_H(K) \geq Rx_0^2 K^2.$$

Thus the Boltzmann density proportional to $\exp(-C_H(K)/\lambda)$ has sub-Gaussian tails in $K$. The multiple local minima in the recreated curve show that this density is not log-concave. We recreate this Figure 1-style landscape from the stated dynamics in Figure 6.

## B.3 Proof for Theorem 4.3

In this subsection, we focus on the proof of Theorem 4.3. To start, the following lemmas are instrumental in establishing Theorem 4.3. They provide a detailed analysis of the smoothed landscape's properties, which underpins the main theorem's conclusions regarding the conditions for strong convexity and the expansion of this convex region.

**Lemma B.2** (Truncated sub-Gaussian moments). *Let $Y \geq 0$ be a random variable. We say that $Y$ is (one-sided) sub-Gaussian with parameter $\tau^2$, and write $Y \sim \mathcal{SG}(0, \tau^2)$, if its tail satisfies*

$$\mathbb{P}(Y > r) \leq \exp\left(-\frac{r^2}{\tau^2}\right), \qquad \forall r \geq 0, \tag{21}$$

*for some $\tau > 0$. Then, for any $a \geq 0$,*

$$\mathbb{E}[Y \mid Y > a] \leq a + \frac{\tau^2}{a + \sqrt{a^2 + \frac{4\tau^2}{\pi}}}, \tag{22}$$

$$\mathbb{E}[Y^2 \mid Y > a] \leq a^2 + \tau^2. \tag{23}$$

*Furthermore, if the exact tail probability at level $a$ is known,*

$$p_a := \mathbb{P}(Y > a),$$

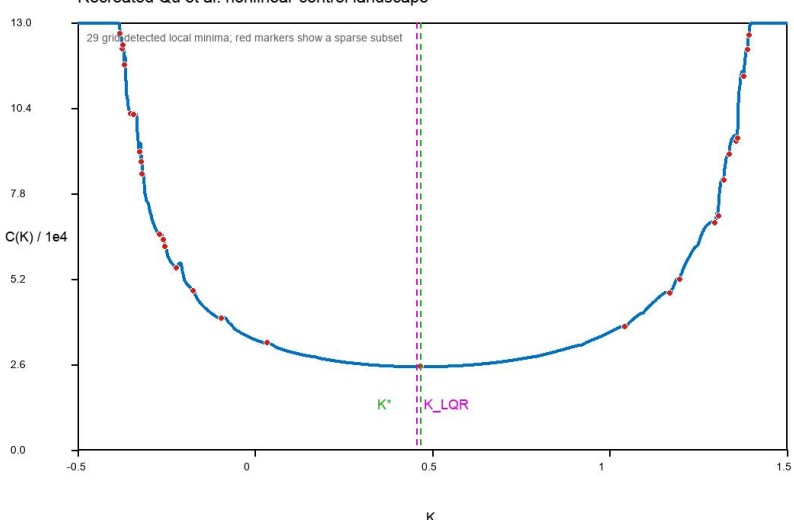

Figure 6: Recreation of the nonlinear-control policy-cost landscape from Qu et al. (2020), Example 1, using the stated scalar dynamics $x_{t+1} = 0.5x_t + u_t + 0.01x_t/(1 + 0.9\sin x_t)$, $u_t = -Kx_t$, $x_0 = 50$, $Q = 10$, and $R = 1$. The finite-horizon cost has quadratic tails in $K$ from the first control term, while the curve illustrates many local minima in the policy parameter $K$.

*it is convenient to introduce the* effective tail radius

$$D := \tau\sqrt{\log\left(\tfrac{1}{p_a}\right)} \qquad (so\ that\ \ e^{-D^2/\tau^2} = p_a),$$

*which satisfies $D \geq a$ by* (21). *In terms of $D$, we have*

$$\mathbb{E}[Y \mid Y > a] \leq D + \frac{\tau^2}{D + \sqrt{D^2 + \frac{4\tau^2}{\pi}}} \ \leq\ D + \tau, \tag{24}$$

$$\mathbb{E}[Y^2 \mid Y > a] \leq D^2 + \tau^2. \tag{25}$$

*Proof.* We start from the tail-integral identities for nonnegative random variables. For any $Z \geq 0$, Tonelli's theorem yields

$$\mathbb{E}[Z] = \int_0^\infty P(Z > r)\, dr, \qquad \mathbb{E}\left[Z^2\right] = \int_0^\infty 2r\, P(Z > r)\, dr.$$

Applying these to $(Y - a)_+$ gives the standard representations

$$\mathbb{E}[Y \mid Y > a] = a + \frac{\int_a^\infty P(Y > r)\, dr}{P(Y > a)}, \tag{26}$$

$$\mathbb{E}\left[Y^2 \mid Y > a\right] = a^2 + \frac{\int_a^\infty 2r\, P(Y > r)\, dr}{P(Y > a)}. \tag{27}$$

**Step 1: Bounds in terms of the tail at level $a$.** Assume $Y \sim \mathcal{SG}(0, \tau^2)$ in the sense of (21). Then for all $r \geq a$, $P(Y > r) \leq e^{-r^2/\tau^2}$, and therefore

$$\int_a^\infty P(Y > r)\, dr \leq \int_a^\infty e^{-r^2/\tau^2}\, dr, \qquad \int_a^\infty 2r\, P(Y > r)\, dr \leq \int_a^\infty 2r\, e^{-r^2/\tau^2}\, dr.$$

For the second integral we have the exact evaluation

$$\int_a^\infty 2r\, e^{-r^2/\tau^2}\, dr = \tau^2 e^{-a^2/\tau^2}.$$

For the first integral we use a Gaussian tail bound (Mills' ratio; e.g. (Laurent & Massart, 2000, Lemma 1.1)):

$$\int_a^\infty e^{-r^2/\tau^2}\, dr \le \frac{\tau^2 e^{-a^2/\tau^2}}{a + \sqrt{a^2 + \frac{4\tau^2}{\pi}}}. \tag{28}$$

Substituting these bounds into (26)–(27) yields

$$\mathbb{E}\left[Y \mid Y > a\right] \le a + \frac{\tau^2 e^{-a^2/\tau^2}}{\left(a + \sqrt{a^2 + \frac{4\tau^2}{\pi}}\right) P(Y > a)}, \qquad \mathbb{E}\left[Y^2 \mid Y > a\right] \le a^2 + \frac{\tau^2 e^{-a^2/\tau^2}}{P(Y > a)}.$$

In the special case where the sub-Gaussian bound is tight at level $a$, namely $P(Y > a) = e^{-a^2/\tau^2}$, these simplify to (22)–(23).

**Step 2: General case via the effective tail radius.** Let $p_a := P(Y > a)$ and define $D := \tau\sqrt{\log(1/p_a)}$, so that $e^{-D^2/\tau^2} = p_a$. By (21), $p_a \le e^{-a^2/\tau^2}$, hence $D \ge a$. Moreover, for all $r \ge a$,

$$P(Y > r) \le \min\{p_a,\, e^{-r^2/\tau^2}\} = \begin{cases} p_a, & a \le r \le D, \\ e^{-r^2/\tau^2}, & r \ge D. \end{cases}$$

*First moment.* Using (26) and splitting the numerator at $D$ gives

$$\int_a^\infty P(Y > r)\, dr \le \int_a^D p_a\, dr + \int_D^\infty e^{-r^2/\tau^2}\, dr$$
$$= p_a(D - a) + \int_D^\infty e^{-r^2/\tau^2}\, dr.$$

Dividing by $p_a$ and applying (28) at $D$ (noting $e^{-D^2/\tau^2} = p_a$) yields

$$\mathbb{E}\left[Y \mid Y > a\right] \le a + (D - a) + \frac{1}{p_a}\int_D^\infty e^{-r^2/\tau^2}\, dr \le D + \frac{\tau^2}{D + \sqrt{D^2 + \frac{4\tau^2}{\pi}}},$$

which is (24). Finally, since $\sqrt{D^2 + \frac{4\tau^2}{\pi}} \ge \frac{2\tau}{\sqrt{\pi}}$, we have $D + \sqrt{D^2 + \frac{4\tau^2}{\pi}} \ge \frac{2\tau}{\sqrt{\pi}}$, and hence

$$\frac{\tau^2}{D + \sqrt{D^2 + \frac{4\tau^2}{\pi}}} \le \frac{\tau^2}{2\tau/\sqrt{\pi}} = \frac{\sqrt{\pi}}{2}\tau \le \tau,$$

so $\mathbb{E}\left[Y \mid Y > a\right] \le D + \tau$.

*Second moment.* Similarly, from (27),

$$\int_a^\infty 2r\, P(Y > r)\, dr \le \int_a^D 2r\, p_a\, dr + \int_D^\infty 2r\, e^{-r^2/\tau^2}\, dr$$
$$= p_a(D^2 - a^2) + \tau^2 e^{-D^2/\tau^2} = p_a(D^2 - a^2 + \tau^2).$$

Dividing by $p_a$ and plugging into (27) gives

$$\mathbb{E}\left[Y^2 \mid Y > a\right] \le a^2 + (D^2 - a^2 + \tau^2) = D^2 + \tau^2,$$

which proves (25). $\qquad\square$

Based on the above lemma, we now investigate bound for the posterior distribution $P_{0|t}$, which the basis of our analysis on the landscape.

**Lemma B.3** (Convex region covariance bound). *The conditional covariance of $y$ given $y \in B_\tau$ and $x_t$ is upper bounded by*

$$\text{Cov}_{[0|t]}[y \mid x_t, y \in B_\tau] \preceq \frac{t(\lambda/\alpha)}{t + (\lambda/\alpha)} \mathbf{I}$$

*Proof.* Fix $x_t$ and consider the posterior restricted to the local ball $B_\tau$:

$$d\mu_{x_t}^{\text{in}}(y) := \frac{\mathbf{1}_{B_\tau}(y) \, p_0(y) \, p_{t|0}(x_t \mid y)}{\int_{B_\tau} p_0(z) \, p_{t|0}(x_t \mid z) \, dz} \, dy.$$

This is precisely the conditional law of $y$ given $x_t$ and $y \in B_\tau$, so

$$\text{Cov}_{[0|t]}[y \mid x_t, y \in B_\tau] = \text{Cov}_{\mu_{x_t}^{\text{in}}}(y).$$

On $B_\tau$, its density is proportional to

$$\exp\left(-\frac{1}{\lambda}f(y) - \frac{1}{2t}\|y - x_t\|^2\right).$$

Equivalently, up to an additive constant, the potential is

$$W_{x_t}(y) = \frac{1}{\lambda}f(y) + \frac{1}{2t}\|y - x_t\|^2.$$

By Assumption 4.2, $f$ is $\alpha$-strongly convex on $B_\tau$. Hence, for every $y \in B_\tau$,

$$\nabla^2 W_{x_t}(y) = \frac{1}{\lambda}\nabla^2 f(y) + \frac{1}{t}\mathbf{I} \succeq \left(\frac{\alpha}{\lambda} + \frac{1}{t}\right)\mathbf{I}.$$

Applying the matrix Brascamp–Lieb inequality in Lemma A.2 to $h(y) = y$, for which $\nabla h(y) = \mathbf{I}$, gives

$$\text{Cov}_{\mu_{x_t}^{\text{in}}}(y) \preceq \int_{B_\tau} \left(\nabla^2 W_{x_t}(y)\right)^{-1} d\mu_{x_t}^{\text{in}}(y) \preceq \left(\frac{\alpha}{\lambda} + \frac{1}{t}\right)^{-1}\mathbf{I}.$$

The use of Brascamp–Lieb on the convex domain $B_\tau$ can be justified either by the convex-domain version of the inequality or by approximating the indicator of $B_\tau$ with a smooth convex barrier and taking the limit. Finally,

$$\left(\frac{\alpha}{\lambda} + \frac{1}{t}\right)^{-1} = \frac{t(\lambda/\alpha)}{t + \lambda/\alpha},$$

which proves the claim. $\square$

**Lemma B.4** (Sub-Gaussian region moment bound). *Recall the setup and notation in Assumption 4.1. The conditional first and second moments of the radius outside the sub-Gaussian region satisfy*

$$\mathbb{E}_{[0|t]}[\|y - x^*\|^2 \mid x_t, y \notin B_\tau] \leq \max\left((D_s + 2r_t')^2, D_s^2 + 3\tau'^2 + 4\tau'r_t'\right)$$

*and*

$$\mathbb{E}_{[0|t]}[\|y - x^*\| \mid x_t, y \notin B_\tau] \leq \max\left(D_s + 2r_t', D_s + 3\tau'\right)$$

*where $D_s := \tau\sqrt{\log(\frac{1}{P_{out}})}$ , $\tau'^2 = \frac{2\tau^2 t}{\tau^2 + 2t}$ and $r_t' = \frac{\tau^2\|x_t - x^*\|}{\tau^2 + 2t}$*

*Proof.* **Sketch**: Consider the following functional:

$$\xi_1(q_0, q_{t|0}) := \frac{\int_{y \notin B_\tau} \|y - x^*\| q_0(y) q_{t|0}(y|x_t) \, dy}{\int_{y \notin B_\tau} q_0(y) q_{t|0}(y|x_t) \, dy}. \tag{29}$$

$$\xi_2(q_0, q_{t|0}) := \frac{\int_{y \notin B_\tau} \|y - x^*\|^2 q_0(y) q_{t|0}(y|x_t) \, dy}{\int_{y \notin B_\tau} q_0(y) q_{t|0}(y|x_t) \, dy}. \tag{30}$$

By definition, $\xi_1(p_0, p_{t|0}) = \mathbb{E}_{[0|t]}[\|y - x^*\| \mid x_t, y \notin B_\tau]$ and $\xi_2(p_0, p_{t|0}) = \mathbb{E}_{[0|t]}[\|y - x^*\|^2 \mid x_t, y \notin B_\tau]$. In the following steps, we aim to show that

$$\sup_{q_0 \in \mathcal{Q}', q_{t|0} \in \mathcal{Q}_{t|}} \xi_1(q_0, q_{t|0}) \leq RHS_1 \tag{31}$$

$$\sup_{q_0 \in \mathcal{Q}', q_{t|0} \in \mathcal{Q}_{t|0}} \xi_2(q_0, q_{t|0}) \leq RHS_2 \tag{32}$$

where $RHS_1$ is the RHS for Equation (29) and $RHS_2$ is the RHS for Equation (30), and $p_0 \in \mathcal{Q}$ and $p_{t|0} \in \mathcal{Q}_{t|0}$. Here, $\mathcal{Q}$ and $\mathcal{Q}_{t|0}$ are sets of feasible probability densities that we will define later.

**Step 1: Reduction to a one-dimensional integral.** Fix any $p_0 \in \mathcal{Q}_0$. In this step, we explain how to bound

$$\sup_{q_{t|0} \in \mathcal{Q}_{t|0}} \xi_1(p_0, q_{t|0}) \quad \text{and} \quad \sup_{q_{t|0} \in \mathcal{Q}_{t|0}} \xi_2(p_0, q_{t|0}),$$

by reducing the optimization over $q_{t|0}$ to a one-dimensional problem in $\|y - x^*\|$. We view $\xi_1$ and $\xi_2$ as functionals of $(q_0, q_{t|0})$, where $q_0$ is a probability density on $\mathbb{R}^d$ and, for each fixed $x_t$, $q_{t|0}(\cdot|x_t)$ is a conditional density of $Y$ given $X_t = x_t$. In particular, the conditional first moment that we want to bound is $\xi_1(p_0, p_{t|0})$. Our first step is to transform this conditional expectation into a one-dimensional integral over the radial variable $\|y - x^*\|$. To this end, we will "replace" $p_{t|0}(\cdot)$ and $p_0(\cdot)$ by simpler radial densities that depend only on $\|y - x^*\|$; the precise meaning of this replacement will be made clear below.

*Replacing $p_{t|0}(\cdot)$.* The forward transition density $p_{t|0}$ satisfies the following bound:

$$(2\pi t)^{-d/2} \exp\left(-\frac{(\|y - x^*\| + \|x_t - x^*\|)^2}{2t}\right) \leq p_{t|0}(x_t|y) \leq (2\pi t)^{-d/2} \exp\left(-\frac{(\|y - x^*\| - \|x_t - x^*\|)^2}{2t}\right) \tag{33}$$

Motivated by this, we define the admissible class of conditional densities as follows:

$$\mathcal{Q}_{t|0} := \left\{ q_{t|0}(\cdot|x_t) \; \middle| \; \begin{array}{l} q_{t|0}(\cdot|x_t) \text{ is a Borel-measurable density on } \mathbb{R}^d \text{ for each } x_t, \\[4pt] \int_{\mathbb{R}^d} q_{t|0}(y|x_t) \, dy = 1, \\[4pt] (2\pi t)^{-d/2} \exp\left(-\frac{(\|y-x^*\|+\|x_t-x^*\|)^2}{2t}\right) \leq q_{t|0}(y|x_t) \\[4pt] \leq (2\pi t)^{-d/2} \exp\left(-\frac{(\|y-x^*\|-\|x_t-x^*\|)^2}{2t}\right), \quad \forall y \notin B_\tau \end{array} \right\}.$$

By construction, $p_{t|0}(\cdot|x_t) \in \mathcal{Q}_{t|0}$ for every $x_t$, and therefore

$$\xi_1(p_0, p_{t|0}) \leq \sup_{q_{t|0} \in \mathcal{Q}_{t|0}} \xi_1(p_0, q_{t|0}).$$

We now justify the worst-case choice of $q_{t|0}(\cdot|x_t)$ within the admissible envelope (33). Fix $q_0 = p_0$ and view

$$\xi_1(p_0, q_{t|0}) = \frac{\int_{y \notin B_\tau} \|y - x^*\| \, p_0(y) \, q_{t|0}(y|x_t) \, dy}{\int_{y \notin B_\tau} p_0(y) \, q_{t|0}(y|x_t) \, dy}$$

as a functional of $q_{t|0}$ under the pointwise constraints $q_-(y) \leq q_{t|0}(y|x_t) \leq q_+(y)$ on $B_\tau^c$, where

$$q_-(y) := (2\pi t)^{-d/2} \exp\left(-\frac{(\|y - x^*\| + \|x_t - x^*\|)^2}{2t}\right),$$

$$q_+(y) := (2\pi t)^{-d/2} \exp\left(-\frac{(\|y - x^*\| - \|x_t - x^*\|)^2}{2t}\right).$$

We next compute the functional derivative of $\xi_1$ with respect to $q_{t|0}(y|x_t)$:

$$\frac{\partial \xi_1(q_0, q_{t|0})}{\partial q_{t|0}(y|x_t)} = \frac{q_0(y)}{\int_{y \notin B_\tau} q_{t|0}(y|x_t)q_0(y)\,dy}(\|y - x^*\| - \xi_1(q_0, q_{t|0})) \tag{34}$$

Since $\xi_1$ is a ratio of two linear functionals in $q_{t|0}$, the extremizer over such box constraints is attained at an extreme point (a "bang–bang" choice): there exists a threshold $r^\star$ such that $q_{t|0}$ takes the lower envelope $q_-$ on $\{\|y - x^*\| < r^\star\}$ and the upper envelope $q_+$ on $\{\|y - x^*\| > r^\star\}$ (ties on the null set $\{\|y - x^*\| = r^\star\}$ are irrelevant). We therefore define

$$\tilde{p}_{t|0}(y|x_t) := \begin{cases} q_-(y), & D_\tau \leq \|y - x^*\| < r^\star, \\ q_+(y), & \|y - x^*\| \geq r^\star, \end{cases} \tag{35}$$

and set

$$\xi_1^\star := \xi_1(p_0, \tilde{p}_{t|0}) = \sup_{q_{t|0} \in \mathcal{Q}_{t|0}} \xi_1(p_0, q_{t|0}). \tag{36}$$

By construction, $p_{t|0}(\cdot|x_t) \in \mathcal{Q}_{t|0}$, hence $\xi_1(p_0, p_{t|0}) \leq \xi_1^\star$.

*Replacing $p_0(\cdot)$:* We now study the following functional for $q_0$:

$$J[q_0] = \xi_1(q_0, \tilde{p}_{t|0}) = \frac{\int_{y \notin B_\tau} \|y - x^*\| q_0(y)\tilde{p}_{t|0}(y|x_t)\,dy}{\int_{y \notin B_\tau} q_0(y)\tilde{p}_{t|0}(y|x_t)dy}.$$

We maximize this functional over $q_0$ subject to the following constraint set:

$$\mathcal{A} = \left\{ q_0 : [a, \infty) \to \mathbb{R}_{\geq 0} \;\middle|\; \begin{array}{l} q_0 \text{ is continuous and a.e. differentiable,} \\[4pt] \int_{y \notin B_\tau} q_0(y)dy = P_{\text{out}}, \quad q_0(y) > 0, \\[4pt] \int_{\|y - x^*\| > r} q_0(y)dy \leq \exp(-r^2/\tau^2), \quad \forall r > D_\tau \end{array} \right\},$$

to obtain the optimal $\tilde{p}_0$. This yields the upper bound

$$\xi_1^* \leq \xi_1(p_0, \tilde{p}_{t|0}) \leq \xi_1(\tilde{p}_0, \tilde{p}_{t|0}) = J[\tilde{p}_0].$$

To maximize $J[q_0]$, we formulate the problem using the method of Lagrange multipliers, treating $J$ as an auxiliary parameter. We introduce the following multipliers:

- $\nu(y) \geq 0$ for the sub-Gaussian bound $\int_{\|y - x^*\| > r} q_0(y)dy \leq \exp(-r^2/\tau^2)$,

- a scalar $\eta$ for the total mass constraint $\int_{y \notin B_\tau} q_0(y) = P_{\text{out}}$,

- and $\theta(y) \geq 0$ for the positivity constraint.

The Lagrangian is given by:

$$\mathcal{L}[q_0] = \int_{\|y - x^*\| > r} \left[ (\|y - x^*\| - J)\,\tilde{p}_{t|0}(y|x_t)\,q_0(y) + \nu(y) \int_{\|y' - x^*\| > r} \left( \exp(-r^2/\tau^2) - q_0(y') \right)dy' \right]dy$$

$$+ \eta \int_{y \notin B_\tau} q_0(y)\,dy + \int_{y \notin B_\tau} q_0(y)\theta(y)\,dy.$$

For the stationary condition, we consider a variation $q_0 \mapsto q_0 + \varepsilon h$ with $\int h = 0$ and $\int_{y \notin B_\tau} h = 0$,

$$0 = \frac{d}{d\varepsilon}\mathcal{L}[q_0 + \varepsilon h]\Big|_{\varepsilon=0} = \int_{\|y - x^*\| > a} \left[ (\|y - x^*\| - J)\tilde{p}_{t|0}(y|x_t) - \int_{\|t - x^*\| \leq \|y - x^*\|} \nu(t) + \eta + \theta(y) \right] h(y)\,dy,$$

hence

$$(\|y - x^*\| - J)\tilde{p}_{t|0}(y|x_t) - \int_{\|y' - x^*\| \le \|y - x^*\|} \nu(y')dy' + \eta + \theta(y) = 0$$

Since $\tilde{p}_{t|0}(y|x_t) > 0$ and the functional derivative changes sign, the stationary condition forces a *bang–bang* structure:

$$\begin{cases} \nu(y) = 0, & \theta(y) > 0, & \|y - x^*\| < \rho, \\ \nu(y) > 0, & \theta(y) = 0 & \|y - x^*\| \ge \rho, \end{cases} \tag{37}$$

Complementary slackness dictates that $\nu(y)$ and $\theta(y)$ cannot both be zero, as $(\|y - x^*\| - J)\tilde{p}_{t|0}(y|x_t)$ is strictly increasing in $\|y - x^*\|$. Thus, there exists a threshold $\rho$ such that $\nu(y) = 0, \theta(y) > 0$ when $\|y - x^*\| < \rho$, and $\nu(y) > 0, \theta(y) = 0$ when $\|y - x^*\| \ge \rho$. This corresponds precisely to the bang–bang structure required by the sub-Gaussian constraint. By further examining the constraint $\int_{D_\tau}^\infty q_0(y)dy = P_{\text{out}}$, we obtain $\rho = D_s$, where $D_s := \tau\sqrt{\log\left(\frac{1}{P_{\text{out}}}\right)}$.

This optimal configuration is illustrated in Figure 7, where the left segment $(D_\tau \le \|y - x^*\| < r^*)$ has smaller $\|y - x^*\|$ values and lower density (minimum $\tilde{p}_{t|0}$), while the right segment $(\|y - x^*\| \ge r^*)$ has larger $\|y - x^*\|$ values and higher density (maximum $\tilde{p}_{t|0}$). Therefore, after self-normalization, the region with larger $\|y - x^*\|$ values has larger density.

In conclusion, we have the following result for $\tilde{p}_0$ (here we give its CDF $\tilde{P}_0(\|y - x^*\| \ge r)$) and $\tilde{p}_{t|0}$ that achieve the maximum $\xi_1^{\star\star}$:

$$\tilde{P}_0(\|y - x^*\| \ge r) = \begin{cases} P_{\text{out}} & \text{if } D_\tau \le r \le D_s \\ \exp(-a^2/\tau^2) & \text{if } r > D_s \end{cases}, \tag{38}$$

$$\tilde{p}_{t|0}(y|x_t) \propto \begin{cases} \exp(-(\|y - x^*\| + \|x_t - x^*\|)^2/2t) & \text{if } D_\tau \le \|y - x^*\| \le r^* \\ \exp(-(\|y - x^*\| - \|x_t - x^*\|)^2/2t) & \text{if } \|y - x^*\| > r^* \end{cases} \tag{39}$$

where

$$\xi_1^\star \le \xi_1^{\star\star} := \sup_{q_0 \in \mathcal{Q}', q_{t|0} \in \mathcal{Q}_{t|}} \xi_1(q_0, q_{t|0}) = \xi_1\left(\tilde{p}_0, \tilde{p}_{t|0}\right) \tag{40}$$

With the above selected $\tilde{p}_0, \tilde{p}_{t|0}$, we can transform $\xi_1$ as follows:

$$\begin{aligned} \xi_1\left(\tilde{p}_0, \tilde{p}_{t|0}\right) &= \frac{\int_{y \notin B_\tau} \|y - x^*\| \tilde{p}_0(y)\tilde{p}_{t|0}(y \mid x_t) \, dy}{\int_{y \notin B_\tau} \tilde{p}_0(y)\tilde{p}_{t|0}(y \mid x_t) \, dy}. \\ &= \frac{\int_{r > D_\tau} \|r\| \int_{\|y - x^*\| = r} \tilde{p}_0(y)\tilde{p}_{t|0}(y \mid x_t) \, dy}{\int_{r > D_\tau} \int_{\|y - x^*\| = r} \tilde{p}_0(y)\tilde{p}_{t|0}(y \mid x_t) \, dy}. \end{aligned} \tag{41}$$

$$:= \frac{\int_{r > D_\tau} \|r\| \tilde{p}_{t|0}^{R_1}(r \mid x_t) \tilde{p}_0^R(r) dr}{\int_{r > D_\tau} \tilde{p}_{t|0}^{R_1}(r \mid x_t) \tilde{p}_0^R(r) dr}. \tag{42}$$

where in Equation (41), we rewrite the integral using radial coordinates centered at $x^*$. The integration proceeds by first integrating over the spherical surface $\{y : \|y - x^*\| = r\}$ for a fixed radius $r$, and then integrating over $r$. In Equation (42), we leverage the property that the optimized transition density $\tilde{p}_{t|0}(y|x_t)$ is assumed to depend only on the radius $r = \|y - x^*\|$, allowing it to be factored out of the surface integral. The remaining surface integral of the optimized prior $\tilde{p}_0(y)$ defines the radial prior density $\tilde{p}_0^R(r)$. Consequently, Equation (42) presents the final result as a simplified one-dimensional integral involving only these radial densities, $\tilde{p}_0^R(r)$ and $\tilde{p}_{t|0}^{R_1}(r)$ as defined below.

$$\tilde{p}_0^R(r) := \int_{\|y-x^*\|=r} \tilde{p}_0(y)dy = -\frac{d}{dr}\tilde{P}_0(\|y-x^*\| \geq r) = \begin{cases} \frac{2r}{\tau^2}\exp(-r^2/\tau^2) & \text{if } r \geq D_s \\ 0 & \text{if } D_\tau \leq r \leq D_s \end{cases},$$

$$\tilde{p}_{t|0}^{R_1}(r, x_t) :\propto \begin{cases} \exp(-(r+\|x_t-x^*\|)^2/2t) & \text{if } D_\tau \leq r \leq r^* \\ \exp(-(r-\|x_t-x^*\|)^2/2t) & \text{if } r \geq r^* \end{cases} \tag{43}$$

with $r = \|y - x^*\|$. To reach the maximum value, we must have $r^* = \xi_1^{\star\star}$, as the switching point is where $\xi_1 - \|y - x^*\|$ changes sign in the first-order condition. Thus, the analysis in Step 1 successfully reduces the multi-dimensional integrals for the first moment $\xi_1$ to one-dimensional integrals involving the radial densities $\tilde{p}_0^R(r)$ and $\tilde{p}_{t|0}^{R_1}(r)$, as given in Equation (42) and the equations above.

**Optimized Distribution**
**(Vertically Separated for Clarity)**

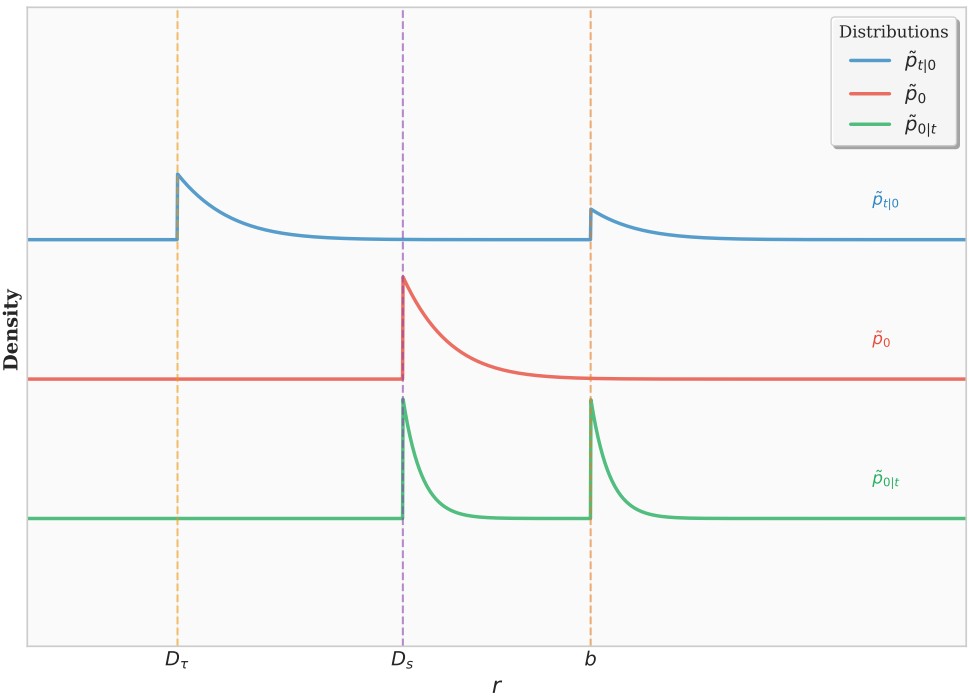

Figure 7: Modified 1-d density function illustration when functional takes maximum.

The same functional optimization framework applies to the second moment as well. While the specific optimal distributions $\tilde{p}_0$ and $\tilde{p}_{t|0}$ will differ (as $r^*$ depend on the threshold $\xi_2^{**}$ rather than $\xi_1^{**}$), the overall methodology yields a similar bounding inequality for

$$(\xi_2^{**})^2 := \sup_{q_{t|0} \in \mathcal{Q}_{t|0}} \xi_2(p_0, q_{t|0}) \tag{44}$$

$$= \frac{\int_{r>D_\tau} \|r\|^2 \tilde{p}_{t|0}^{R_2}(r)\tilde{p}_0^R(r)\,dr}{\int_{r>D_\tau} \tilde{p}_{t|0}^{R_2}(r)\tilde{p}_0^R(r)\,dr} \tag{45}$$

where $\tilde{p}_0^R(r)$ remains the same and

$$\tilde{p}_{t|0}^{R_2}(r) :\propto \begin{cases} \exp(-(r+\|x_t-x^*\|)^2/2t) & \text{if } D_s \leq r \leq r^{\star\star} \\ \exp(-(r-\|x_t-x^*\|)^2/2t) & \text{if } r \geq r^{\star\star} \end{cases} \tag{46}$$

To reach the supremum, we must have $r^{\star\star} = \xi_2^{\star\star}$. This shows that the second-order moment can also be reduced to a one-dimensional integral involving the radial densities $\tilde{p}_0^R(r)$ and $\tilde{p}_{t|0}^{R_2}(r)$.

**Step 2: Calculating 1-d integral.** In this step, we bound $\xi_1^{**}$ and $(\xi_2^{**})^2$ under the polar coordinate setting as outlined in Equations (42) and (46). Recall that $\xi_1^{**} \geq \mathbb{E}_{[0|t]}[\|y - x^*\| \mid x_t, y \notin B_\tau]$ and $(\xi_2^{**})^2 \geq \mathbb{E}_{[0|t]}[\|y - x^*\|^2 \mid x_t, y \notin B_\tau]$. Based on the results in Step 1, we can upper bound $\xi_1^{**}$ and $\xi_2^{**}$ as

$$\xi_1^{**} \int_{D_s}^{\infty} \tilde{p}_{t|0}^{R_1}(r)\tilde{p}_0^R(r)\, dr \leq \int_{D_s}^{\xi_1^{**}} r\tilde{p}_{t|0}^{R_1}(r)\tilde{p}_0^R(r)\, dr + \int_{\xi_1^{**}}^{\infty} r\tilde{p}_{t|0}^{R_1}(r)\tilde{p}_0^R(r)\, dr$$

$$(\xi_2^{**})^2 \int_{D_s}^{\infty} \tilde{p}_{t|0}^{R_2}(r)\tilde{p}_0^R(r)\, dr \leq \int_{D_s}^{\xi_2^{**}} r^2\tilde{p}_{t|0}^{R_2}(r)\tilde{p}_0^R(r)\, dr + \int_{\xi_2^{**}}^{\infty} r^2\tilde{p}_{t|0}^{R_2}(r)\tilde{p}_0^R(r)\, dr$$

**2.1: Piecewise characterization of $\tilde{p}_{0|t}^{R_1}(r)$ and $\tilde{p}_{0|t}^{R_2}(r)$.** Since the definitions of $\xi_1^{**}$ and $(\xi_2^{**})^2$ involve ratios of integrals, the integrands $\tilde{p}_{0|t}^{R_1}(r) \propto \tilde{p}_{t|0}^{R_1}(r)\tilde{p}_0^R(r)$ and $\tilde{p}_{0|t}^{R_2}(r) \propto \tilde{p}_{t|0}^{R_2}(r)\tilde{p}_0^R(r)$ only need to be determined up to a constant normalization factor. We therefore have Equations (47) and (48):

$$\xi_1^{**} \int_{D_s}^{\infty} \tilde{p}_{0|t}^{R_1}(r)\, dr \leq \int_{D_s}^{\xi_1^{**}} r\tilde{p}_{0|t}^{R_1}(r)\, dr + \int_{\xi_1^{**}}^{\infty} r\tilde{p}_{0|t}^{R_1}(r)\, dr \tag{47}$$

$$(\xi_2^{**})^2 \int_{D_s}^{\infty} \tilde{p}_{0|t}^{R_2}(r)\, dr \leq \int_{D_s}^{\xi_2^{**}} r^2\tilde{p}_{0|t}^{R_2}(r)\, dr + \int_{\xi_2^{**}}^{\infty} r^2\tilde{p}_{0|t}^{R_2}(r)\, dr \tag{48}$$

Recall that $\tilde{p}_0^R(r)$ is constructed such that its tail probability matches the sub-Gaussian bound $\exp(-r^2/\tau^2)$ exactly for $r > D_s$, and $\tilde{p}_{t|0}^{R_1}(r)$ and $\tilde{p}_{t|0}^{R_2}(r)$ are piecewise defined based on Gaussian functions (see Equation (39)). Therefore, their product can be analyzed by adapting standard results for products of Gaussian densities. As an example, for $R > \xi_1^{**}$, we consider $\tilde{p}^{R_1}$:

$$\tilde{P}_{0|t}^{R_1}(r > R) \propto \int_R^{\infty} r \exp(-\frac{r^2}{\tau^2}) \exp(-\frac{(r - \|x_t - x^*\|)^2}{2t})\, dr$$

$$\propto \int_R^{\infty} r \exp(-\frac{(r - r_t')^2}{\tau'^2})\, dr$$

$$\propto (1 - \Phi(\frac{R - r_t'}{\tau'}))r_t' + \tau'\phi(\frac{R - r_t'}{\tau'})$$

Here $\phi$ and $\Phi$ are the standard normal distribution's pdf and cdf, respectively, as detailed in Lemma B.9, and

$$r_t' = \frac{\tau^2}{\tau^2 + 2t} \|x_t - x^*\| \quad \text{and} \quad \tau'^2 = \frac{2\tau^2 t}{\tau^2 + 2t}.$$

Without loss of generality, we assume $x^* = 0$ for simplicity in the following analysis, in which case $r_t' = \frac{\tau^2}{\tau^2 + 2t} \|x_t\|$. Furthermore, the above CDF satisfies the following property: for any $R' \geq R$,

$$\frac{(1 - \Phi(\frac{R' - r_t'}{\tau'}))r_t' + \tau'\phi(\frac{R' - r_t'}{\tau'})}{(1 - \Phi(\frac{R - r_t'}{\tau'}))r_t' + \tau'\phi(\frac{R - r_t'}{\tau'})} \leq \frac{\exp(-\frac{(R' - r_t')^2}{\tau'^2})}{\exp(-\frac{(R - r_t')^2}{\tau'^2})}$$

This analysis reveals that the tail of the resulting effective density $\tilde{p}_{0|t}^{R_1}(r)$ for $R > \xi_1^{**}$ is bounded by piecewise sub-Gaussian densities. We use $\mathcal{SG}(r|\mu, \Sigma)$ to denote a density proportional to a sub-Gaussian with parameters $\mu$ and $\Sigma$. More precisely, the tail of $\tilde{p}_{0|t}^{R_1}(r)$ is bounded by the tail of $\mathcal{SG}(r|r_t', \tau'^2)$ for $R > \xi_1^{**}$. For $R < \xi_1^{**}$, we directly compare the density $\tilde{p}_{0|t}^{R_1}(r)$ at two different values $r$ and $r'$, with $r' > r$.

$$\frac{\tilde{p}_{0|t}^{R_1}(r')}{\tilde{p}_{0|t}^{R_1}(r)} \propto \frac{r' \exp(-\frac{(r' + r_t')^2}{\tau'^2})}{r \exp(-\frac{(r + r_t')^2}{\tau'^2})} \leq \frac{r' \exp(-\frac{(r')^2}{\tau'^2})}{r \exp(-\frac{(r)^2}{\tau'^2})} \tag{49}$$

**2.2: Bounding Expectation with a heavy-tailed density.** From the above tail bound, we can see that the tail of $\tilde{p}_{0|t}^{R_1}(r)$ is lighter-tailed than $\mathcal{SG}\left(r|0, \tau'^2\right)$ for $R < \xi_1^{**}$. One can show that if $p_1(x)/p_1(y) \leq p_2(x)/p_2(y)$ holds, then $\mathbb{E}_{p_1}(x) < \mathbb{E}_{p_2}(x)$ holds. Consequently, we can bound $\mathbb{E}_{\tilde{p}_{0|t}^{R_1}}[r]$ using the truncated moments of sub-Gaussian random variables derived in Lemma B.2.

Define the truncated normalizing constants

$$Z_1 := \int_{D_s}^{\xi_1^{**}} \mathcal{SG}\left(r|0, \tau'^2\right)(u)\, du, \qquad Z_2 := \int_{\xi_1^{**}}^{\infty} \mathcal{SG}\left(r|r_t', \tau'^2\right)(u)\, du.$$

Let $\tilde{P}_{0|t}^{R_1}$ denote the reference probability induced by $\tilde{p}_{0|t}^{R_1}$. We define the comparison density $\hat{p}$ on $[D_s, \infty)$ as

$$\hat{p}_1(r) := \tilde{P}_{0|t}^{R_1}(D_s \leq r \leq \xi_1^{**}) \frac{\mathcal{SG}\left(r|0, \tau'^2\right)(r)\,\mathbf{1}\{D_s \leq r \leq \xi_1^{**}\}}{Z_1} + \tilde{P}_{0|t}^{R_1 s}(r \geq \xi_1^{**}) \frac{\mathcal{SG}\left(r|r_t', \tau'^2\right)(r)\,\mathbf{1}\{r > \xi_1^{**}\}}{Z_2}. \tag{50}$$

Therefore, the expectation of $\tilde{p}_{0|t}^{R_1}$ can be bounded by the expectation of Equation (50) since Equation (49) is satisfied. Likewise, for $\tilde{p}_{0|t}^{R_2}$ we can bound the expectation using

$$\hat{p}_2(r) := \tilde{P}_{0|t}^{R_2}(D_s \leq r \leq \xi_2^{**}) \frac{\mathcal{SG}\left(r|0, \tau'^2\right)(r)\,\mathbf{1}\{D_s \leq r \leq \xi_2^{**}\}}{Z_1} + \tilde{P}_{0|t}^{R_2}(r \geq \xi_2^{**}) \frac{\mathcal{SG}\left(r|r_t', \tau'^2\right)(r)\,\mathbf{1}\{r > \xi_2^{**}\}}{Z_2}. \tag{51}$$

This piecewise characterization constructs a comparison density $\hat{p}$ on $[D_s, \infty)$ from sub-Gaussian densities and ensures that it is more heavy-tailed than $\tilde{p}_{0|t}^{R_1}(r)$ and $\tilde{p}_{0|t}^{R_2}(r)$. We can then compute the required moments using Lemma B.2.

**2.3: Calculating the truncated moments.** With the above characterization, we now apply Lemma B.2, which states that for a sub-Gaussian random variable $r$ with density $\mathcal{SG}\left(r|\mu, \tau^2\right)$, we have:

$$\mathbb{E}[(r-\mu)^2|r > a] \leq (a-\mu)^2 + \tau^2$$
$$\mathbb{E}[r-\mu|r > a] \leq a - \mu + \tau$$

Expanding the left-hand side and rearranging terms, we obtain

$$\mathbb{E}[r^2|r > a] \leq a^2 + \tau^2 + 2\mu\tau \tag{52}$$
$$\mathbb{E}[r|r > a] \leq a + \tau \tag{53}$$

Applying Equations (52) and (53) to Equations (47) and (48), we obtain

$$\xi_1^{**}\left(\int_{D_s}^{\infty} \tilde{p}_{0|t}^{R_1}(r)dr\right) \leq (\tau' + D_s)\int_{D_s}^{\xi_1^{**}} \tilde{p}_{0|t}^{R_1}(r)dr + (\tau' + \xi_1^{**})\int_{\xi_1^{**}}^{\infty} \tilde{p}_{0|t}^{R_1}(r)dr \tag{54}$$

$$(\xi_2^{**})^2\left(\int_{D_s}^{\infty} \tilde{p}_{0|t}^{R_2}(r)dr\right) \leq (D_s^2 + \tau'^2)\int_{D_s}^{\xi_2^{**}} \tilde{p}_{0|t}^{R_2}(r)dr + ((\xi_2^{**})^2 + \tau'^2 + 2\tau'r_t')\int_{\xi_2^{**}}^{\infty} \tilde{p}_{0|t}^{R_2}(r)dr \tag{55}$$

Dividing the first inequality by $\int_{D_s}^{\infty} \tilde{p}_{0|t}^{R_1}(r)dr$ and the second by $\int_{D_s}^{\infty} \tilde{p}_{0|t}^{R_2}(r)dr$, we obtain

$$(\xi_2^{**})^2 \leq (D_s^2 + \tau'^2) + (\tau'^2 + 2\tau'r_t')\frac{\int_{\xi_2^{**}}^{\infty} \tilde{p}_{0|t}^{R_2}(r)dr}{\int_{D_s}^{\xi_2^{**}} \tilde{p}_{0|t}^{R_2}(r)dr} \tag{56}$$

$$\xi_1^{**} \leq \tau' + D_s + \tau'\frac{\int_{\xi_1^{**}}^{\infty} \tilde{p}_{0|t}^{R_1}(r)dr}{\int_{D_s}^{\xi_1^{**}} \tilde{p}_{0|t}^{R_1}(r)dr} \tag{57}$$

From Equations (56) and (57), we see that the bounds on the truncated moments $\xi_1^{**}$ and $(\xi_2^{**})^2$ depend on the fractions involving integrals of the effective radial density $\tilde{p}_{0|t}^{R_1}(r)$ and $\tilde{p}_{0|t}^{R_2}(r)$:

$$F_1 = \frac{\int_{\xi_1^{**}}^{\infty} \tilde{p}_{0|t}^{R_1}(r)dr}{\int_{D_s}^{\xi_1^{**}} \tilde{p}_{0|t}^{R_1}(r)dr} \quad \text{and} \quad F_2 = \frac{\int_{\xi_2^{**}}^{\infty} \tilde{p}_{0|t}^{R_2}(r)dr}{\int_{D_s}^{\xi_2^{**}} \tilde{p}_{0|t}^{R_2}(r)dr}.$$

**2.4: Bounding the fractions $F_1$ and $F_2$.** We now bound these fractions by analyzing cases. We first focus on the second moment, $(\xi_2^{**})^2 = \mathbb{E}_{[0|t]}[\|y - x^*\|^2 \mid x_t, y \notin B_\tau]$. We consider two cases based on the relationship between $\xi_2^{**} - r_t'$ and $D_s + r_t'$.

**Case 1:** $\xi_2^{**} - r_t' \geq D_s + r_t' + \tau'$. In this case, we bound the fraction $F_2$. The analysis relies on bounding the numerator and denominator integrals, assuming the integrand $\tilde{p}_{0|t}^R(r)$ behaves similarly to $r \exp(-\frac{(r+r_t')^2}{\tau'^2})$ in the relevant ranges. The denominator integral is bounded below:

$$\int_{D_s}^{\xi_2^{**}} \tilde{p}_{0|t}^R(r)dr \geq C_{\text{lower}} \int_{D_s}^{\infty} r \exp(-\frac{(r+r_t')^2}{\tau'^2})dr = C_{\text{lower}}(\tau'\phi(\frac{D_s + r_t'}{\tau'}) - r_t'(1 - \Phi(\frac{D_s + r_t'}{\tau'}))),$$

The existence of a positive constant factor $C_{\text{lower}} \geq (1 - 1/e)$ is guaranteed because the integration interval $[D_s, \xi_2^{**}]$ contains at least $[D_s, D_s + \tau']$ (due to the properties of sub-Gaussian tails). This ensures that a non-vanishing fraction of the integral from $D_s$ to infinity is captured. The numerator integral is calculated using results similar to those in Lemma B.9:

$$\int_{\xi_2^{**}}^{\infty} \tilde{p}_{0|t}^R(r)dr \propto \tau'\phi(\frac{\xi_2^{**} - r_t'}{\tau'}) + r_t'(1 - \Phi(\frac{\xi_2^{**} - r_t'}{\tau'})).$$

Considering the case $r_t' \leq \tau'$, we can then bound $F_2$ as follows:

$$\begin{aligned}
F_2 &\leq \frac{\tau'\phi(\frac{\xi_2^{**}-r_t'}{\tau'}) + \tau'(1 - \Phi(\frac{\xi_2^{**}-r_t'}{\tau'}))}{C_{\text{lower}}(\tau'\phi(\frac{D_s+r_t'}{\tau'}) - \tau'(1 - \Phi(\frac{D_s+r_t'}{\tau'})))} \\
&= \frac{\phi(\frac{\xi_2^{**}-r_t'}{\tau'}) + (1 - \Phi(\frac{\xi_2^{**}-r_t'}{\tau'}))}{C_{\text{lower}}(\phi(\frac{D_s+r_t'}{\tau'}) - (1 - \Phi(\frac{D_s+r_t'}{\tau'})))} \\
&\leq \frac{1/e}{1 - 1/e} \frac{\phi(\frac{D_s+r_t'}{\tau'}) + (1 - \Phi(\frac{D_s+r_t'}{\tau'}))}{(\phi(\frac{D_s+r_t'}{\tau'}) - (1 - \Phi(\frac{D_s+r_t'}{\tau'})))}
\end{aligned} \tag{58}$$

This inequality is derived from the condition $\xi_2^{**} - r_t' \geq D_s + r_t' + \tau'$, which implies $\frac{\xi_2^{**}-r_t'}{\tau'} \geq \frac{D_s+r_t'}{\tau'} + 1$. We leverage the decay properties of the Gaussian tail function $\phi(z)$ for arguments separated by at least 1, along with the established lower bound $C_{\text{lower}} \geq (1 - 1/e)$. One can verify that $\frac{\phi(z)+\Phi_C(z)}{\phi(z)-\Phi_C(z)}$ is monotonically decreasing when $z > 1$. Through numerical calculation, we obtain

$$F_2 = \frac{\int_{\xi_2^{**}}^{\infty} \tilde{p}_{0|t}^R(r)dr}{\int_{D_s}^{\xi_2^{**}} \tilde{p}_{0|t}^R(r)dr} \leq \frac{3}{2}.$$

Substituting this bound into Equation (56), we get:

$$(\xi_2^{**})^2 \leq (D_s^2 + \tau'^2) + (\tau'^2 + 2\tau' r_t')\frac{3}{2} = D_s^2 + \frac{5}{2}\tau'^2 + 3\tau' r_t'.$$

**Case 2:** $\xi_2^{**} - r_t' < D_s + r_t' + \tau'$. In this case, we have $\xi_2^{**} \leq D_s + 2r_t' + \tau'$. This yields a direct bound on the squared moment:

$$(\xi_2^{**})^2 \leq (D_s + 2r_t' + \tau')^2.$$

We now perform a similar case analysis for the first moment, $\xi_1^{**} = \mathbb{E}_{[0|t]}[\|y - x^*\| \mid x_t, y \notin B_\tau]$, using Equation (57).

**Case 1:** $\xi_1^{**} - r_t' < D_s + r_t' + \tau'$. In this case, we have $\mathbb{E}_{[0|t]}[\|y - x^*\| \mid x_t, y \notin B_\tau] \leq D_s + 2r_t' + \tau'$.

**Case 2:** $\xi_1^{**} - r_t' \geq D_s + r_t' + \tau'$. In this case, we have $\mathbb{E}_{[0|t]}[\|y - x^*\| \mid x_t, y \notin B_\tau] \leq D_s + 3\tau'$ as $F_1$ can be caculated in a similar way as in Equation (58).

**Step 3: Combined Bound for Moments.** Combining the results from both cases for the second moment yields:

$$\mathbb{E}_{[0|t]}[\|y - x^*\|^2 \mid x_t, y \notin B_\tau] = (\xi_2^{**})^2 \leq \max\left((D_s + 2r_t' + \tau')^2, \; D_s^2 + \tfrac{5}{2}\tau'^2 + 3\tau' r_t'\right). \tag{59}$$

Combining the results from both cases for the first moment gives:

$$\mathbb{E}_{[0|t]}[\|y - x^*\| \mid x_t, y \notin B_\tau] = \xi_1^{**} \leq \max\left(D_s + 2r_t' + \tau', \; D_s + 3\tau'\right). \tag{60}$$

This concludes the derivation of the moment bounds stated in Lemma B.4.

$\square$

Based on the above lemmas, we are ready to prove Theorem 4.3 (restated below).

**Theorem 4.3** (Strongly Convex Bound). *Under Assumption 4.1 and Assumption 4.2, let $d$ denote the dimension of the space and define $\kappa_0 := \beta/\alpha$. Fix any $C_\alpha \in (0,1)$ and assume $\lambda \leq \lambda_{\max}$ for some fixed $\lambda_{\max} > 0$. Then, for any $t > 0$, the function $g(x;t)$ is $\frac{C_\alpha \lambda}{t + \frac{\lambda}{\alpha}}$-strongly convex within the region*

$$\mathcal{R}_{SC}(t) := \left\{ x \in \mathbb{R}^d \;\middle|\; \|x - x^*\| \leq C_E \min\left(\sqrt{\frac{t + \frac{\lambda}{\beta}}{\frac{\lambda}{\beta}}}, \; \sqrt{\frac{t + \tau^2}{\tau^2}}\right) D_\tau \right\} \tag{7}$$

*i.e., for all $x \in \mathcal{R}_{SC}(t)$ we have $\nabla^2 g(x;t) \succeq \frac{C_\alpha \lambda}{t + \frac{\lambda}{\alpha}} I$, provided the sufficient parameter conditions in Equation (20) hold. Here $D_\tau$, $\tau$, and $P_{out}$ are the sub-Gaussian/tail parameters from Assumption 4.1, and $C_E$ is the expansion-rate factor defining $\mathcal{R}_{SC}(t)$.*

*Proof.* Recall the Tweedie-form Hessian identity (see Equation (19)):

$$\nabla_x^2 g(x;t) = \frac{\lambda}{t^2}\left[t\,\mathbf{I} - \mathrm{Cov}_{[0|t]}[y \mid x_t]\right], \qquad \mathrm{Cov}_{[0|t]}[y \mid x_t] := \mathrm{Cov}_{[0|t]}[y \mid x_t = x].$$

Hence, to prove $\frac{C_\alpha \lambda}{t + \lambda/\alpha}$-strong convexity of $g(\cdot; t)$ at $x$, it suffices to show

$$t\,\mathbf{I} - \mathrm{Cov}_{[0|t]}[y \mid x_t] \succeq \frac{C_\alpha t^2}{t + \lambda/\alpha}\,\mathbf{I}. \tag{61}$$

**Step 1: Variance decomposition for the conditional distribution.** Consider the conditional distribution $Y \sim p_{0|t}(\cdot \mid x_t)$ and the event $\{Y \in B_\tau\}$. Applying the law of total variance (see Lemma B.8) to this conditional distribution, we have

$$\mathrm{Cov}_{[0|t]}[y \mid x_t] = P_{\mathrm{in}[0|t]}\,\mathrm{Cov}_{[0|t]}[y \mid x_t, y \in B_\tau] + P_{\mathrm{out}[0|t]}\,\mathrm{Cov}_{[0|t]}[y \mid x_t, y \notin B_\tau]$$
$$+ P_{\mathrm{in}[0|t]}P_{\mathrm{out}[0|t]}\big(\mu_{\mathrm{out}} - \mu_{\mathrm{in}}\big)\big(\mu_{\mathrm{out}} - \mu_{\mathrm{in}}\big)^\top,$$

where

$$P_{\mathrm{in}[0|t]} := P_{0|t}(y \in B_\tau \mid x_t), \quad P_{\mathrm{out}[0|t]} := P_{0|t}(y \notin B_\tau \mid x_t),$$

and $\mu_{\mathrm{in}} := \mathbb{E}_{[0|t]}[y \mid x_t, y \in B_\tau]$, $\mu_{\mathrm{out}} := \mathbb{E}_{[0|t]}[y \mid x_t, y \notin B_\tau]$ are the conditional means.

**Step 2: Upper bounding the contribution from the non-convex region.** Let $\Sigma_{\mathrm{in}} := \mathrm{Cov}_{[0|t]}[y \mid x_t, y \in B_\tau]$ be the covariance within the convex region. We now bound the remaining terms in the decomposition.

First, note that for any vector $z$, $zz^\top \preceq \|z\|^2 \mathbf{I}$. Combined with the property that the covariance matrix of any random vector $Z$ satisfies $\mathrm{Cov}(Z) = \mathbb{E}[ZZ^\top] - \mathbb{E}[Z]\mathbb{E}[Z]^\top \preceq \mathbb{E}[Z^\top Z]\mathbf{I}$, we obtain

$$\mathrm{Cov}_{[0|t]}[y \mid x_t, y \notin B_\tau] = \mathrm{Cov}_{[0|t]}[y - x^* \mid x_t, y \notin B_\tau]$$
$$\preceq \mathbb{E}_{[0|t]}\big[\|y - x^*\|^2 \mid x_t, y \notin B_\tau\big]\mathbf{I} = \mathbb{E}_{[0|t]}[\|y - x^*\|^2 \mid x_t, y \notin B_\tau]\mathbf{I}.$$

Second, since $y \in B_\tau = \{y : \|y - x^*\| < D_\tau\}$ implies $\|\mathbb{E}[y - x^* \mid x_t, y \in B_\tau]\| \leq D_\tau$, the distance between the conditional means satisfies:

$$\|\mu_{\mathrm{out}} - \mu_{\mathrm{in}}\| = \big\|\mathbb{E}_{[0|t]}[y - x^* \mid x_t, y \notin B_\tau] - \mathbb{E}_{[0|t]}[y - x^* \mid x_t, y \in B_\tau]\big\| \leq \mathbb{E}_{[0|t]}[\|y - x^*\| \mid x_t, y \notin B_\tau] + D_\tau.$$

Consequently, the cross term in the variance decomposition is bounded by

$$P_{\mathrm{in}[0|t]}P_{\mathrm{out}[0|t]}(\mu_{\mathrm{out}} - \mu_{\mathrm{in}})(\mu_{\mathrm{out}} - \mu_{\mathrm{in}})^\top$$
$$\preceq P_{\mathrm{out}[0|t]}(\mathbb{E}_{[0|t]}[\|y - x^*\| \mid x_t, y \notin B_\tau] + D_\tau)^2\mathbf{I} \preceq P_{\mathrm{out}[0|t]}(\mathbb{E}_{[0|t]}[\|y - x^*\|^2 \mid x_t, y \notin B_\tau] + 2D_\tau\mathbb{E}_{[0|t]}[\|y - x^*\| \mid x_t, y \notin B_\tau] + D_\tau^2)\mathbf{I}.$$

where the second inequality uses Jensen's inequality $(\mathbb{E}_{[0|t]}[\|y - x^*\| \mid x_t, y \notin B_\tau])^2 \leq \mathbb{E}_{[0|t]}[\|y - x^*\|^2 \mid x_t, y \notin B_\tau]$. Combining these bounds, we arrive at

$$\mathrm{Cov}_{[0|t]}[y \mid x_t] \preceq P_{\mathrm{in}[0|t]}\Sigma_{\mathrm{in}} + P_{\mathrm{out}[0|t]}M_{\mathrm{out}}\mathbf{I},$$
$$M_{\mathrm{out}} := 2\mathbb{E}_{[0|t]}[\|y - x^*\|^2 \mid x_t, y \notin B_\tau] + 2D_\tau\mathbb{E}_{[0|t]}[\|y - x^*\| \mid x_t, y \notin B_\tau] + D_\tau^2. \tag{62}$$

**Step 3: Reduction of strong convexity to a bound on tail moments.** Substituting the bound from Equation (62) into the strong convexity requirement Equation (61), it is sufficient to show that

$$t\mathbf{I} - P_{\mathrm{in}[0|t]}\Sigma_{\mathrm{in}} - P_{\mathrm{out}[0|t]}M_{\mathrm{out}}\mathbf{I} \succeq \frac{C_\alpha t^2}{t + \lambda/\alpha}\mathbf{I}.$$

Applying Lemma B.3, which states $\Sigma_{\mathrm{in}} \preceq \frac{t(\lambda/\alpha)}{t+\lambda/\alpha}\mathbf{I}$, and noting that $P_{\mathrm{in}[0|t]} \leq 1$, we find it sufficient to require

$$\left(t - \frac{t(\lambda/\alpha)}{t + \lambda/\alpha}\right)\mathbf{I} - P_{\mathrm{out}[0|t]}M_{\mathrm{out}}\mathbf{I} \succeq \frac{C_\alpha t^2}{t + \lambda/\alpha}\mathbf{I}.$$

Since $t - \frac{t(\lambda/\alpha)}{t+\lambda/\alpha} = \frac{t^2}{t+\lambda/\alpha}$, this simplifies to the following sufficient condition:

$$P_{\mathrm{out}[0|t]}M_{\mathrm{out}} \leq \frac{(1 - C_\alpha)t^2}{t + \lambda/\alpha}. \tag{63}$$

Thus, proving $\frac{C_\alpha\lambda}{t+\lambda/\alpha}$-strong convexity reduces to bounding (i) the conditional tail mass $P_{\mathrm{out}[0|t]}$ and (ii) the outside-moment term $M_{\mathrm{out}}$ so that Equation (63) holds.

**Step 4: Upper bound on $M_{\mathrm{out}}$.** We introduce unified "effective mean/variance" notation that mirrors the product-of-Gaussians parameters used in Step 5. Define

$$\mu_{\mathrm{in}}' := \frac{\frac{\lambda}{\beta}}{\frac{\lambda}{\beta} + 2t}x_t + \frac{t}{\frac{\lambda}{\beta} + 2t}x^*, \qquad (\sigma_{\mathrm{in}}')^2 := \frac{t\frac{\lambda}{\beta}}{\frac{\lambda}{\beta} + 2t}, \tag{64}$$

and

$$\mu_{\mathrm{out}}' := \frac{\tau^2}{\tau^2 + 2t}x_t + \frac{t}{\tau^2 + 2t}x^*, \qquad (\sigma_{\mathrm{out}}')^2 := \frac{t\tau^2}{t + \tau^2}. \tag{65}$$

Note that $(\sigma_{\mathrm{out}}')^2 = \tau'^2$, where

$$\tau'^2 := \frac{t\tau^2}{2t + \tau^2}.$$

Define the scalar shrinkage radius

$$r'_t := \|\mu'_{\text{out}} - x^*\| = \frac{\tau^2}{2t + \tau^2}\|x_t - x^*\|.$$

Also as in Table 3, we have that

$$D_s := \tau\sqrt{\log\left(\frac{1}{P_{\text{out}}}\right)}.$$

Lemma B.4 gives

$$\mathbb{E}_{[0|t]}[\|y - x^*\|^2 \mid x_t, y \notin B_\tau] \leq \max\left((D_s + 2r'_t)^2, \ D_s^2 + 3\tau'^2 + 4\tau'r'_t\right),$$
$$\mathbb{E}_{[0|t]}[\|y - x^*\| \mid x_t, y \notin B_\tau] \leq \max(D_s + 2r'_t, \ D_s + 3\tau').$$

To simplify the algebra, we upper bound the maxima by slightly looser expressions. Since $D_s \geq \tau'$ and $r'_t \geq 0$, we have $4\tau'r'_t \leq 4D_s r'_t \leq 4D_s r'_t + 4(r'_t)^2$, hence

$$D_s^2 + 3\tau'^2 + 4\tau'r'_t \leq (D_s + 2r'_t)^2 + 3\tau'^2,$$

which implies

$$\mathbb{E}_{[0|t]}[\|y - x^*\|^2 \mid x_t, y \notin B_\tau] \leq (D_s + 2r'_t)^2 + 3\tau'^2,$$
$$\mathbb{E}_{[0|t]}[\|y - x^*\| \mid x_t, y \notin B_\tau] \leq D_s + 2r'_t + 3\tau'.$$

Substituting these into $M_{\text{out}} = 2\mathbb{E}_{[0|t]}[\|y - x^*\|^2 \mid x_t, y \notin B_\tau] + 2D_\tau\mathbb{E}_{[0|t]}[\|y - x^*\| \mid x_t, y \notin B_\tau] + D_\tau^2$, we obtain

$$M_{\text{out}} \leq 2\left[(D_s + 2r'_t)^2 + 3\tau'^2\right] + 2D_\tau\left[D_s + 2r'_t + 3\tau'\right] + D_\tau^2$$
$$= 2D_s^2 + 8D_s r'_t + 8(r'_t)^2 + 6\tau'^2 + 2D_\tau D_s + 4D_\tau r'_t + 6D_\tau\tau' + D_\tau^2. \tag{66}$$

In particular, $M_{\text{out}}$ is controlled by $(D_\tau, \tau, P_{\text{out}})$ and $\|x_t - x^*\|$ through $r'_t$.

**Simplifications under the two distance regimes.**

**Case 1** ($\|x_t - x^*\| \leq \frac{1}{3}D_\tau$). Using $r'_t \leq \|x_t - x^*\| \leq \frac{1}{3}D_\tau$ and $\tau' \leq \tau$, the bound Equation (66) implies

$$M_{\text{out}} \leq 2D_s^2 + 8D_s r'_t + 8(r'_t)^2 + 6\tau'^2 + 2D_\tau D_s + 4D_\tau r'_t + 6D_\tau\tau' + D_\tau^2$$
$$\leq 2D_s^2 + 5D_s D_\tau + 5D_\tau^2 \leq 12\max(D_\tau^2, D_s^2), \tag{67}$$

for all sufficiently large $d$ (since $D_\tau^2 = \Theta(d)$, equivalently $D_\tau = \Theta(\sqrt{d})$, while $\tau = \Theta(1)$).

**Case 2** ($\|x_t - x^*\| > \frac{1}{3}D_\tau$). Recall Equation (66). Under the expansion-rate bounds used later (so that $r'_t \leq C_E D_\tau$ and $\tau' \leq \tau$), we have

$$M_{\text{out}} \leq 2D_s^2 + 8C_E D_s D_\tau + 8C_E^2 D_\tau^2 + 6\tau^2 + 2D_s D_\tau + 4C_E D_\tau^2 + 6D_\tau\tau + D_\tau^2$$
$$\leq 2D_s^2 + 3D_s D_\tau + 4D_\tau^2 \leq 9\max(D_\tau^2, D_s^2), \tag{68}$$

for all sufficiently large $d$ (since $C_E \sim \Theta(\sqrt{\log d/d})$, $D_\tau^2 \sim \Theta(d)$, and $\tau = \Theta(1)$).

The next step is to bound $P_{\text{out}[0|t]}$ and verify that $P_{\text{out}[0|t]}M_{\text{out}}$ satisfies Equation (63).

**Step 5: Upper bound on $P_{\text{out}[0|t]}$.** Since $p_{0|t}(y \mid x_t) \propto p_0(y)\,p_{t|0}(x_t \mid y)$, define the unnormalized masses

$$Z_{\text{in}}(x_t) := \int_{y \in B_\tau} p_0(y)\,p_{t|0}(x_t \mid y)\,dy, \qquad Z_{\text{out}}(x_t) := \int_{y \notin B_\tau} p_0(y)\,p_{t|0}(x_t \mid y)\,dy.$$

Then $P_{\text{out}[0|t]}/P_{\text{in}[0|t]} = Z_{\text{out}}(x_t)/Z_{\text{in}}(x_t)$. We next upper bound $Z_{\text{out}}$ and lower bound $Z_{\text{in}}$ to obtain bounds on $P_{\text{out}[0|t]}$ and $P_{\text{in}[0|t]}$.

**Upper bound on $Z_{\text{out}}(x_t)$.** By definition of $\min_{y \notin B_\tau} \|y - x_t\| := \min_{y \notin B_\tau} \|y - x_t\|$, for all $y \notin B_\tau$,

$$p_{t|0}(x_t \mid y) = (2\pi t)^{-d/2} \exp\left(-\frac{\|x_t - y\|^2}{2t}\right) \le (2\pi t)^{-d/2} \exp\left(-\frac{\min_{y \notin B_\tau} \|y - x_t\|^2}{2t}\right).$$

Therefore, using $\int_{y \notin B_\tau} p_0(y)\, dy = P_{\text{out}}$,

$$Z_{\text{out}}(x_t) \le P_{\text{out}} \, (2\pi t)^{-d/2} \exp\left(-\frac{\min_{y \notin B_\tau} \|y - x_t\|^2}{2t}\right).$$

**Lower bound on $Z_{\text{in}}(x_t)$.** Since $f$ is $\alpha$-strongly convex and $\beta$-smooth on $B_\tau$, and $\nabla f(x^*) = 0$, for all $y \in B_\tau$,

$$\frac{\alpha}{2}\|y - x^*\|^2 \le f(y) - f(x^*) \le \frac{\beta}{2}\|y - x^*\|^2.$$

Recalling $p_0(y) = p_0(x^*) \exp(-(f(y) - f(x^*))/\lambda)$, this implies for all $y \in B_\tau$,

$$p_0(x^*) \exp\left(-\frac{\beta}{2\lambda}\|y - x^*\|^2\right) \le p_0(y) \le p_0(x^*) \exp\left(-\frac{\alpha}{2\lambda}\|y - x^*\|^2\right).$$

Using $P_{\text{in}} = \int_{y \in B_\tau} p_0(y)\, dy$ and $\int_{\mathbb{R}^d} \exp(-\frac{\alpha}{2\lambda}\|u\|^2)\, du = (2\pi\lambda/\alpha)^{d/2}$, we obtain

$$p_0(x^*) \ge P_{\text{in}} \left(\frac{\alpha}{2\pi\lambda}\right)^{d/2}.$$

For $y \in B_\tau$, we have $p_0(y) \ge p_0(x^*) \exp\left(-\frac{\beta}{2\lambda}\|y - x^*\|^2\right)$, and $p_{t|0}(x_t \mid y) = k(x_t - y; t)$, the density of $P_{t|0}(\cdot \mid y) = \mathcal{N}(y, tI_d)$ evaluated at $x_t$. By the product-of-Gaussians identity (with $\mu'_{\text{in}}$ and $(\sigma'_{\text{in}})^2$ as defined in Equation (64)), we have

$$\mathcal{N}(y \mid x_t, tI) \exp\left(-\frac{\beta}{2\lambda}\|y - x^*\|^2\right) = \exp\left(-\frac{\beta}{2(Lt + \lambda)}\|x_t - x^*\|^2\right) \left(\frac{Lt + \lambda}{\lambda}\right)^{-d/2} \mathcal{N}(y \mid \mu'_{\text{in}}, (\sigma'_{\text{in}})^2 I).$$

Integrating the above bound over $y \in B_\tau$, we obtain

$$Z_{\text{in}}(x_t) = \int_{y \in B_\tau} p_0(y) \, p_{t|0}(x_t \mid y)\, dy$$

$$\ge p_0(x^*) \exp\left(-\frac{\beta}{2(\beta t + \lambda)}\|x^* - x_t\|^2\right) \left(\frac{\beta t + \lambda}{\lambda}\right)^{-d/2}$$

$$\cdot \int_{y \in B_\tau} \mathcal{N}\left(y \mid \mu'_{\text{in}}, (\sigma'_{\text{in}})^2 I\right) dy.$$

Combining the previous display with $p_0(x^*) \ge P_{\text{in}} \left(\frac{\alpha}{2\pi\lambda}\right)^{d/2}$, we obtain

$$Z_{\text{in}}(x_t) \ge P_{\text{in}} \left(\frac{\alpha}{2\pi}\right)^{d/2} (\beta t + \lambda)^{-d/2} \exp\left(-\frac{\beta}{2(\beta t + \lambda)}\|x^* - x_t\|^2\right) \int_{y \in B_\tau} \mathcal{N}\left(y \mid \mu'_{\text{in}}, (\sigma'_{\text{in}})^2 I\right) dy.$$

Therefore,

$$\frac{P_{\text{out}[0|t]}}{P_{\text{in}[0|t]}} = \frac{\int_{y \notin B_\tau} p_0(y) p_{t|0}(x_t \mid y)\, dy}{\int_{y \in B_\tau} p_0(y) p_{t|0}(x_t \mid y)\, dy}$$

$$\le \frac{P_{\text{out}} \exp\left(-\frac{1}{2t} \min_{y \notin B_\tau} \|y - x_t\|^2\right)}{P_{\text{in}} \exp\left(-\frac{\beta}{2(\beta t + \lambda)}\|x^* - x_t\|^2\right) \left(\frac{t\alpha}{\beta t + \lambda}\right)^{d/2} \int_{y \in B_\tau} \mathcal{N}\left(y \mid \mu'_{\text{in}}, (\sigma'_{\text{in}})^2 I\right) dy.} \quad (69)$$

**Lower bound on** $\int_{y \in B_\tau} \mathcal{N}\left(y \mid \mu'_{\text{in}}, (\sigma'_{\text{in}})^2 I\right) dy$**.** We now lower bound the Gaussian mass term $\int_{y \in B_\tau} \mathcal{N}\left(y \mid \mu'_{\text{in}}, (\sigma'_{\text{in}})^2 I\right) dy$ appearing in Equation (69). As stated in the theorem, $x_t \in \mathcal{R}_{SC}(t)$, i.e.

$$\|x^* - x_t\| \leq C_E \min\left(\sqrt{\frac{\frac{\lambda}{\beta} + t}{\frac{\lambda}{\beta}}}, \sqrt{\frac{\tau^2 + t}{\tau^2}}\right) D_\tau \tag{70}$$

This also implies the corresponding bound on the effective mean $\mu'_{\text{in}}$:

$$\|\mu'_{\text{in}} - x^*\| = \frac{\frac{\lambda}{\beta}}{t + \frac{\lambda}{\beta}} \|x_t - x^*\| \leq \|x_t - x^*\| \leq C_E D_\tau.$$

Moreover,

$$\|\mu'_{\text{in}} - x^*\| = \|x_t - x^*\| \frac{\frac{\lambda}{\beta}}{t + \frac{\lambda}{\beta}} \leq C_E \sqrt{\frac{\frac{\lambda}{\beta}}{\frac{\lambda}{\beta} + t}} D_\tau \leq C_E D_\tau, \tag{71}$$

Most importantly, from our assumption, we have

$$\frac{\beta}{Lt + \lambda} \|x_t - x^*\|^2 \leq C_E^2 \frac{\beta}{\lambda} D_\tau^2 \tag{72}$$

Note that

$$\int_{B_\tau} \mathcal{N}\left(y \mid \mu'_{\text{in}}, (\sigma'_{\text{in}})^2 I\right) dy = P(\|Z + \mu'_{\text{in}} - x^*\| \leq D_\tau) \geq P(\|Z\| \leq D_\tau - \|\mu'_{\text{in}} - x^*\|),$$

where $Z \sim \mathcal{N}(0, (\sigma'_{\text{in}})^2 I)$. In particular, if we choose $C_E \leq \frac{1}{3}$, then Equation (71) implies $\|\mu'_{\text{in}} - x^*\| \leq \frac{1}{3} D_\tau$, and hence $D_\tau - \|\mu'_{\text{in}} - x^*\| \geq \frac{2}{3} D_\tau$. Therefore,

$$\int_{B_\tau} \mathcal{N}\left(y \mid \mu'_{\text{in}}, (\sigma'_{\text{in}})^2 I\right) dy \geq P_{Z \sim \mathcal{N}(0, (\sigma'_{\text{in}})^2 I)}\left(\|Z\| \leq \frac{2}{3} D_\tau\right)$$

$$\geq P_{Z \sim \mathcal{N}(0, I)}\left(\|Z\|^2 \leq \frac{4 D_\tau^2}{9 (\sigma'_{\text{in}})^2}\right) \geq P_{Z \sim \mathcal{N}(0, I)}\left(\|Z\|^2 \leq \frac{4 D_\tau^2}{9} \frac{\beta}{\lambda}\right)$$

$$= \frac{\gamma\left(\frac{d}{2}, \frac{2 D_\tau^2}{9} \frac{\beta}{\lambda}\right)}{\Gamma\left(\frac{d}{2}\right)}. \tag{73}$$

We assume that $\lambda$ is upper bounded by a fixed constant $\lambda_{\max} > 0$ (a practical modeling choice since $\lambda \to \infty$ makes $p_0$ nearly uniform and removes learning signal). Then the incomplete-gamma argument satisfies

$$\frac{2 D_\tau^2}{9} \frac{\beta}{\lambda} \geq \frac{2 D_\tau^2}{9} \frac{\beta}{\lambda_{\max}}.$$

We will require:

$$\frac{2 D_\tau^2}{9} \frac{\beta}{\lambda_{\max}} \sim \Theta(d) \geq \frac{d}{2} \tag{74}$$

Under Equation (74), the lower-tail probability $\frac{\gamma\left(\frac{d}{2}, \frac{2 D_\tau^2}{9} \frac{\beta}{\lambda}\right)}{\Gamma\left(\frac{d}{2}\right)}$ is bounded below by a universal constant (e.g., $\geq \frac{1}{2}$ for all sufficiently large $d$) by standard Chernoff bounds for $\Gamma(\frac{d}{2}, 1)$.

Plugging Equation (73) into Equation (69) and using $P_{\text{in}[0|t]} \leq 1$, and as given in the assumption that $P_{\text{out}} \leq P_{\text{in}}$, we have

$$P_{\text{out}[0|t]} \leq 2 P_{\text{out}} \frac{\Gamma\left(\frac{d}{2}\right)}{\gamma\left(\frac{d}{2}, \frac{2 D_\tau^2 \beta}{9 \lambda}\right)} \frac{\exp\left(-\frac{1}{2t} \min_{y \notin B_\tau} \|y - x_t\|^2\right)}{\exp\left(-\frac{\beta}{2(\beta t + \lambda)} \|x^* - x_t\|^2\right)} \left(\frac{Lt + \lambda}{\alpha t}\right)^{d/2}$$

$$\leq 4 P_{\text{out}} \frac{\exp\left(-\frac{1}{2t} \min_{y \notin B_\tau} \|y - x_t\|^2\right)}{\exp\left(-\frac{1}{2} C_E^2 \frac{\beta}{\lambda} D_\tau^2\right)} \left(\frac{Lt + \lambda}{\alpha t}\right)^{d/2} \tag{75}$$

where the second inequality comes from Equation (72).

*A second (geometric) bound.* Define the inside-distance

$$\max_{y \in B_\tau} \|y - x_t\| := \min_{y \in B_\tau} \|y - x_t\|.$$

Then, by lower bounding the inside mass with $\exp(-\max_{y \in B_\tau} \|y - x_t\|^2/(2t))$ and upper bounding the outside mass with $\exp(-\min_{y \notin B_\tau} \|y - x_t\|^2/(2t))$, we also have

$$\frac{P_{\text{out}[0|t]}}{P_{\text{in}[0|t]}} \leq \frac{P_{\text{out}} \exp\left(-\frac{1}{2t} \min_{y \notin B_\tau} \|y - x_t\|^2\right)}{P_{\text{in}} \exp\left(-\frac{1}{2t} \max_{y \in B_\tau} \|y - x_t\|^2\right)}. \tag{76}$$

**Case analysis for $P_{\text{out}[0|t]}$.**

**Case 1:** $\|x_t - x^*\| \leq \frac{1}{3} D_\tau$.   In this case, we have two bounds. First, using Equation (75), we obtain

$$P_{\text{out}[0|t]} \leq 4 P_{\text{out}} \frac{\exp\left(-\frac{2D_\tau^2}{9t}\right)}{\exp\left(-\frac{1}{2} C_E^2 \frac{\beta}{\lambda} D_\tau^2\right)} \left(\frac{Lt + \lambda}{\alpha t}\right)^{d/2} \tag{77}$$

Second, using Equation (76), we obtain

$$P_{\text{out}[0|t]} \leq \frac{P_{\text{out}} \exp\left(-\frac{1}{2t} \min_{y \notin B_\tau} \|y - x_t\|^2\right)}{P_{\text{in}} \exp\left(-\frac{1}{2t} \max_{y \in B_\tau} \|y - x_t\|^2\right)}$$

$$\leq 2 \frac{P_{\text{out}} \exp\left(-\frac{2D_\tau^2}{9t}\right)}{P_0(\|x - x^*\| \leq \frac{1}{3} D_\tau) \exp\left(-\frac{2D_\tau^2}{9}\right)} \tag{78}$$

where we have used $\min_{y \notin B_\tau} \|y - x_t\|^2 \geq \frac{4}{9} D_\tau^2$. Since $D_\tau \sim \Theta(d)$ while $\alpha$ and $\beta$ are constants, $P(\|x - x^*\| \leq \frac{1}{3} D_\tau) \geq 1/2 P_{\text{in}}$ holds for all sufficiently large $d$. Therefore,

$$P_{\text{out}[0|t]} \leq 4 P_{\text{out}} \tag{79}$$

**Case 2:** $\|x_t - x^*\| > \frac{1}{3} D_\tau$.   In this case, we upper bound the probability as follows:

$$P_{\text{out}[0|t]} \leq \frac{P_{\text{out}}}{P_{\text{in}} \exp(-\frac{1}{2t}(D_\tau + \|x_t - x^*\|)^2)} \tag{80}$$

$$\leq \frac{2 P_{\text{out}}}{\exp(-\frac{8}{t}(\|x_t - x^*\|)^2)} \tag{81}$$

$$\leq 2 P_{\text{out}} \exp(8 \frac{\frac{\lambda}{\beta} + t}{\frac{\lambda}{\beta} t} C_E^2 D_\tau^2) \tag{82}$$

$$\leq 2 P_{\text{out}} \exp(8 \frac{\lambda}{\beta} C_E^2 D_\tau^2), \tag{83}$$

where the second inequality uses $\|x_t - x^*\| \geq \frac{1}{3} D_\tau$ (so $D_\tau + \|x_t - x^*\| \leq 4 \|x_t - x^*\|$), and the third inequality uses the convex shrinkage/region condition Equation (70).

**Step 6: Putting the bounds together.** Next, we plug in the upper bounds for $M_{\text{out}}$ from Equations (67) and (68) and for $P_{\text{out}[0|t]}$ from Equations (77), (79) and (83) under different conditions into the sufficient condition Equation (63) to achieve the strong convexity condition. More specifically, we organize the discussion with the following hierarchy:

- **Primary split (distance):** Case 1/2 by whether $\|x_t - x^*\| \leq \frac{1}{3} D_\tau$ or $\|x_t - x^*\| \geq \frac{1}{3} D_\tau$.

- **Secondary split (tail scale):** Case 1.1/1.2/2.1/2.2 by whether $D_s \geq D_\tau$ or $D_s < D_\tau$.

- **Tertiary split (time, only when needed):** further split by regimes of $t$ to simplify the sufficient conditions.

**Case 1:** $\|x_t - x^*\| \leq \frac{1}{3}D_\tau$. In this regime we use the simplified bound Equation (67).

**Case 1.1:** $D_s \geq D_\tau$. The following inequality is sufficient for Equation (63) to hold:

$$12D_s^2 \min(1, 4P_{\text{out}} \frac{\exp\left(-\frac{2D_\tau^2}{9t}\right)}{\exp\left(-\frac{1}{2}C_E^2 \frac{\beta}{\lambda}D_\tau^2\right)} (\frac{Lt + \lambda}{\alpha t})^{d/2}) \leq \frac{(1 - C_\alpha)t^2}{t + \frac{\lambda}{\alpha}}$$

where we have used (77) and the trivial upper bound of 1 on any probability. We further simplify the above equation to a sufficient condition, by plugging in $D_s = \tau\sqrt{\log \frac{1}{P_{\text{out}}}}$ the fact that $x\log(1/x) < \sqrt{x}, \forall 0 < x < 1$.

$$48\tau^2\sqrt{P_{\text{out}}} \frac{\exp\left(-\frac{2D_\tau^2}{9t}\right)}{\exp\left(-\frac{1}{2}C_E^2 \frac{\beta}{\lambda}D_\tau^2\right)} (\frac{Lt + \lambda}{\alpha t})^{d/2} \leq \frac{(1 - C_\alpha)t^2}{t + \frac{\lambda}{\alpha}} \tag{84}$$

Rearranging (84) yields the equivalent sufficient condition

$$48\tau^2\sqrt{P_{\text{out}}} \leq \frac{(1 - C_\alpha)t^2}{t + \frac{\lambda}{\alpha}} \frac{\exp\left(-\frac{1}{2}C_E^2 \frac{\beta}{\lambda}D_\tau^2\right)}{\exp\left(-\frac{2D_\tau^2}{9t}\right)} \left(\frac{\alpha t}{Lt + \lambda}\right)^{d/2}. \tag{85}$$

A sufficient condition for the above is

$$48\tau^2\sqrt{P_{\text{out}}} \leq (1 - C_\alpha)t(\frac{t}{t + \frac{\lambda}{\alpha}})^{\frac{d}{2}+1}\kappa_0^{-\frac{d}{2}} \frac{\exp\left(-\frac{1}{2}C_E^2 \frac{\beta}{\lambda}D_\tau^2\right)}{\exp\left(-\frac{2D_\tau^2}{9t}\right)}, \tag{86}$$

where $\kappa_0 = \frac{\beta}{\alpha}$ is the condition number. Since $\exp\left(-\frac{1}{2}C_E^2 \frac{\beta}{\lambda}D_\tau^2\right) \leq 1$, it suffices to lower bound the right-hand side. We focus on bounding the following auxiliary function (local to Case 1.1):

For $t > 0$, define

$$\mathcal{F}_{1.1}(t) := t\left(\frac{t}{t + \frac{\lambda}{\alpha}}\right)^{\frac{d}{2}+1} \exp\left(\frac{2D_\tau^2}{9t}\right).$$

*Subcase 1.1.a:* $0 < t < 2\kappa_0\sqrt{t}$. In this regime, we have the uniform lower bound

$$\mathcal{F}_{1.1}(t) \geq c_{1.1,\text{lb}},$$

where

$$C_{1.1} := \frac{2D_\tau^2}{9}, \qquad a_{1.1} := \frac{d}{4} + \frac{3}{2} = \frac{d + 6}{4},$$

and

$$c_{1.1,\text{lb}} := (3\kappa_0)^{-\left(\frac{d}{2}+1\right)} \left(\frac{e\,C_{1.1}}{a_{1.1}}\right)^{a_{1.1}} = (3\kappa_0)^{-\left(\frac{d}{2}+1\right)} \left(\frac{8eD_\tau^2}{9(d + 6)}\right)^{\frac{d+6}{4}}.$$

With the assumption $\lambda \leq 2\alpha$, and $t < 2\kappa_0\sqrt{t}$, we can derive that

$$\kappa_0\sqrt{t} \geq \kappa_0\frac{\lambda}{\beta} \geq \frac{\lambda}{\alpha},$$

where we used $\kappa_0 = \frac{\beta}{\alpha}$ in the last inequality. Together with $t < 2\kappa_0\sqrt{t}$ this yields

$$t + \frac{\lambda}{\alpha} \leq 2\kappa_0\sqrt{t} + \kappa_0\sqrt{t} = 3\kappa_0\sqrt{t}.$$

Hence

$$\frac{t}{t + \frac{\lambda}{\alpha}} \geq \frac{t}{3\kappa_0\sqrt{t}} = \frac{\sqrt{t}}{3\kappa_0}.$$

Plugging this into $\mathcal{F}_{1.1}(t)$ gives

$$\mathcal{F}_{1.1}(t) \geq t\left(\frac{\sqrt{t}}{3\kappa_0}\right)^{\frac{d}{2}+1} \exp\left(\frac{2D_\tau^2}{9t}\right) = (3\kappa_0)^{-\left(\frac{d}{2}+1\right)} t^{a_{1.1}} \exp\left(\frac{C_{1.1}}{t}\right),$$

with $a_{1.1}$ and $C_{1.1}$ as defined above.

Define

$$h_{1.1}(t) := t^{a_{1.1}} \exp\left(\frac{C_{1.1}}{t}\right), \qquad t > 0.$$

Then

$$\log h_{1.1}(t) = a_{1.1}\log t + \frac{C_{1.1}}{t}, \quad \frac{\mathrm{d}}{\mathrm{d}t}\log h_{1.1}(t) = \frac{a_{1.1}}{t} - \frac{C_{1.1}}{t^2} = \frac{a_{1.1}t - C_{1.1}}{t^2}.$$

Thus $\log h_{1.1}$ has a unique critical point at

$$t_{1.1}^* = \frac{C_{1.1}}{a_{1.1}} > 0,$$

and

$$\frac{\mathrm{d}^2}{\mathrm{d}t^2}\log h_{1.1}(t) = -\frac{a_{1.1}}{t^2} + \frac{2C_{1.1}}{t^3}, \quad \frac{\mathrm{d}^2}{\mathrm{d}t^2}\log h_{1.1}(t_{1.1}^*) = \frac{C_{1.1}}{(t_{1.1}^*)^3} > 0.$$

Hence $t_{1.1}^*$ is the global minimizer of $h_{1.1}$ on $(0,\infty)$, and

$$h_{1.1}(t) \geq h_{1.1}(t_{1.1}^*) = \left(\frac{C_{1.1}}{a_{1.1}}\right)^{a_{1.1}} \exp\left(\frac{C_{1.1}}{t_{1.1}^*}\right) = \left(\frac{C_{1.1}}{a_{1.1}}\right)^{a_{1.1}} e^{a_{1.1}} = \left(\frac{e\,C_{1.1}}{a_{1.1}}\right)^{a_{1.1}}, \quad \forall t > 0.$$

Combining this with the previous inequality yields

$$\mathcal{F}_{1.1}(t) \geq (3\kappa_0)^{-\left(\frac{d}{2}+1\right)}\left(\frac{4e\,C_{1.1}}{d+6}\right)^{\frac{d+6}{4}} = c_{1.1,\mathrm{lb}},$$

In conclusion, the sufficient condition becomes

$$48\tau^2\sqrt{P_{\mathrm{out}}} \leq (1 - C_\alpha)(3\kappa_0)^{-\left(\frac{d}{2}+1\right)}\left(\frac{4e\,C_{1.1}}{d+6}\right)^{\frac{d+6}{4}} \kappa_0^{-\frac{d}{2}} \exp(-\frac{1}{2}C_E^2\frac{\beta}{\lambda}D_\tau^2).$$

By taking the maximum of $P_{\mathrm{out}}$ (which occurs at $P_{\mathrm{out}} = 1/e$), we obtain the sufficient condition

$$33\tau^2 \leq (1 - C_\alpha)(3\kappa_0^2)^{-\left(\frac{d}{2}+1\right)}\left(\frac{4e\,C_{1.1}}{d+6}\right)^{\frac{d+6}{4}} \exp(-\frac{1}{2}C_E^2\frac{\beta}{\lambda}D_\tau^2).$$

In other words, for all sufficiently large $d$, the sufficient condition is satisfied if

$$(3\kappa_0^2)^{-\left(\frac{d}{2}\right)}\left(\frac{8e\,D_\tau^2}{9d}\right)^{\frac{d}{4}} \exp\left(-\frac{1}{2}C_E^2\frac{\beta}{\lambda}D_\tau^2\right) > 33\tau^2 \tag{87}$$

*Subcase 1.1.b:* $t > 4\kappa_0^2$. For $t > 4\kappa_0^2$, we use another bound on $P_{\mathrm{out}[0|t]}$ from Equation (79), which leads to the following condition:

$$48\tau^2\log\frac{1}{P_{\mathrm{out}}}P_{\mathrm{out}} \leq (1 - C_\alpha)\frac{(4\kappa_0^2)^2}{4\kappa_0^2 + 2\kappa_0^2}.$$

By taking the maximum of $P_{\mathrm{out}}\log(1/P_{\mathrm{out}})$ (which is $1/e$), we obtain the sufficient condition

$$18\tau^2 \leq (1 - C_\alpha)\frac{(4\kappa_0^2)^2}{4\kappa_0^2 + 2\kappa_0^2} \tag{88}$$

where the factor $1/e$ comes from the fact that $P_{\mathrm{out}}\log(1/P_{\mathrm{out}}) \leq 1/e$ for all $P_{\mathrm{out}} \in (0,1)$.

**Case 1.2:** $D_s < D_\tau$. In this case, the sufficient condition is given by

$$12D_\tau^2 \min(1, 4P_{\text{out}}) \frac{\exp\left(-\frac{2D_\tau^2}{9t}\right)}{\exp\left(-\frac{1}{2}C_E^2 \frac{\beta}{\lambda} D_\tau^2\right)} \left(\frac{Lt + \lambda}{\alpha t}\right)^{d/2} \le \frac{(1 - C_\alpha)t^2}{t + \frac{\lambda}{\alpha}}.$$

We consider two subcases based on which bound from Equations (77) and (79) is used:

$$48D_\tau^2 P_{\text{out}} \frac{\exp\left(-\frac{2D_\tau^2}{9t}\right)}{\exp\left(-\frac{1}{2}C_E^2 \frac{\beta}{\lambda} D_\tau^2\right)} \left(\frac{Lt + \lambda}{\alpha t}\right)^{d/2} \le \frac{(1 - C_\alpha)t^2}{t + \frac{\lambda}{\alpha}},$$

and

$$12D_\tau^2 \min(1, 4P_{\text{out}}) \le \frac{(1 - C_\alpha)t^2}{t + \frac{\lambda}{\alpha}}.$$

*Subcase 1.2.a:* $0 < t < 2\kappa_0 \sqrt{t}$. Similarly,

$$48D_\tau^2 P_{\text{out}} \le (1 - C_\alpha)t \left(\frac{t}{t + \frac{\lambda}{\alpha}}\right)^{\frac{d}{2}+1} \kappa_0^{-\frac{d}{2}} \frac{\exp\left(-\frac{1}{2}C_E^2 \frac{\beta}{\lambda} D_\tau^2\right)}{\exp\left(-\frac{2D_\tau^2}{9t}\right)}, \tag{89}$$

And all the calculation for $\mathcal{F}_{1.1}(t)$ is the same as in Subcase 1.1.a, so the sufficient condition is given by

$$48D_\tau^2 P_{\text{out}} \le (1 - C_\alpha)(3\kappa_0^2)^{-\left(\frac{d}{2}+1\right)} \left(\frac{4e\, C_{1.1}}{d+6}\right)^{\frac{d+6}{4}} \exp\left(-\frac{1}{2}C_E^2 \frac{\beta}{\lambda} D_\tau^2\right)$$

In other words, for all sufficiently large $d$, the sufficient condition is satisfied if

$$(1 - C_\alpha)(3\kappa_0^2)^{-\left(\frac{d}{2}\right)} \left(\frac{8e\, D_\tau^2}{9d}\right)^{\frac{d}{4}} \exp\left(-\frac{1}{2}C_E^2 \frac{\beta}{\lambda} D_\tau^2\right) > 48D_\tau^2 P_{\text{out}} \tag{90}$$

by taking the maximum of $P_{\text{out}}$.

*Subcase 1.2.b:* $t > 4\kappa_0^2$. In this regime, we have the sufficient condition

$$48D_\tau^2 P_{\text{out}} \le (1 - C_\alpha)\frac{(4\kappa_0^2)^2}{4\kappa_0^2 + 2\kappa_0^2} \tag{91}$$

**Case 2:** $\|x_t - x^*\| > \frac{1}{3}D_\tau$. In this regime, we use the simplified bound Equation (83). From the expansion rate condition, we have $C_E \sqrt{\frac{t + \frac{\lambda}{\beta}}{\frac{\lambda}{\beta}}} \ge \frac{1}{3}$. Therefore,

$$t + \frac{\lambda}{\beta} \ge \frac{1}{9C_E^2} \frac{\lambda}{\beta}.$$

With our assumption $\lambda \le 2\alpha$ and $C_E \sim \Theta(\sqrt{\frac{\log(d)}{d}})$ as in the theorem statement, we can show that $\frac{t^2}{t + \frac{\lambda}{\beta}} \ge \frac{1}{10C_E^2} \frac{\lambda}{\beta}$ for all sufficiently large $d$.

**Case 2.1:** $D_s \ge D_\tau$. In this case we have $\max(D_\tau^2, D_s^2) = D_s^2$, so it suffices that

$$9D_s^2 \min(1, 2P_{\text{out}} \exp(8\frac{\lambda}{\beta}C_E^2 D_\tau^2)) \le (1 - C_\alpha)\frac{1}{10C_E^2} \frac{\lambda}{\beta} \tag{92}$$

Plugging in $D_s = \tau\sqrt{\log\frac{1}{P_{\text{out}}}}$, (92) is equivalent to

$$9\tau^2 \log\frac{1}{P_{\text{out}}} \ \min\left(1, 2P_{\text{out}}\exp(8\tfrac{\lambda}{\beta}C_E^2 D_\tau^2)\right) \ \leq \ (1-C_\alpha)\tfrac{1}{10C_E^2}\tfrac{\lambda}{\beta} \tag{93}$$

Here the $t$-dependence has already been absorbed into the lower bound on $\frac{t^2}{t+\lambda/\beta}$, so we do not need an additional $t$-subcase split. We now split into two regimes based on the value of $P_{\text{out}}$ (via the $\min(\cdot,\cdot)$).

*Regime A: $P_{out} \geq \frac{1}{2}\exp(-8\tfrac{\lambda}{\beta}C_E^2 D_\tau^2)$.* In this regime, the minimum equals 1 and a sufficient condition is

$$9\tau^2 \log\frac{1}{P_{\text{out}}} \ \leq \ (1-C_\alpha)\tfrac{1}{10C_E^2}\tfrac{\lambda}{\beta} \tag{94}$$

Moreover, the case assumption itself implies

$$\log\frac{1}{P_{\text{out}}} \ \leq \ 8\tfrac{\lambda}{\beta}C_E^2 D_\tau^2$$

Therefore, a purely-parameter sufficient condition ensuring (94) for all $P_{\text{out}}$ in this regime is

$$720\tau^2 C_E^4 D_\tau^2 \leq (1-C_\alpha) \tag{95}$$

*Regime B: $P_{out} \leq \frac{1}{2}\exp(-8\tfrac{\lambda}{\beta}C_E^2 D_\tau^2)$.* In this regime, the minimum equals $2P_{\text{out}}\exp(8\tfrac{\lambda}{\beta}C_E^2 D_\tau^2)$ and Equation (93) reduces to

$$18\tau^2 P_{\text{out}}\log\frac{1}{P_{\text{out}}} \ \leq \ \frac{(1-C_\alpha)\tfrac{1}{10C_E^2}\tfrac{\lambda}{\beta}}{\exp(8\tfrac{\lambda}{\beta}C_E^2 D_\tau^2)}. \tag{96}$$

This yields the sufficient condition

$$67\tau^2 C_E^2 \exp(8\tfrac{\lambda}{\beta}C_E^2 D_\tau^2) \leq (1-C_\alpha)\tfrac{\lambda}{\beta} \tag{97}$$

**Case 2.2: $D_s < D_\tau$.** Likewise, here we have the following condition to satisfy

$$9D_\tau^2 \min(1, 2P_{\text{out}}\exp(8\tfrac{\lambda}{\beta}C_E^2 D_\tau^2)) \leq (1-C_\alpha)\tfrac{1}{10C_E^2}\tfrac{\lambda}{\beta}$$

The following two conditions are sufficient to satisfy the above inequality

$$18D_\tau^2 P_{\text{out}}\exp(8\tfrac{\lambda}{\beta}C_E^2 D_\tau^2) \leq (1-C_\alpha)\tfrac{1}{10C_E^2}\tfrac{\lambda}{\beta}$$

Because of Equation (91), we have

$$D_\tau^2 P_{\text{out}} \ \leq \ (1-C_\alpha)\frac{(4\kappa_0^2)^2}{48(4\kappa_0^2 + 4\kappa_0^3)} \ = \ (1-C_\alpha)\frac{\kappa_0^2}{12(1+\kappa_0)}.$$

Plugging this into the left-hand side yields

$$18D_\tau^2 P_{\text{out}}\exp(8\tfrac{\lambda}{\beta}C_E^2 D_\tau^2) \ \leq \ (1-C_\alpha)\frac{3}{2}\frac{\kappa_0^2}{1+\kappa_0} \ \exp(8\tfrac{\lambda}{\beta}C_E^2 D_\tau^2),$$

so it suffices that

$$\frac{3}{2}\frac{\kappa_0^2}{1+\kappa_0} \ \exp(8\tfrac{\lambda}{\beta}C_E^2 D_\tau^2) \ \leq \ \tfrac{1}{10C_E^2}\tfrac{\lambda}{\beta} \tag{98}$$

In summary, (92) is ensured by either the constant-parameter condition (95) (Regime A), or the small-$P_{\text{out}}$ condition (97) (Regime B).

**Step 7: Collecting the boxed conditions and an explicit final regime.** We now show that the boxed sufficient conditions appearing in the case analysis can be met simultaneously by an explicit parameter choice. Throughout, treat $(\beta, \alpha, \lambda_{\max}, C_\alpha)$ and hence $\kappa_0 := \beta/\alpha$ as fixed constants (independent of $d$), and assume $\lambda \leq \lambda_{\max}$.

**(7.1) Choosing $D_\tau^2 = \Theta(d)$ and deriving $P_{\mathbf{out}} \le 1/d$.**   Pick a constant $b > 0$ and set

$$D_\tau^2 := bd.$$

To satisfy the key scaling requirement Equation (74), it suffices that

$$b \ \ge \ \frac{9}{4} \frac{\lambda_{\max}}{\beta},$$

since then $\frac{2D_\tau^2}{9}\frac{\beta}{\lambda_{\max}} = \frac{2b}{9}\frac{\beta}{\lambda_{\max}}d \ge \frac{d}{2}$.

Next, the boxed condition Equation (91) implies

$$48D_\tau^2 P_{\text{out}} \le (1 - C_\alpha)\frac{(4\kappa_0^2)^2}{4\kappa_0^2 + 2\kappa_0^2} = (1 - C_\alpha)\frac{8}{3}\kappa_0^2,$$

so

$$P_{\text{out}} \le \frac{1 - C_\alpha}{18}\frac{\kappa_0^2}{D_\tau^2} = \frac{1 - C_\alpha}{18}\frac{\kappa_0^2}{b} \cdot \frac{1}{d}.$$

Therefore, if we additionally choose

$$b \ \ge \ \frac{1 - C_\alpha}{18}\kappa_0^2,$$

then the same boxed condition Equation (91) yields the clean tail-mass bound

$$P_{\text{out}} \le \frac{1}{d}.$$

**(7.2) An explicit $\tau$ upper bound from $\kappa_0$.**   From the boxed condition Equation (88) (and simplifying its right-hand side),

$$18\tau^2 \le (1 - C_\alpha)\frac{(4\kappa_0^2)^2}{4\kappa_0^2 + 2\kappa_0^2} = (1 - C_\alpha)\frac{8}{3}\kappa_0^2,$$

it suffices to impose

$$\tau \ \le \ \tau_{\max} \ := \ \frac{2}{3\sqrt{3}}\sqrt{1 - C_\alpha}\,\kappa_0.$$

**(7.3) Choosing $C_E$ from the boxed $C_E$-constraints.**   We now *derive* a sufficient scaling for $C_E$ directly from the boxed inequalities Equations (97) and (98). Let $u := C_E^2$ and $C_a := 8\frac{\lambda}{\beta}D_\tau^2$. Then:

- From Equation (97) we have

$$67\tau^2\,u\,e^{C_a u} \le (1 - C_\alpha)\frac{\lambda}{\beta} \quad\implies\quad u\,e^{C_a u} \le \frac{1 - C_\alpha}{67\tau^2}\frac{\lambda}{\beta}.$$

- From Equation (98) we have

$$\frac{3}{2}\frac{\kappa_0^2}{1 + \kappa_0}e^{C_a u} \le \frac{1}{10u}\frac{\lambda}{\beta} \quad\implies\quad u\,e^{C_a u} \le \frac{1 + \kappa_0}{15\kappa_0^2}\frac{\lambda}{\beta}.$$

Therefore it suffices to choose $u$ such that

$$u\,e^{C_a u} \ \le \ \frac{\lambda}{\beta}\,\min\left\{\frac{1 - C_\alpha}{67\tau^2}, \frac{1 + \kappa_0}{15\kappa_0^2}\right\}.$$

Under $D_\tau^2 = bd$ (so $C_a = \Theta(d)$) and $\lambda \le \lambda_{\max}$, one concrete sufficient choice is

$$C_E^2 \le \frac{\beta}{16\lambda_{\max}D_\tau^2}\log d \ = \ \frac{\beta}{16\lambda_{\max}b}\frac{\log d}{d},$$

for which $C_a u \le \frac{1}{2}\log d$ and hence

$$u\,e^{C_a u} \ \le \ \frac{\beta}{16\lambda_{\max}b}\frac{\log d}{d}d^{1/2} \ = \ \frac{\beta}{16\lambda_{\max}b}\frac{\log d}{\sqrt{d}} \xrightarrow[d\to\infty]{} 0.$$

Thus both boxed $C_E$-constraints Equations (97) and (98) hold for all sufficiently large $d$.

**(7.4) Compatibility with the remaining boxed conditions.** At this point, the only remaining requirements are constraints on $(P_{\text{out}}, D_\tau, C_E)$ that ensure the boxed inequalities used in Case 1.2 and Case 2.2.[4] We list them explicitly:

- *(Tail-mass scaling.)* The boxed condition Equation (91) implies

$$P_{\text{out}} \le \frac{1 - C_\alpha}{18} \frac{\kappa_0^2}{D_\tau^2}.$$

  Under $D_\tau^2 = bd$, choosing $b \ge \frac{1-C_\alpha}{18} \kappa_0^2$ yields $P_{\text{out}} \le 1/d$.

- *(Key $D_\tau$ scaling.)* The boxed condition Equation (74) is enforced by taking $D_\tau^2 = bd$ with

$$b \ge \frac{9}{4} \frac{\lambda_{\max}}{\beta}.$$

- *(Expansion-rate scale $C_E$.)* Choose $C_E$ as in Paragraph (iii), i.e.

$$C_E^2 \le \frac{\beta}{16\lambda_{\max}b} \frac{\log d}{d},$$

  which in particular implies $C_E^2 = \Theta((\log d)/d)$.

With these choices, the remaining boxed conditions are satisfied by choosing $b$ above an explicit (parameter-only) lower bound:

- *(Case 1.2.)* The boxed growth condition Equation (90) holds provided the exponential-in-$d$ term has positive rate, i.e.

$$\log\left(\frac{8eb}{9}\right) > 2\log(3\kappa_0^2) \quad \Longleftrightarrow \quad b > \frac{81}{8e}\kappa_0^4.$$

**Conclusion.** Choose constants $b$ and $\tau$, and pick $C_E = C_E(d)$, such that

$$b \ge \max\left\{\frac{9}{4}\frac{\lambda_{\max}}{\beta}, \ \frac{1-C_\alpha}{18}\kappa_0^2, \ \frac{81}{8e}\kappa_0^4\right\}, \qquad \tau \le \tau_{\max}.$$

Let $D_\tau^2 = bd$ and assume $P_{\text{out}} \le 1/d$ (which is implied by Equation (91) under the chosen $b$). Finally, choose $C_E$ to satisfy the explicit upper bound from Paragraph (iii),

$$C_E^2 \le \frac{\beta}{16\lambda_{\max} D_\tau^2} \log d = \frac{\beta}{16\lambda_{\max}b} \frac{\log d}{d}.$$

Then all boxed sufficient conditions used in the Case 1.2 and Case 2.2 analysis hold for all sufficiently large $d$. Therefore the sufficient condition Equation (63) holds, completing the proof of Theorem 4.3.

$\square$

## B.4 Proof for Theorem 4.4

Here we provide the proof for Theorem 4.4, which is the main theorem in this paper and is built on top of the proof of Theorem 4.3.

---

[4]With $P_{\text{out}} \le 1/d$ and $D_\tau^2 = bd$, we have $D_s = \tau\sqrt{\log(1/P_{\text{out}})} = \tau\sqrt{\log d}$, so $D_s < D_\tau$ for all sufficiently large $d$. Hence only the $D_s < D_\tau$ branches (Case 1.2 and Case 2.2) are relevant asymptotically.

**Theorem 4.4** (Optimizer displacement). *Let Assumption 4.1 and Assumption 4.2 hold. For any smoothing parameter $t > 0$, let $x_t^*$ denote the unique minimizer of the smoothed function $g(x; t)$ within the strongly convex region $\mathcal{R}_{SC}(t)$, as established in Theorem 4.3. If $x_t^*$ also lies within the region of local strong convexity and smoothness for the original function $f(x)$, i.e., $x_t^* \in B_\tau = \{x \mid \|x - x^*\| < D_\tau\}$, then the optimizer displacement $\|x_t^* - x^*\|$ is bounded by:*

$$\|x_t^* - x^*\| \leq \min\left\{\frac{(1 - C_\alpha)t}{C_\alpha}\frac{1}{4D_\tau}, (D_\tau + \tau)\frac{t + \frac{\lambda}{\alpha}}{C_\alpha t}\right\} + \frac{\kappa_0 - 1}{2C_\alpha}\sqrt{\frac{1}{2\pi(\frac{1}{t} + \frac{\alpha}{\lambda})}}. \tag{8}$$

*where $C_\alpha \in (0, 1)$ is the strong convexity parameter from Theorem 4.3, and $D_\tau$, $\tau$ are the parameters from Assumption 4.1.*

*Proof.* **Step 1: Setup and Reduction via Strong Convexity.** By the strong convexity of $g(x; t)$ established in Theorem 4.3, the function $g(x; t)$ is $\frac{C_\alpha \lambda}{t + \frac{\lambda}{\alpha}}$-strongly convex within the region $\mathcal{R}_{SC}(t)$. Since $x_t^*$ is the minimizer of $g(x; t)$ and $x^*$ lies within the strongly convex region, the distance to the optimizer is bounded by the gradient norm at the original optimizer $x^*$:

$$\|x_t^* - x^*\| \leq \frac{1}{\frac{C_\alpha \lambda}{t + \frac{\lambda}{\alpha}}}\|\nabla g(x^*; t)\| = \frac{t + \frac{\lambda}{\alpha}}{C_\alpha \lambda}\|\nabla g(x^*; t)\|.$$

Using Tweedie's formula, the gradient at $x^*$ is given by

$$\nabla g(x^*; t) = \frac{\lambda}{t}(x^* - \mathbb{E}[y|x^*]).$$

Without loss of generality, we assume $x^* = 0$ in this proof. Consequently, the problem reduces to bounding the conditional expectation $\|\mathbb{E}[y|x^*]\|$. Therefore, we have the reduction:

$$\|x_t^* - x^*\| \leq \frac{t + \frac{\lambda}{\alpha}}{C_\alpha t}\|\mathbb{E}[y|x^*]\|.$$

**Step 2: Decomposition.** We decompose the expectation into contributions from inside and outside the ball $B_\tau$:

$$\|\mathbb{E}[y|x^*]\| \leq \underbrace{\|\mathbb{E}[y|y \in B_\tau, x^*]\| \cdot P(y \in B_\tau|x^*)}_{\text{Term I: Local Conditional Mean}} + \underbrace{\|\mathbb{E}[y|y \notin B_\tau, x^*]\| \cdot P(y \notin B_\tau|x^*)}_{\text{Term II: Tail Conditional Mean}}. \tag{99}$$

And for the bias bound, we have

$$\|x_t^* - x^*\| \leq \left(\underbrace{\frac{t + \frac{\lambda}{\alpha}}{C_\alpha t}\|\mathbb{E}[y|y \in B_\tau, x^*]\| \cdot P(y \in B_\tau|x^*)}_{\text{Term I: Local Asymmetry}} + \underbrace{'\frac{t + \frac{\lambda}{\alpha}}{C_\alpha t}\|\mathbb{E}[y|y \notin B_\tau, x^*]\| \cdot P(y \notin B_\tau|x^*)'}_{\text{Term II: Tail Contribution}}\right) \tag{100}$$

**Step 3: Bounding Term I (Local Asymmetry).** We now bound the contribution to the bias coming from the *local* region $B_\tau = \{y : \|y - x^*\| \leq D_\tau\}$, where $f$ is $\alpha$-strongly convex and $\beta$-smooth. Intuitively, if $f$ were perfectly symmetric around $x^*$, then the posterior mean under the Gaussian smoothing would remain centered and this term would vanish. Thus, Term I captures the worst-case *directional asymmetry* of the local landscape permitted by the curvature bounds $(\alpha, \beta)$. Formally, it suffices to control the directional conditional mean $\mathbb{E}[\langle y, \hat{e}\rangle \mid x^*, y \in B_\tau]$ uniformly over unit vectors $\hat{e}$; we will upper bound this quantity by replacing $p_0(y) \propto e^{-f(y)/\lambda}$ on $B_\tau$ with extremal envelopes consistent with the local curvature constraints.

**(3.1) Worst-case envelope inside $B_\tau$.** On $B_\tau = \{\|y\| \le D_\tau\}$, by $\alpha$-strong convexity and $\beta$-smoothness around $x^*$ (with $\nabla f(x^*) = 0$), we have

$$\frac{\alpha}{2}\|y\|^2 \le f(y) - f(0) \le \frac{\beta}{2}\|y\|^2.$$

Since $p_0(y) \propto \exp(-f(y)/\lambda)$, this implies the pointwise envelope

$$\exp\Big(-\frac{\beta}{2\lambda}\|y\|^2\Big) \;\le\; \frac{p_0(y)}{p_0(0)} \;\le\; \exp\Big(-\frac{\alpha}{2\lambda}\|y\|^2\Big), \qquad y \in B_\tau.$$

Define $\tilde{p}_0^{\max}(y) := \exp(-\frac{\beta}{2\lambda}\|y\|^2)$ and $\tilde{p}_0^{\min}(y) := \exp(-\frac{\alpha}{2\lambda}\|y\|^2)$. We omit $p_0(0)$ in the following calculations as it is a constant appearing in both numerator and denominator.

**(3.2) Directional conditional mean bound.** We calculate the conditional expectation separately for two hemispheres: one where $\langle y, \hat{e}\rangle > 0$ and the other where $\langle y, \hat{e}\rangle < 0$.

$$
\begin{aligned}
\mathbb{E}_{0|t}[\langle y, \hat{e}\rangle | x^*, y \in B_\tau] &= \frac{\int_{B_\tau} \langle y, \hat{e}\rangle p_0(y) p_{t|0}(x^* \mid y) dy}{\int_{B_\tau} p_0(y) p_{t|0}(x^* \mid y) dy} \\
&= \frac{\int_{B_\tau, \langle y, \hat{e}\rangle > 0} \langle y, \hat{e}\rangle p_0(y) p_{t|0}(x^* \mid y) dy + \int_{B_\tau, \langle y, \hat{e}\rangle < 0} \langle y, \hat{e}\rangle p_0(y) p_{t|0}(x^* \mid y) dy}{\int_{B_\tau, \langle y, \hat{e}\rangle > 0} p_0(y) p_{t|0}(x^* \mid y) dy + \int_{B_\tau, \langle y, \hat{e}\rangle < 0} p_0(y) p_{t|0}(x^* \mid y) dy} \\
&\le \frac{\int_{B_\tau, \langle y, \hat{e}\rangle > 0} \langle y, \hat{e}\rangle \tilde{p}_0^{\max}(y) p_{t|0}(x^* \mid y) dy + \int_{B_\tau, \langle y, \hat{e}\rangle < 0} \langle y, \hat{e}\rangle \tilde{p}_0^{\min}(y) p_{t|0}(x^* \mid y) dy}{\int_{B_\tau, \langle y, \hat{e}\rangle > 0} \tilde{p}_0^{\max}(y) p_{t|0}(x^* \mid y) dy + \int_{B_\tau, \langle y, \hat{e}\rangle < 0} \tilde{p}_0^{\min}(y) p_{t|0}(x^* \mid y) dy}
\end{aligned}
\tag{101}
$$

The inequality in (101) follows from the pointwise envelope $\tilde{p}_0^{\max}(y) \ge p_0(y)/p_0(0) \ge \tilde{p}_0^{\min}(y)$ on $B_\tau$: on the half-space $\{y_1 > 0\}$ the integrand $\langle y, \hat{e}\rangle = y_1$ is nonnegative, so replacing $p_0$ by the upper envelope increases the numerator; on $\{y_1 < 0\}$ the integrand is nonpositive, so replacing $p_0$ by the lower envelope increases the numerator (makes it less negative). In the denominator the integrand is nonnegative everywhere, hence using $\tilde{p}_0^{\max}$ on $\{y_1 > 0\}$ and $\tilde{p}_0^{\min}$ on $\{y_1 < 0\}$ yields an upper bound on the ratio.

**(3.3) Monotonicity in $D_\tau$ and the limit $D_\tau \to \infty$.** We next show that the upper bound in (101) is monotone in the ball radius $D_\tau$. This allows us to take $D_\tau \to \infty$ and reduce the calculation to standard Gaussian integrals.

$$
\text{RHS of (101)} = \frac{\int_{r=0}^{D_\tau} \Big[\int_{\|y\|=r, \langle y, \hat{e}\rangle > 0} \langle y, \hat{e}\rangle \tilde{p}_0^{\max}(y) p_{t|0}(x^*|y) dy + \int_{\|y\|=r, \langle y, \hat{e}\rangle < 0} \langle y, \hat{e}\rangle \tilde{p}_0^{\min}(y) p_{t|0}(x^*|y) dy\Big] dr}{\int_{r=0}^{D_\tau} \Big[\int_{\|y\|=r, \langle y, \hat{e}\rangle > 0} \tilde{p}_0^{\max}(y) p_{t|0}(x^*|y) dy + \int_{\|y\|=r, \langle y, \hat{e}\rangle < 0} \tilde{p}_0^{\min}(y) p_{t|0}(x^*|y) dy\Big] dr}
\tag{102}
$$

To this end, express both numerator and denominator in (101) in polar coordinates. At each radius $r$, define the corresponding "shell ratio" as in (103). Then (102) is precisely the ratio of the integrals of these shell contributions over $r \in [0, D_\tau]$.

$$
\frac{\int_{\|y\|=r, \langle y, \hat{e}\rangle > 0} \langle y, \hat{e}\rangle \tilde{p}_0^{\max}(y) p_{t|0}(x^*|y) dy + \int_{\|y\|=r, \langle y, \hat{e}\rangle < 0} \langle y, \hat{e}\rangle \tilde{p}_0^{\min}(y) p_{t|0}(x^*|y) dy}{\int_{\|y\|=r, \langle y, \hat{e}\rangle > 0} \tilde{p}_0^{\max}(y) p_{t|0}(x^*|y) dy + \int_{\|y\|=r, \langle y, \hat{e}\rangle < 0} \tilde{p}_0^{\min}(y) p_{t|0}(x^*|y) dy}.
\tag{103}
$$

For a fixed radius $r > 0$, the shell ratio in (103) is increasing in $r$. Indeed, after rotating so that $\hat{e} = e_1$, the integrands depend on $y$ only through $y_1$ and $\|y\| = r$, and the resulting one-dimensional form reduces (up to positive multiplicative factors) to

$$h(y) := \frac{ye^{-y^2/a} - ye^{-y^2/b}}{e^{-y^2/a} + e^{-y^2/b}} = y \tanh\Big(\frac{y^2}{2}\Big(\frac{1}{b} - \frac{1}{a}\Big)\Big), \qquad a > b > 0,$$

This ratio is increasing in $y$ (one can verify this by reducing to the 1D form $y \mapsto \frac{ye^{-y^2/a} - ye^{-y^2/b}}{e^{-y^2/a} + e^{-y^2/b}} = y \tanh(y^2(\frac{1}{b} - \frac{1}{a})/2)$ with $a > b > 0$). Consequently, writing (102) as $\frac{\int_0^{D_\tau} u(r)\,dr}{\int_0^{D_\tau} v(r)\,dr}$ with $u(r)/v(r)$ increasing and $v(r) > 0$, we obtain that the ratio is increasing in $D_\tau$ (a standard "ratio of integrals" monotonicity argument).

Therefore, the RHS of (101) is monotone increasing in $D_\tau$, and we may upper bound it by taking the limit $D_\tau \to \infty$. In the next step we evaluate the resulting Gaussian half-space integrals by bounding the numerator and denominator separately.

**(3.4) Closed-form bound via Gaussian integrals.** Taking $D_\tau \to \infty$, the terms in (101) become Gaussian half-space integrals with quadratic exponents. We evaluate the numerator and denominator separately and then combine them to obtain a closed-form upper bound.

$$\int_{B_\tau, \langle y, \hat{e} \rangle > 0} \langle y, \hat{e} \rangle \tilde{p}_0^{\max}(y) p_{t|0}(x^* \mid y) dy = \int_{B_\tau, \langle y, \hat{e} \rangle > 0} \langle y, \hat{e} \rangle \tilde{p}_0^{\max}(y) p_{t|0}(x^* \mid y) dy$$
$$= \int_{B_\tau, \langle y, \hat{e} \rangle > 0} \langle y, \hat{e} \rangle \exp\left(-\left(\frac{\beta}{2\lambda} + \frac{1}{2t}\right) \|y\|^2\right) dy$$

Using $B_\tau^+$ to represent $B_\tau \cap \{y : \langle y, \hat{e} \rangle > 0\}$, we have:

$$\int_{B_\tau^+} \langle y, \hat{e} \rangle \exp\left((-\frac{\beta}{2\lambda} - \frac{1}{2t}) \|y\|^2\right) dy = \frac{1}{-\frac{\beta}{\lambda} - \frac{1}{t}} \int_{B_\tau^+} \nabla\left(\exp\left((-\frac{\beta}{2\lambda} - \frac{1}{2t}) \|y\|^2\right)\right)^\top \hat{e} dy.$$

Applying Gauss's theorem yields:

$$\frac{1}{-\frac{\beta}{\lambda} - \frac{1}{t}} \int_{B_\tau^+} \nabla\left(\exp\left((-\frac{\beta}{2\lambda} - \frac{1}{2t}) \|y\|^2\right)\right)^\top \hat{e} dy = \frac{1}{-\frac{\beta}{\lambda} - \frac{1}{t}} \iint_{\partial B_\tau^+} \exp\left((-\frac{\beta}{2\lambda} - \frac{1}{2t}) \|y\|^2\right) (\hat{e}^\top dS)$$

The surface integral splits into two parts: the hemispherical surface $\partial B_\tau^+ \cap \{y : \|y\| = D_\tau\}$ and the flat surface $\{y : \langle y, \hat{e} \rangle = 0\} \cap B_r$. The integral over the hemisphere vanishes as $D_\tau \to \infty$:

$$\lim_{D_\tau \to \infty} \iint_{\partial B_\tau^+ \cap \{y : \|y\| = D_\tau\}} \exp\left(\left(-\frac{\beta}{2\lambda} - \frac{1}{2t}\right) \|y\|^2\right) \hat{e}^\top dS$$
$$= \lim_{D_\tau \to \infty} \exp\left(\left(-\frac{\beta}{2\lambda} - \frac{1}{2t}\right) \|D_\tau\|^2\right) \frac{\pi^{(d-1)/2}}{\Gamma((d+1)/2)} D_\tau^{d-1} \hat{e} = 0$$

For the integral over the flat surface $\{y : \langle y, \hat{e} \rangle = 0\} \cap B_r$:

$$\iint_{\langle y, \hat{e} \rangle = 0, y \in B_\tau} \exp\left((-\frac{\beta}{2\lambda} - \frac{1}{2t}) \|y\|^2\right) \hat{e}^\top dS = \frac{(2\pi)^{(d-1)/2}}{(\frac{\beta}{\lambda} + \frac{1}{t})^{(d-1)/2} \Gamma((d-1)/2)} \gamma(\frac{d-1}{2}, D_\tau^2(\frac{\beta}{2\lambda} + \frac{1}{2t}))$$

Take the limit, we have that

$$\lim_{D_\tau \to \infty} \iint_{\langle y, \hat{e} \rangle = 0, y \in B_\tau} \exp\left((-\frac{\beta}{2\lambda} - \frac{1}{2t}) \|y\|^2\right) \hat{e} dS = \frac{(2\pi)^{(d-1)/2}}{(\frac{\beta}{\lambda} + \frac{1}{t})^{(d-1)/2}}$$

As for the denominator of (101), we have that The gaussian integral can be calculated that

$$\lim_{D_\tau \to \infty} \int_{B_\tau} \exp\left((-\frac{\beta}{2\lambda} - \frac{1}{2t}) \|y\|^2\right) dy = \frac{(2\pi)^{(d)/2}}{(\frac{\beta}{\lambda} + \frac{1}{t})^{(d)/2}}$$

It is straightforward to show that for the other side of the ball, we just have to replace $\frac{\beta}{\lambda}$ with $\frac{\alpha}{\lambda}$. Hereby, we have the final bound for Equation (101).

**(3.5) Final bound for the conditional expectation inside the ball.** As we have shown that the conditional expectation is monotonically increasing in $D_\tau$, we can take the limit of $D_\tau$ to infinity for equation Equation (101) to get upper bound. In what follows we separately upper bound the numerator and denominator of (101). We first calculate the terms in the numerator:

$$\mathbb{E}_{0|t}[\langle y, \hat{e}\rangle | x^*, y \in B_\tau] \le \frac{\frac{1}{-\frac{\beta}{\lambda}-\frac{1}{t}}\frac{(2\pi)^{(d-1)/2}}{(\frac{\beta}{\lambda}+\frac{1}{t})^{(d-1)/2}} - \frac{1}{-\frac{\alpha}{\lambda}-\frac{1}{t}}\frac{(2\pi)^{(d-1)/2}}{(\frac{\alpha}{\lambda}+\frac{1}{t})^{(d-1)/2}}}{\frac{(2\pi)^{(d)/2}}{(\frac{\beta}{\lambda}+\frac{1}{t})^{(d)/2}} + \frac{(2\pi)^{(d)/2}}{(\frac{\alpha}{\lambda}+\frac{1}{t})^{(d)/2}}}$$

Collecting the limiting numerator/denominator terms from the two half-spaces (with parameters $\beta$ and $\alpha$, respectively) and simplifying yields the following expression. For notational convenience, define

$$T(\alpha, \lambda, t) := \frac{\frac{1}{t} + \frac{\beta}{\lambda}}{\frac{1}{t} + \frac{\alpha}{\lambda}}.$$

And we can further simplify it to

$$\frac{1}{\sqrt{2\pi}} \frac{-\frac{1}{\left(\frac{1}{t}+\frac{\beta}{\lambda}\right)^{\frac{d+1}{2}}} + \frac{1}{\left(\frac{1}{t}+\frac{\alpha}{\lambda}\right)^{\frac{d+1}{2}}}}{\frac{1}{\left(\frac{1}{t}+\frac{\alpha}{\lambda}\right)^{\frac{d}{2}}} + \frac{1}{\left(\frac{1}{t}+\frac{\beta}{\lambda}\right)^{\frac{d}{2}}}} = \frac{1}{\sqrt{2\pi}} \frac{T(\alpha, \lambda, t)^{\frac{d+1}{2}} - 1}{T(\alpha, \lambda, t)^{\frac{d}{2}} + 1} \frac{1}{\sqrt{\frac{1}{t}+\frac{\beta}{\lambda}}}.$$

**(3.6) A uniform simplification.** The closed-form expression above can be further simplified into a dimension-free bound in terms of the local condition number $\kappa_0 = \beta/\alpha$. To this end, define $\phi(T) := \frac{T^{\frac{d+1}{2}}-1}{T^{\frac{d}{2}}+1}$ and consider

$$L_2 := \phi(T)\sqrt{\frac{1}{t}+\frac{\alpha}{\lambda}}.$$

We next upper bound $\phi(T)$ by $\sqrt{T}-1$.

$$\frac{1}{C_\alpha}\frac{\lambda}{\alpha}\left(\frac{\alpha}{\lambda}+\frac{1}{t}\right)\left[\frac{1}{\sqrt{2\pi}}\underbrace{\frac{T(\alpha,\lambda,t)^{\frac{d+1}{2}}-1}{T(\alpha,\lambda,t)^{\frac{d}{2}}+1}}_{:=\phi(T)}\frac{1}{\sqrt{\frac{1}{t}+\frac{\beta}{\lambda}}}\right] \tag{104}$$

Here we consider $\phi(T) = \frac{T^{\frac{d+1}{2}}-1}{T^{\frac{d}{2}}+1}$ (matching the definition in (104)) and $L_2 = \phi(T)\sqrt{\frac{1}{t}+\frac{\alpha}{\lambda}}$. We calculate the derivative of $\phi(T)$:

$$\frac{d\phi(T)}{dT} = \frac{\frac{d+1}{2}T^{\frac{d-1}{2}}(T^{\frac{d}{2}}+1) - \frac{d}{2}T^{\frac{d}{2}-1}(T^{\frac{d+1}{2}}-1)}{(T^{\frac{d}{2}}+1)^2}.$$

After simplification, this yields:

$$\frac{d}{dT}\log\phi(T) \le \frac{1}{2T}.$$

This leads to

$$\phi(T) \le \sqrt{T}-1$$

Substituting $\phi(T) \le \sqrt{T}-1$ and using $\sqrt{T} = \sqrt{\frac{\frac{1}{t}+\frac{\beta}{\lambda}}{\frac{1}{t}+\frac{\alpha}{\lambda}}}$ yields

$$L_2 \le \sqrt{\frac{1}{t}+\frac{\beta}{\lambda}} - \sqrt{\frac{1}{t}+\frac{\alpha}{\lambda}} = \frac{\frac{\beta-\alpha}{\lambda}}{\sqrt{\frac{1}{t}+\frac{\beta}{\lambda}}+\sqrt{\frac{1}{t}+\frac{\alpha}{\lambda}}} \le \frac{\alpha}{\lambda}\frac{\kappa_0-1}{2\sqrt{\frac{1}{t}+\frac{\alpha}{\lambda}}}.$$

Therefore, for Equation (104), we have

$$\frac{1}{C_\alpha}\frac{\lambda}{\alpha}\left(\frac{\alpha}{\lambda}+\frac{1}{t}\right)\left[\frac{1}{\sqrt{2\pi}}\frac{T(\alpha,\lambda,t)^{\frac{d+1}{2}}-1}{T(\alpha,\lambda,t)^{\frac{d}{2}}+1}\frac{1}{\sqrt{\frac{1}{t}+\frac{\beta}{\lambda}}}\right]\leq\frac{\kappa_0-1}{2C_\alpha}\sqrt{\frac{1}{2\pi(\frac{1}{t}+\frac{\alpha}{\lambda})}}.$$

Note that as $t\to 0$, this term grows in the order of $\sqrt{t}$, and as $t\to\infty$, it is bounded by $\frac{\kappa_0-1}{2C_\alpha}\sqrt{\frac{\lambda}{2\pi\alpha}}$.

**Step 4: Bounding Term II (Tail Contribution).**

From Equation (63) together with Equations (67) and (68) in the proof of Theorem 4.5,

$$P_{\mathrm{out}[0|t]}|_{x_t=x^*}9\max(D_s^2,D_\tau^2)\leq\frac{(1-C_\alpha)t^2}{t+\frac{\lambda}{\alpha}},$$

where $D_s$ is as defined in Table 3. Therefore, we have that

$$P_{\mathrm{out}[0|t]}|_{x_t=x^*}(D_\tau+\tau)\leq\frac{(1-C_\alpha)t^2}{t+\frac{\lambda}{\alpha}}\frac{D_\tau+\tau}{9D_\tau^2}\leq\frac{(1-C_\alpha)t^2}{t+\frac{\lambda}{\alpha}}\frac{1}{4D_\tau}$$

And given that $P_{\mathrm{out}[0|t]}|_{x_t=x^*}\leq 1$, we have that

$$P_{\mathrm{out}[0|t]}|_{x_t=x^*}(D_\tau+\tau)\leq(D_\tau+\tau)$$

As the conditional expectation $\mathbb{E}[\|x-x^*\|\,|x_t=x^*,x\notin B_\tau]$ is bounded by $D_\tau+\frac{\tau^2}{2D_\tau}$, we have the total conditional expectation bound for the second term

$$\frac{t+\frac{\lambda}{\alpha}}{C_\alpha t}\|\mathbb{E}[y|y\notin B_\tau,x^*]\|\cdot P(y\notin B_\tau|x^*)$$

$$\leq\frac{t+\frac{\lambda}{\alpha}}{C_\alpha t}P_{\mathrm{out}[0|t]}|_{x_t=x^*}(D_\tau+\tau)$$

$$\leq\min\left\{\frac{(1-C_\alpha)t}{C_\alpha}\frac{1}{4D_\tau},(D_\tau+\tau)\frac{t+\frac{\lambda}{\alpha}}{C_\alpha t}\right\}$$

**Step 5: Combination.** Combining Terms I and II from the decomposition (99), we have the total conditional expectation:

$$\|x_t^*-x^*\|\leq\frac{\kappa_0-1}{2C_\alpha}\sqrt{\frac{1}{2\pi(\frac{1}{t}+\frac{\alpha}{\lambda})}}$$

$$+\min\left\{\frac{(1-C_\alpha)t}{C_\alpha}\frac{1}{4D_\tau},(D_\tau+\tau)\frac{t+\frac{\lambda}{\alpha}}{C_\alpha t}\right\}.$$

$\square$

## B.5 Bias compared to convex radius

*Remark* B.5. Though the convex radius seems expanding with order of $O(\sqrt{t})$, and the bias seems expanding with order of $O(t)$, the convex radius is actually much larger than the bias because of the Theorem B.6. One can show that only when $t$ is large enough, the bias is larger than $\frac{1}{4}D_\tau$. Namely at least

$$t\frac{1-C_\alpha}{C_\alpha}\frac{1}{4D_\tau}\geq\frac{1}{8}D_\tau$$

which implies

$$t\geq\frac{1}{2}D_\tau^2\frac{C_\alpha}{1-C_\alpha}$$

Pluging this into Equation (7). One can show that the convex radius

$$\mathcal{R}_{SC}(t) \geq C_E \sqrt{\frac{\frac{1}{2}\frac{C_\alpha}{1-C_\alpha}}{\frac{\lambda}{\beta}}} D_\tau^2 \sim \Theta(\sqrt{d \log d})$$

Recall the sufficient condition for $C_E$ as in theorem statement Theorem 4.3 and Equation (20). well on the other hand, the bias is bounded by $(D_\tau + \tau)\frac{2}{C_\alpha} \sim \Theta(\sqrt{d})$. Therefore, we have that

$$\mathcal{R}_{SC}(t) \gg \|x_t - x^*\|$$

holds for all $t$ given $d$ is sufficiently large.

### B.6 Other Auxilliary Results

**Theorem B.6** (Covex Region Lower Bound). *Let all the conditions in Theorem 4.3 hold. The convex region is lower bounded by the following inequality for all t for sufficiently large d.*

$$\mathcal{R}_{SC}(t) \geq \frac{1}{4}D_\tau \tag{105}$$

*and there exists $t_F$,*

$$t_F \geq (\frac{1}{16C_E^2} + 1)\min(\frac{\lambda}{\beta}, \tau^2) \tag{106}$$

*that for any $t < t_F$ $\mathcal{R}_{SC}(t)$ is not shrinking.*

At the beginning stage, when $t$ is small, the landscape of $g(x;t)$ is approximately the same as the landscape of $f(x)$. Therefore, the convex region should be very similar to the convex region of $f(x)$. Here we improve Theorem 4.3 by the above theorem showing that when $t$ is small, the radius of the convex region does not start from 0 but instead, should be at least of the same order as $D_\tau$.

*Proof.* Recall Equation (76), we can get something similar to Equation (83).

$$P_{\text{out}[0|t]} \leq 2 \frac{P_{\text{out}} \exp\left(-\frac{9D_\tau^2}{32t}\right)}{P_0(\|x - x^*\| \leq \frac{1}{4}D_\tau)\exp\left(-\frac{D_\tau^2}{8}\right)} \tag{107}$$

As in the theorem statement Equation (20), while $\alpha, \beta$ are still constants. $P(\|x - x^*\| \leq \frac{1}{4}D_\tau) \geq 1/2P_{\text{in}}$ holds trivially when $d$ is large. Then

$$P_{\text{out}[0|t]} \leq 4P_{\text{out}} \exp\left(-\frac{5D_\tau^2}{32t}\right)$$

The sufficient condition for the convex radius now becomes

$$4P_{\text{out}} \exp\left(-\frac{5D_\tau^2}{32t}\right) M_{\text{out}} \leq \frac{C_\alpha t^2}{t + \lambda/\alpha}$$

Moreover, with Equations (67) and (68), this leads to

$$48\max(D_\tau^2, D_s^2)P_{\text{out}} \leq \frac{(1 - C_\alpha)t^2}{t + \lambda/\alpha}\exp\left(\frac{D_\tau^2}{32t}\right)$$

Now we first focus on lower bound the RHS of the above inequality.

$$\mathcal{F}_{1.2}(t) := t\left(\frac{t}{t + \frac{\lambda}{\alpha}}\right)\exp\left(\frac{5D_\tau^2}{32\,t}\right), \qquad t > 0.$$

Write

$$\log \mathcal{F}_{1.2}(t) = 2 \log t - \log\left(t + \frac{\lambda}{\alpha}\right) + \frac{5D_\tau^2}{32\,t}.$$

Differentiate and set to zero:

$$\frac{d}{dt} \log \mathcal{F}_{1.2}(t) = \frac{2}{t} - \frac{1}{t + \frac{\lambda}{\alpha}} - \frac{5D_\tau^2}{32\,t^2} = 0.$$

Multiplying by $t^2\left(t + \frac{\lambda}{\alpha}\right)$ gives the quadratic

$$t^2 + \left(\frac{2\lambda}{\alpha} - \frac{5D_\tau^2}{32}\right) t - \frac{D_\tau^2}{32}\frac{\lambda}{\alpha} = 0,$$

whose unique positive root is

$$t^\star = \frac{1}{2}\left(\frac{5D_\tau^2}{32} - \frac{2\lambda}{\alpha} + \sqrt{\left(\frac{D_\tau^2}{32}\right)^2 + 4\left(\frac{\lambda}{\alpha}\right)^2}\right).$$

This $t^\star$ is the (global) minimizer of $\mathcal{F}_{1.1}(t)$ over $t > 0$.

Let

$$s := \sqrt{\left(\frac{5D_\tau^2}{32}\right)^2 + 4\left(\frac{\lambda}{\alpha}\right)^2}.$$

Then $t^\star + \frac{\lambda}{\alpha} = \frac{1}{2}\left(\frac{D_\tau^2}{32} + s\right)$, and

$$\mathcal{F}_{1.2}(t^\star) = \frac{\left(\frac{5D_\tau^2}{32} - \frac{2\lambda}{\alpha} + s\right)^2}{2\left(\frac{5D_\tau^2}{32} + s\right)} \exp\left(\frac{\frac{5D_\tau^2}{16}}{\frac{5D_\tau^2}{32} - \frac{2\lambda}{\alpha} + s}\right).$$

Given our assumption that $D_\tau^2 \sim \Theta(d)$, and $\lambda/\beta \le 2\kappa_0$, we have that

$$\frac{5D_\tau^2}{32} \gg \frac{\lambda}{\alpha}, \quad t^\star = \frac{5D_\tau^2}{32} - \frac{\lambda}{\alpha} + O\left(\frac{(\lambda/\alpha)^2}{D_\tau^2}\right),$$

and consequently

$$\mathcal{F}_{1.1}(t^\star) = e\left(\frac{5D_\tau^2}{32} - \frac{\lambda}{\alpha}\right) + O\left(\frac{(\lambda/\alpha)^2}{D_\tau^2}\right) \ge 0.42 D_\tau^2$$

Putting pieces together, we get the sufficient condition for the convex radius as follows

$$115 \max(D_\tau^2, D_s^2) P_{\text{out}} \le (1 - C_\alpha) D_\tau^2$$

Recall the sufficient condition for $P_{\text{out}}$ and $D_\tau^2$ as in theorem statement Theorem 4.3, this holds trivially for sufficiently large $d$. □

**Theorem B.7** (Local Condition Number Property)**.** *Let Assumption 4.2 and Assumption 4.1 hold. The local condition number inside the strongly convex region $\mathcal{R}_{SC}(t)$ is bounded that*

$$\kappa(t) \le \frac{\alpha t + \lambda}{\beta t + \lambda}\frac{1}{C_\alpha}\kappa_0 + \frac{t\lambda/\beta}{t + \lambda/\beta}\frac{1}{9D_\tau^2} \tag{108}$$

*Proof.* Recall that we have Equation (63) holds. Therefore,

$$P_{\text{out}[0|t]} \le \frac{1}{9\max(D_s^2, D_\tau^2)}\frac{(1 - C_\alpha)t^2}{t + \lambda/\alpha} \tag{109}$$

With the variance fraction, we get that

$$\text{Cov}_{[0|t]}[y \mid x_t] \geq P_{\text{in}[0|t]} \text{Cov}_{[0|t]}[y \mid x_t, y \in B_\tau] \geq (1 - P_{\text{out}[0|t]}) \frac{t\lambda/\beta}{t + \lambda/\beta}$$

Recall that Equation (63) holds. Therefore, we have that

$$P_{\text{out}[0|t]} M_{\text{out}} \leq \frac{(1 - C_\alpha)t^2}{t + \lambda/\alpha}$$

and given Equation (19)

$$\nabla^2 g(x; t) = \frac{\lambda}{t^2}(t - \text{Cov}_{[0|t]}[y \mid x_t])$$

The condition number can be bounded by

$$\kappa_t \leq \frac{t - (1 - P_{\text{out}[0|t]}) \frac{t\lambda/\beta}{t+\lambda/\beta}}{\frac{C_\alpha t^2}{t+\lambda/\alpha}} \leq \frac{\alpha t + \lambda}{\beta t + \lambda} \frac{1}{C_\alpha} \kappa_0 + \frac{t\lambda/\beta}{t + \lambda/\beta} \frac{1}{9 D_\tau^2}$$

$\square$

**Lemma B.8** (Variance Fraction). *The total variance of a random variable $X$ (scalar or vector) can be decomposed based on a partition of the sample space. Consider a partition into a set $\mathbb{A}$ and its complement $\mathbb{A}^{\complement}$. Let $P(\mathbb{A})$ be the probability that an outcome is in $\mathbb{A}$, and $P(\mathbb{A}^{\complement}) = 1 - P(\mathbb{A})$. Let $\mu_{\mathbb{A}} = \mathbb{E}[X | X \in \mathbb{A}]$ and $\mu_{\mathbb{A}^{\complement}} = \mathbb{E}[X | X \in \mathbb{A}^{\complement}]$. Then the decomposition is:*

$$\text{Var}(X) = \underbrace{P(\mathbb{A}) \text{Var}(X|X \in \mathbb{A}) + P(\mathbb{A}^{\complement}) \text{Var}(X|X \in \mathbb{A}^{\complement})}_{\text{Expected Conditional Variance (Variance within groups)}}$$

$$+ \underbrace{P(\mathbb{A})P(\mathbb{A}^{\complement})(\mu_{\mathbb{A}} - \mu_{\mathbb{A}^{\complement}})(\mu_{\mathbb{A}} - \mu_{\mathbb{A}^{\complement}})^T}_{\text{Variance of Conditional Expectations (Variance between groups)}} .$$

*This lemma is often used to break down the overall variability of $X$ into components attributable to variability within specified subgroups and variability between these subgroups.*

*Proof.* Let $p := P(\mathbb{A})$ so that $P(\mathbb{A}^{\complement}) = 1 - p$. Define the conditional means

$$\mu_{\mathbb{A}} := \mathbb{E}[X \mid X \in \mathbb{A}], \qquad \mu_{\mathbb{A}^{\complement}} := \mathbb{E}[X \mid X \in \mathbb{A}^{\complement}].$$

By the law of total expectation,

$$\mathbb{E}[X] = p\,\mu_{\mathbb{A}} + (1 - p)\,\mu_{\mathbb{A}^{\complement}}.$$

Next, decompose the second moment via the same partition:

$$\mathbb{E}[XX^T] = p\,\mathbb{E}[XX^T \mid X \in \mathbb{A}] + (1 - p)\,\mathbb{E}[XX^T \mid X \in \mathbb{A}^{\complement}].$$

Within each conditional expectation insert

$$\text{Var}(X \mid X \in \mathbb{A}) = \mathbb{E}[XX^T \mid X \in \mathbb{A}] - \mu_{\mathbb{A}}\mu_{\mathbb{A}}^T,$$

and its analogue for $\mathbb{A}^{\complement}$, to find

$$\mathbb{E}[XX^T] = p\left(\text{Var}(X \mid X \in \mathbb{A}) + \mu_{\mathbb{A}}\mu_{\mathbb{A}}^T\right) + (1 - p)\left(\text{Var}(X \mid X \in \mathbb{A}^{\complement}) + \mu_{\mathbb{A}^{\complement}}\mu_{\mathbb{A}^{\complement}}^T\right).$$

Subtracting $\mathbb{E}[X]\,\mathbb{E}[X]^T$ yields

$$\text{Var}(X) = p\,\text{Var}(X \mid X \in \mathbb{A}) + (1 - p)\,\text{Var}(X \mid X \in \mathbb{A}^{\complement}) + p(1 - p)\left(\mu_{\mathbb{A}} - \mu_{\mathbb{A}^{\complement}}\right)\left(\mu_{\mathbb{A}} - \mu_{\mathbb{A}^{\complement}}\right)^T.$$

Recalling $p = P(\mathbb{A})$ completes the decomposition. $\square$

**Lemma B.9** (Integrals of a product of two Gaussians truncated above $a$)**.** *Let $\phi, \Phi$ denote the standard normal pdf and CDF. Fix $\tau^2 > 0$, $t > 0$, $\mu \in \mathbb{R}$, and $a \in \mathbb{R}$. Define*

$$\eta = \frac{\tau^2 \mu}{\tau^2 + t}, \qquad \sigma^2 = \frac{\tau^2 t}{\tau^2 + t}, \qquad k = \frac{1}{\sqrt{2\pi(\tau^2 + t)}} \exp\left(-\frac{\mu^2}{2(\tau^2 + t)}\right),$$

*and the standardized cutoff $z := (a - \eta)/\sigma$. Set*

$$\Phi_\mu := \eta + \sigma \frac{\phi(z)}{1 - \Phi(z)}, \qquad \Phi_\sigma := \sigma^2 \left[ 1 + z \frac{\phi(z)}{1 - \Phi(z)} - \left(\frac{\phi(z)}{1 - \Phi(z)}\right)^2 \right].$$

*Then*

$$\text{(i)} \quad \int_a^\infty N(x \mid 0, \tau^2) \, N(x \mid \mu, t) \, dx = k \, [1 - \Phi(z)],$$

$$\text{(ii)} \quad \int_a^\infty x \, N(x \mid 0, \tau^2) \, N(x \mid \mu, t) \, dx = k \, [1 - \Phi(z)] \, \Phi_\mu,$$

$$\text{(iii)} \quad \int_a^\infty x^2 \, N(x \mid 0, \tau^2) \, N(x \mid \mu, t) \, dx = k \, [1 - \Phi(z)] \, \left[ \Phi_\sigma + (\Phi_\mu)^2 \right].$$

*Proof.* A standard completion-of-squares argument gives a Gaussian-in-$x$ representation for the product:

$$N(x \mid 0, \tau^2) \, N(x \mid \mu, t) = k \, N(x \mid \eta, \sigma^2), \tag{110}$$

with $\eta, \sigma^2, k$ as stated. Consequently, for any integrable test function $h$,

$$\int_a^\infty h(x) \, N(x \mid 0, \tau^2) \, N(x \mid \mu, t) \, dx = k \int_a^\infty h(x) \, N(x \mid \eta, \sigma^2) \, dx. \tag{111}$$

Let $X \sim N(\eta, \sigma^2)$ and write $z = (a - \eta)/\sigma$. Then

$$\int_a^\infty N(x \mid \eta, \sigma^2) \, dx = \Pr(X > a) = 1 - \Phi(z),$$

which combined with (111) for $h \equiv 1$ yields (i).

For the first moment, using $x = \eta + \sigma u$ with $u = (x - \eta)/\sigma$,

$$\int_a^\infty x \, N(x \mid \eta, \sigma^2) \, dx = \eta \int_a^\infty N(x \mid \eta, \sigma^2) \, dx + \sigma \int_z^\infty u \, \phi(u) \, du.$$

Since $\int_z^\infty u \, \phi(u) \, du = \phi(z)$, we obtain

$$\int_a^\infty x \, N(x \mid \eta, \sigma^2) \, dx = \eta \, [1 - \Phi(z)] + \sigma \, \phi(z) = [1 - \Phi(z)] \left( \eta + \sigma \frac{\phi(z)}{1 - \Phi(z)} \right) = [1 - \Phi(z)] \, \Phi_\mu.$$

Plugging this into (111) with $h(x) = x$ gives (ii).

For the second moment, similarly expand $x^2 = (\eta + \sigma u)^2 = \eta^2 + 2\eta\sigma u + \sigma^2 u^2$ to get

$$\int_a^\infty x^2 \, N(x \mid \eta, \sigma^2) \, dx = \eta^2 [1 - \Phi(z)] + 2\eta\sigma \int_z^\infty u \, \phi(u) \, du + \sigma^2 \int_z^\infty u^2 \phi(u) \, du.$$

We already have $\int_z^\infty u \, \phi(u) \, du = \phi(z)$, and integration by parts gives $\int_z^\infty u^2 \phi(u) \, du = z \, \phi(z) + [1 - \Phi(z)]$. Hence

$$\int_a^\infty x^2 \, N(x \mid \eta, \sigma^2) \, dx = (\eta^2 + \sigma^2)[1 - \Phi(z)] + \sigma\phi(z) \, (2\eta + \sigma z).$$

It is convenient to express this in terms of the truncated-normal mean and variance. With $\lambda(z) := \phi(z)/(1 - \Phi(z))$, we have $\Phi_\mu = \eta + \sigma\lambda(z)$ and $\Phi_\sigma = \sigma^2 \big(1 + z\lambda(z) - \lambda(z)^2\big)$, so that

$$\Phi_\sigma + (\Phi_\mu)^2 = \eta^2 + \sigma^2 + 2\eta\sigma\lambda(z) + \sigma^2 z\lambda(z).$$

Multiplying by $1 - \Phi(z)$ yields exactly the previous expression for $\int_a^\infty x^2 N(x \mid \eta, \sigma^2) \, dx$. Substituting into (111) with $h(x) = x^2$ gives (iii). $\qquad\square$

## C  Convergence Analysis

### C.1  Zeroth-Order Gradient Bounds Proof

**Theorem C.1** (Gradient Estimator Bounds)**.** *Given a zeroth-order gradient estimator Equation* (6)*, its bias and variance are bounded as:*

$$\mathbb{E}[\|\nabla_x g^{(0)}(x;t) - \nabla_x g(x;t)\|_2] \leq \frac{\lambda d}{N}\left(M_{-\frac{1}{2}} t^{-\frac{1}{2}} + M_0 \frac{L}{\lambda} + M_{\frac{1}{2}}\left(\frac{L}{\lambda}\right)^2 t^{\frac{1}{2}}\right) \tag{112}$$

$$\mathbb{E}[\|\nabla_x g^{(0)}(x;t) - \nabla_x g(x;t)\|_2^2] \leq \frac{\lambda^2 d}{N}\left(V_{-1} t^{-1} + V_0\left(\frac{L}{\lambda}\right)^2 + V_1\left(\frac{L}{\lambda}\right)^4 t\right) \tag{113}$$

*where $p \in (1, +\infty)$ and $M_{-\frac{1}{2}}, M_0, M_{\frac{1}{2}}, V_{-1}, V_0, V_1$ are positive constants that are independent of $N, t, \lambda, L$, and of the dimension $d$.*

**Lemma C.2** (Moment Bounds for Lipschitz Functions)**.** *Let $x \sim \mathcal{N}(0, t)$ be a random variable and $f : \mathbb{R} \to \mathbb{R}$ be an L-Lipschitz function. Then the following bounds hold for the raw moments and central moments:*

$$\mathbb{E}[f(x)^{2n}] \leq C_n^{(f)} L^{2n} t^n \tag{114}$$

$$m_{2n}[f(x)] = \mathbb{E}[|f(x) - \mathbb{E}[f(x)]|^{2n}] \leq C_n^{(f)} L^{2n} t^n \tag{115}$$

$$\mathbb{E}[(xf(x))^{2n}] \leq C_n^{(xf)} t^n + C_{2n}^{(xf)} L^{2n} t^{2n} \tag{116}$$

$$m_{2n}[xf(x)] = \mathbb{E}[|xf(x) - \mathbb{E}[xf(x)]|^{2n}] \leq C_n^{(xf)} t^n + C_{2n}^{(xf)} L^{2n} t^{2n} \tag{117}$$

*where $C_n^{(f)}$ and $C_n^{(xf)}$ are constants depending only on $n$.*

*Proof.* We begin with the triangle inequality: for any $x, y \in \mathbb{R}$ and $n \in \mathbb{N}$,

$$(x + y)^n \leq 2^{n-1}(x^n + y^n) \tag{118}$$

Applying this to the central moment of $xf(x)$:

$$\mathbb{E}[|xf(x) - \mathbb{E}[xf(x)]|^{2n}] \leq 2^{2n-1}(\mathbb{E}[|xf(x)|^{2n}] + |\mathbb{E}[xf(x)]|^{2n}) \tag{119}$$

For the first term, we use the fact that $f$ is $L$-Lipschitz, which means $|f(x) - f(0)| \leq L|x|$. This implies:

$$|f(x)| \leq |f(0)| + L|x| \tag{120}$$

$$|xf(x)| \leq |x| \cdot |f(x)| \leq |x| \cdot (|f(0)| + L|x|) = |f(0)||x| + Lx^2 \tag{121}$$

Therefore:

$$(xf(x))^{2n} \leq 2^{2n-1}(|f(0)|^{2n}|x|^{2n} + L^{2n}x^{4n}) \tag{122}$$

$$\mathbb{E}[|xf(x)|^{2n}] \leq 2^{2n-1}(|f(0)|^{2n}\mathbb{E}[|x|^{2n}] + L^{2n}\mathbb{E}[x^{4n}]) \tag{123}$$

Since $x \sim \mathcal{N}(0, t)$, we know that $\mathbb{E}[|x|^{2n}] = C_n t^n$ and $\mathbb{E}[x^{4n}] = C_{2n} t^{2n}$ for some constants $C_n, C_{2n}$ depending only on $n$. Thus:

$$\mathbb{E}[|xf(x)|^{2n}] \leq 2^{2n-1}(|f(0)|^{2n}C_n t^n + L^{2n}C_{2n} t^{2n}) \tag{124}$$

$$= C_n' t^n + C_{2n}' L^{2n} t^{2n} \tag{125}$$

where we've absorbed the constants into new constants $C_n'$ and $C_{2n}'$.

For the second term, we have:

$$\mathbb{E}[|xf(x)|] \le \mathbb{E}[|f(0)||x|] + \mathbb{E}[Lx^2] \tag{126}$$

$$= |f(0)|\mathbb{E}[|x|] + L\mathbb{E}[x^2] \tag{127}$$

$$= |f(0)|K_{\frac{1}{2}}t^{\frac{1}{2}} + L \cdot K_1 t \tag{128}$$

where $K_{\frac{1}{2}} = \sqrt{\frac{2}{\pi}}$ and $K_1 = 1$ for the standard normal distribution scaled by $\sqrt{t}$.

Raising this to the power of $2n$:

$$\mathbb{E}[|xf(x)|]^{2n} \le 2^{2n-1}((|f(0)|K_{\frac{1}{2}}t^{\frac{1}{2}})^{2n} + (L \cdot K_1 t)^{2n}) \tag{129}$$

$$= 2^{2n-1}(|f(0)|^{2n}K_{\frac{1}{2}}^{2n}t^n + L^{2n}K_1^{2n}t^{2n}) \tag{130}$$

$$= C_n^{(2)}t^n + C_{2n}^{(2)}L^{2n}t^{2n} \tag{131}$$

Therefore:

$$\mathbb{E}[|xf(x) - \mathbb{E}[xf(x)]|^{2n}] \le 2^{2n-1}(\mathbb{E}[|xf(x)|^{2n}] + |\mathbb{E}[xf(x)]|^{2n}) \tag{132}$$

$$\le 2^{2n-1}((C_n^{(1)}t^n + C_{2n}^{(1)}L^{2n}t^{2n}) + (C_n^{(2)}t^n + C_{2n}^{(2)}L^{2n}t^{2n})) \tag{133}$$

$$= 2^{2n-1}((C_n^{(1)} + C_n^{(2)})t^n + (C_{2n}^{(1)} + C_{2n}^{(2)})L^{2n}t^{2n}) \tag{134}$$

$$= C_n^{(xf)}t^n + C_{2n}^{(xf)}L^{2n}t^{2n} \tag{135}$$

where we've combined all constants into final constants $C_n^{(xf)}$ and $C_{2n}^{(xf)}$ as stated in equation (117).

The proofs for the other bounds follow similar reasoning, applying the Lipschitz property of $f$ and the moment properties of the normal distribution. $\qquad\square$

**Theorem C.1** (Gradient Estimator Bounds). *Given a zeroth-order gradient estimator Equation* (6), *its bias and variance are bounded as:*

$$\mathbb{E}[\|\nabla_x g^{(0)}(x;t) - \nabla_x g(x;t)\|_2] \le \frac{\lambda d}{N}\left(M_{-\frac{1}{2}}t^{-\frac{1}{2}} + M_0\frac{L}{\lambda} + M_{\frac{1}{2}}\left(\frac{L}{\lambda}\right)^2 t^{\frac{1}{2}}\right) \tag{112}$$

$$\mathbb{E}[\|\nabla_x g^{(0)}(x;t) - \nabla_x g(x;t)\|_2^2] \le \frac{\lambda^2 d}{N}\left(V_{-1}t^{-1} + V_0\left(\frac{L}{\lambda}\right)^2 + V_1\left(\frac{L}{\lambda}\right)^4 t\right) \tag{113}$$

*where $p \in (1, +\infty)$ and $M_{-\frac{1}{2}}, M_0, M_{\frac{1}{2}}, V_{-1}, V_0, V_1$ are positive constants that are independent of $N, t, \lambda, L$, and of the dimension $d$.*

*Proof of Theorem C.1.* We apply the non-asymptotic moment bound of self-normalized importance sampling estimators Agapiou et al. (2017). When the following quantity is bounded:

$$\mathcal{C}_{\text{MSE}} := \frac{3}{\pi(g)^2}m_2[\phi g] + \frac{3}{\pi(g)^4}\pi(|\phi g|^{2r})^{\frac{1}{r}}\Gamma_{2s}^{\frac{1}{s}}m_{2s}[\phi g]^{\frac{1}{s}} \tag{136}$$

$$+ \frac{3}{\pi(g)^{2(1+\frac{1}{p})}}\pi(|\phi|^{2p})^{\frac{1}{p}}\Gamma_{2q(1+\frac{1}{p})}^{\frac{1}{q}}m_{2q(1+\frac{1}{p})}[g]^{\frac{1}{q}} \tag{137}$$

where the constants $\Gamma_t > 0, t \ge 2$, satisfy $\Gamma_t^{\frac{1}{t}} \le t - 1$ and the two pairs of parameters $r, s$, and $p, q$ are conjugate pairs of indices satisfying $r, s, p, q \in (1, \infty)$ and $r^{-1} + s^{-1} = 1$, $p^{-1} + q^{-1} = 1$.

The bias and MSE of the importance sampling estimator are bounded by:

$$\left|\mathbb{E}\left[\mu^N(\phi) - \mu(\phi)\right]\right| \le \frac{1}{N}\left(\frac{2}{\pi(g)^2}m_2[g]^{\frac{1}{2}}m_2[\phi g]^{\frac{1}{2}} + 2\mathcal{C}_{\text{MSE}}^{\frac{1}{2}}\frac{\pi(g^2)^{\frac{1}{2}}}{\pi(g)}\right), \tag{138}$$

$$\mathbb{E}\left[\left(\mu^N(\phi) - \mu(\phi)\right)^2\right] \le \frac{1}{N}\mathcal{C}_{\text{MSE}} \tag{139}$$

In our case, the test function is $\phi(x) = x$ and the target function is $g(x) = \exp\left(-\frac{J(x)}{\lambda}\right)$. Given that $J(x)$ is $L$-Lipschitz, the function $g$ is $\frac{L}{\lambda}$-Lipschitz.

Note that $\pi(g)$ is the normalizer, which is constant up to a multiplicative factor. Thus, we focus on the moment terms like $m_2[\phi g]$, $m_2[g]$, $\pi(|\phi g|^{2d})$, and $\pi(|g|^{2q})$.

For the first term in $\mathcal{C}_{\mathrm{MSE}}$, applying the central moment lemma from Lemma C.2:

$$m_2[\phi g] \le A_1 t d + A_2 \left(\frac{L}{\lambda}\right)^2 t^2 d \tag{140}$$

For the second term in $\mathcal{C}_{\mathrm{MSE}}$:

$$\pi(|\phi g|^{2r})^{\frac{1}{r}} \le \left(B_r t^r d^r + B_{2r}\left(\frac{L}{\lambda}\right)^{2r} t^{2r} d^r\right)^{\frac{1}{r}} \tag{141}$$

$$\le \left(B_1 t + B_2 \left(\frac{L}{\lambda}\right)^2 t^2\right)d \tag{142}$$

$$m_{2e}[g]^{\frac{1}{e}} \le B_3 \left(\frac{L}{\lambda}\right)^2 t d \tag{143}$$

For the third term in $\mathcal{C}_{\mathrm{MSE}}$:

$$\pi(|\phi|^{2p})^{\frac{1}{p}} \le S_1 t d \tag{144}$$

$$m_{2q(1+\frac{1}{p})}[g]^{\frac{1}{q}} \le S_{1+\frac{1}{p}} \left(\frac{L}{\lambda}\right)^{2+\frac{2}{p}} t^{1+\frac{1}{p}} d \tag{145}$$

Combining these results, we get (for the estimation of $\mathbb{E}_{y \sim p_{0|t}}[y|x]$):

$$\left|\mathbb{E}\left[\mu^N(\phi) - \mu(\phi)\right]\right| \le \frac{d}{N}\left(E_{\frac{1}{2}}t^{\frac{1}{2}} + E_1 \frac{L}{\lambda}t + E_{1+\frac{1}{2p}}\left(\frac{L}{\lambda}\right)^{1+\frac{1}{p}}t^{1+\frac{1}{2p}} + E_{\frac{3}{2}}\left(\frac{L}{\lambda}\right)^2 t^{\frac{3}{2}}\right) \tag{146}$$

$$\mathbb{E}\left[\left(\mu^N(\phi) - \mu(\phi)\right)^2\right] \le \frac{d}{N}\left(F_1 t + F_2\left(\frac{L}{\lambda}\right)^2 t^2 + F_{2+\frac{1}{p}}\left(\frac{L}{\lambda}\right)^{2+\frac{2}{p}}t^{2+\frac{1}{p}} + F_3\left(\frac{L}{\lambda}\right)^4 t^3\right) \tag{147}$$

Note that

$$\mathbb{E}[|\nabla_x g^{(0)}(x;t) - \nabla_x g(x;t)|] = \frac{\lambda}{t}\mathbb{E}[|\mu^N(\phi) - \mu(\phi)|] \tag{148}$$

$$\mathbb{E}[|\nabla_x g^{(0)}(x;t) - \nabla_x g(x;t)|^2] = \frac{\lambda^2}{t^2}\mathbb{E}[|\mu^N(\phi) - \mu(\phi)|^2] \tag{149}$$

Substituting these relationships from equations (146) and (147) into (148) and (149), and simplifying the exponents of $t$, we arrive at the final result as stated in equations (112) and (113):

$$\mathbb{E}[|\nabla_x g^{(0)}(x;t) - \nabla_x g(x;t)|] \le \frac{\lambda d}{N}\left(M_{-\frac{1}{2}}t^{-\frac{1}{2}} + M_0\frac{L}{\lambda} + M_{\frac{1}{2p}}\left(\frac{L}{\lambda}\right)^{1+\frac{1}{p}}t^{\frac{1}{2p}} + M_{\frac{1}{2}}\left(\frac{L}{\lambda}\right)^2 t^{\frac{1}{2}}\right) \tag{150}$$

$$\mathbb{E}[|\nabla_x g^{(0)}(x;t) - \nabla_x g(x;t)|^2] \le \frac{\lambda^2 d}{N}\left(V_{-1}t^{-1} + V_0\left(\frac{L}{\lambda}\right)^2 + V_{\frac{1}{p}}\left(\frac{L}{\lambda}\right)^{2+\frac{2}{p}}t^{\frac{1}{p}} + V_1\left(\frac{L}{\lambda}\right)^4 t\right) \tag{151}$$

where $M_{-\frac{1}{2}}, M_0, M_{\frac{1}{2}}, M_{\frac{1}{2p}}, M_{\frac{1}{2}}, V_{-1}, V_0, V_1, V_{\frac{1}{p}}$ are positive constants.

Given that $p \in (1, +\infty)$, under worst case, $p \to 1$, plugging in $p = 1$ into the above bound, term $M_{\frac{1}{2p}}$ and $V_{\frac{1}{p}}$ can be merged into $M_{\frac{1}{2}}$ and $V_1$ respectively.

$$\mathbb{E}[|\nabla_x g^{(0)}(x;t) - \nabla_x g(x;t)|] \leq \frac{\lambda d}{N} \left( M_{-\frac{1}{2}} t^{-\frac{1}{2}} + M_0 \frac{L}{\lambda} + M_{\frac{1}{2}} \left(\frac{L}{\lambda}\right)^2 t^{\frac{1}{2}} \right) \tag{152}$$

$$\mathbb{E}[|\nabla_x g^{(0)}(x;t) - \nabla_x g(x;t)|^2] \leq \frac{\lambda^2 d}{N} \left( V_{-1} t^{-1} + V_0 \left(\frac{L}{\lambda}\right)^2 + V_1 \left(\frac{L}{\lambda}\right)^4 t \right) \tag{153}$$

$\square$

## C.2 Single-Stage Convergence with Fixed Smoothing Parameter

**Theorem C.3** (Convergence of SGD with bounded bias and variance). *Let $f : \mathbb{R}^d \to \mathbb{R}$ be an $\alpha$-strongly convex and $\beta$-smooth function. Let the step size satisfy $\eta \leq \frac{\alpha}{4\beta^2}$. Assume a gradient estimator $\nabla \hat{f}(x_k) = \nabla f(x_k) + b_k + w_k$ with bounded bias $\|b_k\| \leq K$ and bounded variance $\mathbb{E}[\|w_k\|^2] \leq \sigma^2$. Then, with probability at least $(1 - \delta)$, the error satisfies*

$$\|x_k - x^*\|^2 \leq (1 - \frac{\eta\alpha}{2})^k \|x_0 - x^*\|^2 + (\frac{1}{\alpha} + 2\eta) \left( \frac{1}{\delta} \frac{4(K^2 + \sigma^2)}{\alpha} \right)$$

*Proof.* Let $\Delta_k = x_k - x^*$. Then,

$$\begin{aligned}
\|\Delta_{k+1}\|^2 &= \|\Delta_k - \eta \nabla f(x_k) - \eta b_k - \eta w_k\|^2 \\
&= \|\Delta_k\|^2 - 2\eta\langle\Delta_k, \nabla f(x_k)\rangle - 2\eta\langle\Delta_k, b_k + w_k\rangle + \eta^2\|\nabla f(x_k) + b_k + w_k\|^2.
\end{aligned}$$

For the gradient term, using $\alpha$-strong convexity:

$$-2\eta\langle\Delta_k, \nabla f(x_k)\rangle \leq -2\eta\alpha\|\Delta_k\|^2.$$

For bias and noise term, first using Cauchy-Schwarz inequality:

$$-2\eta\langle\Delta_k, b_k + w_k\rangle \leq 2\eta\|\Delta_k\|\|b_k + w_k\|.$$

Then using Young's inequality ($2ab \leq \epsilon a^2 + b^2/\epsilon$, let $\epsilon = \alpha$):

$$2\eta\|\Delta_k\|\|b_k + w_k\| \leq \eta\alpha\|\Delta_k\|^2 + \frac{\eta}{\alpha}\|b_k + w_k\|^2.$$

For the quadratic term, using $(a + b)^2 \leq 2a^2 + 2b^2$ and $\beta$-smoothness $\|\nabla f(x_k)\| \leq \beta\|\Delta_k\|$:

$$\eta^2\|\nabla f(x_k) + b_k + w_k\|^2 \leq 2\eta^2\beta^2\|\Delta_k\|^2 + 2\eta^2\|b_k + w_k\|^2.$$

Organize the terms, we get recurrence:

$$\|\Delta_{k+1}\|^2 \leq (1 - 2\eta\alpha + \eta\alpha + 2\eta^2\beta^2)\|\Delta_k\|^2 + (2\eta^2 + \frac{\eta}{\alpha})\|b_k + w_k\|^2.$$

For contraction coefficient $\rho$, due to $2\eta\beta^2 \leq \frac{\alpha}{2}$, we have:

$$\rho = 1 - 2\eta\alpha + \eta\alpha + 2\eta^2\beta^2 \leq 1 - \frac{\eta\alpha}{2}.$$

The recurrence is now:

$$\|\Delta_{k+1}\|^2 \le \rho \|\Delta_k\|^2 + (\frac{\eta}{\alpha} + 2\eta^2)\|b_k + w_k\|^2.$$

Unrolling the recurrence from $k$ down to 0, we get:

$$\|\Delta_k\|^2 \le \rho^k \|\Delta_0\|^2 + (\frac{\eta}{\alpha} + 2\eta^2) \sum_{i=0}^{k-1} \rho^{k-1-i} \|b_i + w_i\|^2.$$

For the bias and variance term, its expected value is bounded by:

$$\mathbb{E}[\|b_i + w_i\|^2] \le \mathbb{E}[2\|b_i\|^2 + 2\|w_i\|^2] \le 2\mathbb{E}[\|b_i\|^2] + 2\mathbb{E}[\|w_i\|^2] \le 2K^2 + 2\sigma^2.$$

Then the expected value of the sum $Z = \sum_{i=0}^{k-1} \rho^{k-1-i} \|b_i + w_i\|^2$ is bounded by:

$$\begin{aligned}
\mathbb{E}[Z] &= \sum_{i=0}^{k-1} \rho^{k-1-i} \mathbb{E}[\|b_i + w_i\|^2] \\
&\le \sum_{i=0}^{\infty} \rho^i (2K^2 + 2\sigma^2) \\
&= \frac{2(K^2 + \sigma^2)}{1 - \rho} \\
&= \frac{2(K^2 + \sigma^2)}{\eta\alpha/2} = \frac{4(K^2 + \sigma^2)}{\eta\alpha}.
\end{aligned}$$

With Markov's inequality, we have:

$$P(Z \ge \epsilon) \le \frac{\mathbb{E}[Z]}{\epsilon} = \frac{4(K^2 + \sigma^2)}{\eta\alpha\epsilon}.$$

With probability at least $(1 - \delta)$, we have:

$$Z \le \frac{1}{\delta} \frac{4(K^2 + \sigma^2)}{\eta\alpha}.$$

Substitute the bound for $Z$ back into the unrolled equation, we get:

$$\|\Delta_k\|^2 \le \rho^k \|\Delta_0\|^2 + \frac{1}{\delta}(\frac{1}{\alpha} + 2\eta)\frac{4(K^2 + \sigma^2)}{\alpha}.$$

$$\square$$

**Theorem 4.5** (Convergence of Sampling without Annealing over $t$). *For any fixed smoothing level $t > 0$, the function $g(x; t)$ is $\alpha_t$-strongly convex, $\beta_t$-smooth, $L_t$-Lipschitz, and has condition number $\kappa_t = \frac{\beta_t}{\alpha_t}$. The step size is $\eta = \frac{\alpha_t}{4\beta_t^2}$. If initial iterate lies in original convex region: $x_0 \in \mathcal{R}_{SC}(t)$ (Theorem 4.3), with probability at least $(1 - \delta)$, the error satisfies:*

$$\|x_k - x^*\|^2 \le \|x_t^* - x^*\|^2 + (1 - \frac{1}{4\kappa_t^2})^k \|x_0 - x^*\|^2 + \frac{4(K_t^2 + \sigma_t^2)}{\delta} \frac{t}{\alpha_t}(\frac{t}{2\lambda} + \frac{1}{\alpha_t})$$

*where $\|x_t^* - x^*\|^2$ follows Theorem 4.4, $K_t = \frac{\lambda}{N}\left(M_{-\frac{1}{2}} t^{-\frac{1}{2}} + M_0 \frac{L_t}{\lambda} + M_{\frac{1}{2}}\left(\frac{L_t}{\lambda}\right)^2 t^{\frac{1}{2}}\right)$ and $\sigma_t^2 = \frac{\lambda^2}{N}\left(V_{-1} t^{-1} + V_0\left(\frac{L_t}{\lambda}\right)^2 + V_1\left(\frac{L_t}{\lambda}\right)^4 t\right)$, where $M_{-1/2}, M_0, M_{1/2}$ and $V_{-1}, V_0, V_1$ are the dimensional-independent parameters come from former gradient estimator bounds in Theorem C.1.*

*Proof.* Plugging in gradient estimator bounds and step size $\eta = \frac{\alpha}{4\beta^2}$, we get:

$$\|x_k - x_k^*\|^2 \leq (1 - \frac{1}{4\kappa_t^2})^k \|x_0 - x^*\|^2 + \frac{4(K_t^2 + \sigma_t^2)}{\delta \, \alpha_t}(\frac{t}{2\lambda} + \frac{1}{\alpha_t})$$

Use triangle inequality, we have:

$$\|x_k - x^*\|^2 \leq \|x_k - x_k^*\|^2 + \|x_k^* - x^*\|^2$$

Plugging in the bound for $\|x_k - x_k^*\|^2$, we get:

$$\|x_k - x^*\|^2 \leq (1 - \frac{1}{4\kappa_t^2})^k \|x_0 - x^*\|^2 + \frac{4(K_t^2 + \sigma_t^2)}{\delta \, \alpha_t}(\frac{t}{2\lambda} + \frac{1}{\alpha_t}) + \|x_k^* - x^*\|^2$$

$\square$

## C.3 Multi-Stage Convergence with Varying Smoothing Parameter

**Theorem C.4** (Global Convergence of Multi-Stage Algorithm). *Let $\{g(x;t)\}_{t\geq0}$ be a family of objective functions with minimizer $x_t^*$ such that $\|x_t^* - x^*\|^2 \leq k_b t$. Assume gradient estimator error $e(x;t) = b(x;t) + w(x;t)$, with bias expectation $\mathbb{E}[\|b(x;t)\|] \leq K_m$ and variance $\mathbb{E}[\|w(x;t)\|^2] \leq \sigma_m^2$. Let $\rho_m = 1 - \frac{\alpha_m^2}{8\beta_m^2}$. Define the effective contraction $\tilde{\rho}_m = (1 + \frac{1-\rho_m}{4})^2 \rho_m < 1$. With the following feasible condition for the noise and bias:*

$$\frac{2M}{\delta}K_m^2 + \sigma_m^2 \leq \frac{2E_0 r_{min}\beta_1^2\delta}{3M} \tag{154}$$

$$\frac{k_g}{k_b} \leq K_0 \tag{155}$$

*where $\rho'_m = \tilde{\rho}_m + \frac{\mathcal{P}_m k_b}{k_g}, \epsilon_m = \frac{1-\rho_m}{4}, \mathcal{P}_m = (1 + \epsilon_m)\rho_m(1 + \epsilon_m^{-1}) + (1 + \epsilon_m^{-1}), E_0 = \frac{512\,\kappa_0^4 + 56\,\kappa_0^2 + 1}{8192\,\kappa_0^6 + 256\,\kappa_0^4}, K_0 = \frac{\kappa_0^2(524288\kappa_0^6 + 57344\kappa_0^4 + 2304\kappa_0^2 + 32)}{512\kappa_0^4 + 56\kappa_0^2 + 1}, \kappa_0$ is the condition number of the initial objective function $f(x) = g(x; t = 0)$. Then if initial iterate $\|x_0 - x^*\|^2 \leq r_{min} + k_g t_0$, the sequence $\{x_m\}_{m=0}^M$ satisfies $\|x_m - x^*\|^2 \leq r_{min} + k_g t_m$ for all $m$ with probability $1 - \delta$.*

*Proof.* By the Union Bound over $M$ stages, it suffices to show that $\|x_{m+1} - x^*\|^2 \leq r_{\min} + k_g t_{m+1}$ given $\|x_m - x^*\|^2 \leq r_{\min} + k_g t_m$.

Let the error be $e_m = b_m + w_m$. By Markov's inequality:

$$\mathbb{P}(\|b_m\| \geq \frac{2M}{\delta}K_m) \leq \frac{\delta}{2M},$$

$$\mathbb{P}(\|w_m\|^2 \geq \frac{2M}{\delta}\sigma_m^2) \leq \frac{\delta}{2M}$$

By the Union Bound, with probability at least $1 - \frac{\delta}{M}$, both hold simultaneously, yielding

$$\|e_m\|^2 \leq 2\|b_m\|^2 + 2\|w_m\|^2 \leq \frac{8M^2}{\delta^2}K_m^2 + \frac{4M}{\delta}\sigma_m^2$$

Using Young's Inequality twice with $\epsilon_m = \frac{1-\rho_m}{4}$, with probability at least $1 - \frac{\delta}{M}$:

$$\|x_{m+1} - x^*\|^2 \leq (1 + \epsilon_m)\|x_{m+1} - x_m^*\|^2 + (1 + \epsilon_m^{-1})\|x_m^* - x^*\|^2$$
$$\leq (1 + \epsilon_m)\left[\rho_m\|x_m - x_m^*\|^2 + E_m\right] + (1 + \epsilon_m^{-1})\|x_m^* - x^*\|^2$$

where $E_m = \left(\frac{\eta_m}{\alpha_m} + 2\eta_m^2\right)\left(\frac{8M^2}{\delta^2}K_m^2 + \frac{4M}{\delta}\sigma_m^2\right)$. We expand $\|x_m - x_m^*\|^2 \leq (1+\epsilon_m)\|x_m - x^*\|^2 + (1+\epsilon_m^{-1})\|x^* - x_m^*\|^2$. Substituting this back and grouping terms:

$$\|x_{m+1} - x^*\|^2 \leq \underbrace{(1+\epsilon_m)^2\rho_m}_{\tilde{\rho}_m}\|x_m - x^*\|^2 + (1+\epsilon_m)E_m + \underbrace{\left[(1+\epsilon_m)\rho_m(1+\epsilon_m^{-1}) + (1+\epsilon_m^{-1})\right]}_{\mathcal{P}_m}\|x_m^* - x^*\|^2$$

To make sure the next stage still stays within $\|x_{m+1} - x^*\|^2 \leq r_{\min} + k_g t_{m+1}$, we require:

$$\tilde{\rho}_m(r_{\min} + k_g t_m) + (1+\epsilon_m)E_m + \mathcal{P}_m k_b t_m \leq r_{\min} + k_g t_{m+1}$$

Rearranging terms, we get:

$$t_{m+1} \geq \underbrace{(\tilde{\rho}_m + \frac{\mathcal{P}_m k_b}{k_g})t_m}_{\text{contraction factor}} - \underbrace{\frac{1}{k_g}(1-\tilde{\rho}_m)r_{\min}}_{\text{bias term}} + \underbrace{\frac{1}{k_g}(1+\epsilon_m)E_m}_{\text{noise term}}$$

To make sure $t_m$ is decreasing, let

$$(1+\epsilon_m)E_m - (1-\tilde{\rho}_m)r_{\min} \leq 0$$
$$\tilde{\rho}_m + \frac{\mathcal{P}_m k_b}{k_g} < 1$$

For the contraction factor, plugging in bounds for step size:

$$\rho_m = 1 - \frac{\alpha_m^2}{8\beta_m^2} = 1 - \frac{1}{8\kappa_m^2}$$

where $\kappa_m = \frac{\beta_m}{\alpha_m} \in [1, \kappa_0]$, where $\kappa_0$ is the condition number of the initial objective function. Given $\epsilon_m = \frac{1-\rho_m}{4}$, $\tilde{\rho}_m = (1+\epsilon_m)^2\rho_m = (1+\epsilon_m)^2(1-4\epsilon_m)$, $\mathcal{P}_m = (1+\epsilon_m)\rho_m(1+\epsilon_m^{-1}) + (1+\epsilon_m^{-1}) = 2\epsilon_m^{-1} + 9\epsilon_m + 4\epsilon_m^2 + 7$, the contraction factor is bounded by:

$$\frac{7}{8} \leq \rho_m \leq 1 - \frac{1}{8\kappa_0^2}$$
$$\frac{1}{32\kappa_0^2} \leq \epsilon_m \leq \frac{1}{32}$$
$$\frac{7623}{8192} \leq \tilde{\rho}_m \leq (1 + \frac{1}{32\kappa_0^2})^2(1 - \frac{4}{32\kappa_0^2})$$
$$\frac{18249}{256} \leq \mathcal{P}_m \leq 64\kappa_0^2 + 7 + \frac{9}{32\kappa_0^2} + \frac{1}{256\kappa_0^4}$$

For the contraction factor to be less than 1, we need the convex expansion factor to be greater than:

$$k_g > \max_m \frac{\mathcal{P}_m k_b}{1 - \tilde{\rho}_m} \geq \frac{\kappa_0^2(524288\kappa_0^6 + 57344\kappa_0^4 + 2304\kappa_0^2 + 32)k_b}{512\kappa_0^4 + 56\kappa_0^2 + 1}$$

For the bias term to be negative, we need:

$$E_m \leq \min_m \frac{(1-\tilde{\rho}_m)r_{\min}}{1+\epsilon_m} = \frac{512\,\kappa_0^4 + 56\,\kappa_0^2 + 1}{8192\,\kappa_0^6 + 256\,\kappa_0^4}r_{\min} = E_0 r_{\min}$$

Plugging $E_m$ back, we get:

$$\frac{8M^2}{\delta^2}K_m^2 + \frac{4M}{\delta}\sigma_m^2 \leq \min_m \frac{E_0 r_{\min}}{\frac{\eta_m}{\alpha_m} + 2\eta_m^2}$$

where $\frac{\eta_m}{\alpha_m} = \frac{1}{4\beta_m^2} \leq \frac{1}{4\beta_1^2}$, and $\beta_1$ is the smoothness of the smoothed objective function at $t_1$. Step size is bounded by $\eta_m = \frac{\alpha_m}{4\beta_m^2} = \frac{1}{4\kappa_m\beta_m} \leq \frac{1}{4\beta_1}$. Consequently, we can bound the step size factor: $\frac{\eta_m}{\alpha_m} + 2\eta_m^2 \leq \frac{1}{4\beta_1^2} + \frac{2}{16\beta_1^2} = \frac{3}{8\beta_1^2}$. Plugging this in, we get:

$$\frac{8M^2}{\delta^2}K_m^2 + \frac{4M}{\delta}\sigma_m^2 \leq \frac{8E_0 r_{\min}\beta_1^2}{3}$$
$$\frac{2M}{\delta}K_m^2 + \sigma_m^2 \leq \frac{2E_0 r_{\min}\beta_1^2\delta}{3M}$$

$\square$

**Theorem 4.6** (Conditional Global Convergence of Dual-Level Annealing Algorithm). *Given initial iterate lies in original convex region: $x_{t_0} \in \mathcal{R}_{SC}(t_0)$ (Theorem 4.3), consider the update rule from time $t_0$ to $t_F$ with the following geometric time schedule:*

$$t_{m+1} = \begin{cases} \gamma t_m, & m < M_0, \\ t_F, & m \geq M_0, \end{cases} \tag{9}$$

$$x_{t_{m+1}} = x_{t_m} - \eta\nabla\hat{g}(x_{t_m}; t_m) \tag{10}$$

*where $t_F$ is the final sampling kernel, $M_0 = \lceil \log_\gamma(\frac{t_0}{t_F}) \rceil$ is the iteration at which the annealing stops, the sampling temperature is set adaptively as $\lambda_m = \beta\sqrt{t_m}$, and step size is set as $\eta = \frac{\alpha_m}{4\beta_m^2}$. With adaptive annealing $\lambda_m = \beta\sqrt{t_m}$, the required sample size $N$ is bounded by:*

$$N = \max\left\{ \frac{3\beta^2 Md}{E_0 C_E^2 D_\tau^2 \beta_1^2 \delta}(V_{-1} + V_0 + V_1), \; \frac{2M}{\delta d}\frac{\left(M_{-\frac{1}{2}} + M_0\frac{L}{\beta} + M_{\frac{1}{2}}\frac{L^2}{\beta^2}\right)^2}{V_{-1} + V_0\frac{L^2}{\beta^2} + V_1\frac{L^4}{\beta^4}} \right\} \tag{11}$$

*where $V_{-1}, V_0, V_1$ and $M_{-1/2}, M_0, M_{1/2}$ are dimension-independent constants in the gradient estimator bounds from Theorem C.1. Without adaptive annealing $\lambda_m = \lambda_0$, let $\chi_0 := L/\lambda_0$. The required sample size $N$ is bounded by:*

$$N = \max\left\{ N_{acc}(t_0), N_{acc}(t_c), \max_m N_{bias}(t_m, \lambda_0) \right\}. \tag{12}$$

*Here $N_{acc}(t) = \frac{3\lambda_0^2 Md}{E_0 C_E^2 D_\tau^2 \beta_1^2 \delta}(V_{-1}t^{-1} + V_0\chi_0^2 + V_1\chi_0^4 t)$ and $N_{bias}(t) = \frac{2M}{\delta d}(M_{-1/2}t^{-1/2} + M_0\chi_0 + M_{1/2}\chi_0^2 t^{1/2})^2(V_{-1}t^{-1} + V_0\chi_0^2 + V_1\chi_0^4 t)^{-1}$. Then with probability at least $1 - \delta$, the dual-level annealing algorithm converges to*

$$\|x_M - x^*\|^2 \leq \|x_F^* - x^*\|^2 + \left(1 - \frac{1}{4\kappa_F^2}\right)^{M-M_0}(C_E^2 D_\tau^2 + k_g t_F) + \frac{4\left(\frac{2M}{\delta}K_F^2 + \sigma_F^2\right)}{\delta\,\alpha_F}\left(\frac{t_F}{2\lambda_F} + \frac{1}{\alpha_F}\right) \tag{13}$$

*where $k_g = C_E^2\min\{\frac{\beta^2}{\lambda^2}, \frac{1}{\tau^4}\}$.*

*Proof.* Consider the convex radius bound in Theorem 4.3:

$$\mathcal{R}_{SC}(t) := \left\{ x \in \mathbb{R}^d \;\middle|\; \|x - x^*\| \leq C_E\min\left(\sqrt{\frac{t + \frac{\lambda}{L}}{\frac{\lambda}{L}}}, \; \sqrt{\frac{t + \tau^2}{\tau^2}}\right) D_\tau \right\}$$

We can identify the minimum convex radius as $r_{\min} = C_E^2 D_\tau^2$ and the convex radius expansion speed as $k_g = C_E^2\min\{\frac{L^2}{\lambda^2}, \frac{1}{\tau^4}\}$. Consider the bias bound in Theorem 4.4:

$$\|x_t^* - x^*\| \leq \min\left\{ \frac{(1 - C_\alpha)t}{C_\alpha}\frac{1}{4D_\tau}, (D_\tau + \tau)\frac{t + \frac{\lambda}{\alpha}}{C_\alpha t} \right\} + \frac{\kappa - 1}{2C_\alpha}\sqrt{\frac{1}{2\pi(\frac{1}{t} + \frac{\alpha}{\lambda})}}.$$

Given $t \in [t_F, t_0]$, we have:

$$\|x_t^* - x^*\| \leq \frac{(1-C_\alpha)t}{4C_\alpha D_\tau} + \frac{\kappa-1}{2C_\alpha}\sqrt{\frac{1}{2\pi/t}} \qquad \text{(Select 1st term of min; drop } \tfrac{\alpha}{\lambda} \geq 0)$$

$$= \frac{1-C_\alpha}{4C_\alpha D_\tau}t + \frac{\kappa-1}{2C_\alpha\sqrt{2\pi}}\sqrt{t} \qquad \text{(Simplify)}$$

$$\leq \frac{1-C_\alpha}{4C_\alpha D_\tau}t + \frac{\kappa-1}{2C_\alpha\sqrt{2\pi}}\left(\frac{t}{\sqrt{t_F}}\right) \qquad \text{(Since } t \geq t_F \implies \sqrt{t} \leq \tfrac{t}{\sqrt{t_F}})$$

$$= \underbrace{\left(\frac{1-C_\alpha}{4C_\alpha D_\tau} + \frac{\kappa-1}{2C_\alpha\sqrt{2\pi t_F}}\right)}_{k_b}t \qquad \text{(Definition of } k_b)$$

Next, consider the gradient estimator bound in Theorem C.1: When $t_m < t_F$, the gradient estimator is dominated by the proposal sampling variance $t_m$. Because the bias scales as $K_m = \mathcal{O}(1/N)$ and the variance scales as $\sigma_m^2 = \mathcal{O}(1/N)$, the penalized squared bias term decays as $\mathcal{O}(1/N^2)$. To ensure the variance strictly dominates the penalized expected squared error, we enforce $\frac{2M}{\delta}K_m^2 \leq \sigma_m^2$. Substituting the gradient bounds, this requires a minimum sample size:

$$N \geq \frac{2M}{\delta d}\frac{\left(M_{-\frac{1}{2}}t_m^{-\frac{1}{2}} + M_0\frac{L}{\lambda_m} + M_{\frac{1}{2}}\left(\frac{L}{\lambda_m}\right)^2 t_m^{\frac{1}{2}}\right)^2}{V_{-1}t_m^{-1} + V_0\left(\frac{L}{\lambda_m}\right)^2 + V_1\left(\frac{L}{\lambda_m}\right)^4 t_m} := N_{\text{bias}}(t_m, \lambda_m)$$

Given $N \geq N_{\text{bias}}(t_m, \lambda_m)$, we can bound the total penalized error safely by twice the variance, bypassing the $1/\delta^2$ penalty in our final sample complexity:

$$\frac{2M}{\delta}K_m^2 + \sigma_m^2 \leq 2\sigma_m^2 \leq 2\frac{\lambda_m^2 d}{N}\left(V_{-1}t_m^{-1} + V_0\left(\frac{L}{\lambda_m}\right)^2 + V_1\left(\frac{L}{\lambda_m}\right)^4 t_m\right)$$

Let $D = 2\frac{\lambda_m^2 d}{N}\left(V_{-1}t_m^{-1} + V_0(\frac{L}{\lambda_m})^2 + V_1(\frac{L}{\lambda_m})^4 t_m\right)$. From the updated feasible condition, we require $D \leq \frac{2E_0 C_E^2 D_\tau^2 \beta_1^2 \delta}{3M}$. Solving for $N$, we get an accuracy bound:

$$N_{\text{acc}}(t_m, \lambda_m) = \frac{3\lambda_m^2 dM}{E_0 C_E^2 D_\tau^2 \beta_1^2 \delta}\left(V_{-1}t_m^{-1} + V_0\left(\frac{L}{\lambda_m}\right)^2 + V_1\left(\frac{L}{\lambda_m}\right)^4 t_m\right)$$

The required sample size must satisfy $N \geq \max\{N_{\text{acc}}(t_m, \lambda_m), N_{\text{bias}}(t_m, \lambda_m)\}$. When $\lambda_m = \lambda_0$, to make sure $N$ works for all $m$, we need to ensure:

$$N \geq \max\left\{\max_m N_{\text{acc}}(t_m, \lambda_0), \max_m N_{\text{bias}}(t_m, \lambda_0)\right\}$$

Where $\max_m N_{\text{acc}}(t_m, \lambda_0) = \max\{N_{\text{acc}}(t_0, \lambda_0), N_{\text{acc}}(t_c, \lambda_0)\}$. When adaptive annealing $\lambda_m = \beta\sqrt{t_m}$ is used, the time variable $t_m$ cancels out entirely in the bias dominance constraint, yielding a global constant:

$$N_{\text{bias}} = \frac{2M}{\delta d}\frac{\left(M_{-\frac{1}{2}} + M_0\frac{L}{\beta} + M_{\frac{1}{2}}\frac{L^2}{\beta^2}\right)^2}{V_{-1} + V_0\frac{L^2}{\beta^2} + V_1\frac{L^4}{\beta^4}}$$

And similarly, for the accuracy constraint, substituting $\lambda_m = \beta\sqrt{t_m}$, we have:

$$N_{\text{acc}} = \frac{3\beta^2 dM}{E_0 C_E^2 D_\tau^2 \beta_1^2 \delta}(V_{-1} + V_0 + V_1)$$

Thus, we have a uniform bound for $N$:

$$N \geq \max\{N_{\text{acc}}, N_{\text{bias}}\}$$

With the above $N$, the sequence $\{x_m\}_{m=0}^{M_0}$ satisfies $\|x_m - x^*\|^2 \leq r_{\min} + k_g t_m$ for all $m$ with probability $1 - \delta$, where $r_{\min} = C_E^2 D_\tau^2$ and $k_g = C_E^2 \min\{\frac{\beta^2}{\lambda^2}, \frac{1}{\tau^4}\}$. Once $t_m \leq t_F$, it will be fixed to $t_F < t_F$ to run local search by $M - M_0$ steps. Applying Theorem C.3 and our bounded error limits, we have:

$$\|x_M - x^*\|^2 \leq \|x_F^* - x^*\|^2 + \left(1 - \frac{1}{4\kappa_F^2}\right)^{M-M_0} (C_E^2 D_\tau^2 + k_g t_F) + \frac{4(\frac{2M}{\delta}K_F^2 + \sigma_F^2)}{\delta\,\alpha_F}\left(\frac{t_F}{2\lambda_F} + \frac{1}{\alpha_F}\right)$$

where $x_F^*$ is the minimizer of the final sampling kernel $t_F$, $K_F = \frac{\lambda_F}{N}\left(M_{-\frac{1}{2}}t_F^{-\frac{1}{2}} + M_0\frac{L_F}{\lambda_F} + M_{\frac{1}{2}}\left(\frac{L_F}{\lambda_F}\right)^2 t_F^{\frac{1}{2}}\right)$

and $\sigma_F^2 = \frac{\lambda_F^2}{N}\left(V_{-1}t_F^{-1} + V_0\left(\frac{L_F}{\lambda_F}\right)^2 + V_1\left(\frac{L_F}{\lambda_F}\right)^4 t_F\right)$. Finally, plugging in the convex radius - bias bound:

$$\frac{k_g}{k_b} \leq K_0$$

where $K_0 = \frac{\kappa_0^2(524288\kappa_0^6 + 57344\kappa_0^4 + 2304\kappa_0^2 + 32)}{512\kappa_0^4 + 56\kappa_0^2 + 1}$, $k_g = C_E^2 \min\{\frac{\beta^2}{\lambda^2}, \frac{1}{\tau^4}\}$ and $k_b = \frac{1-C_\alpha}{4C_\alpha D_\tau} + \frac{\kappa-1}{2C_\alpha\sqrt{2\pi t_F}}$. $\qquad\square$

## D  Experiment Details

To validate the empirical performance of the proposed DIDA, we compare it against the following sampling-based baselines (see also Table 1):

- **Model-Based Diffusion (MBD)** (Pan et al., 2024): anneals over $t$ only, with fixed $\lambda$.

- **Simulated Annealing (SA)** (Coleman et al., 1993): anneals over $\lambda$ only, with fixed $t$.

- **Model Predictive Path Integral (MPPI)** (Williams et al., 2018): no annealing; both $t$ and $\lambda$ are fixed, softmax update.

- **Cross-Entropy Method (CEM)** (Rubinstein & Kroese, 2004): no annealing; uses the top-$k$ update rule.

- **CMA-ES** (Akimoto et al., 2012): no explicit annealing of $t$, but adapts the proposal covariance from sampled points.

For the black-box tasks, each method uses 100 optimization iterations and 64 samples per iteration, and the table reports mean $\pm$ standard deviation over 8 random seeds. The implementation stores the best reward $J = -f$ observed during the run and reports the corresponding cost $-\max J = \min f$. The trajectory-optimization entries use the same lower-is-better reporting convention with the normalized task objective provided by the benchmark evaluator.

The GMM and checkerboard experiments are numerical visualizations of the smoothing mechanism rather than claims that every benchmark satisfies the sufficient assumptions. For the 1D GMM figure, the plotted function is $g(x,t) = -\log \rho_t(x)$ with $\rho_t(x) = \sum_{i=1}^{128} s_i \exp\left(-\frac{(x-\mu_i)^2}{2(t+3\cdot 10^{-3})}\right) + 10^{-2}$ on the plotted domain $[-1.5, 1.5]$. The means $\mu_i$ are sampled uniformly from $[-1, 1]$ with a fixed seed; most weights are equal, and two selected components near the reference optima are assigned larger weights to create the displayed dominant modes. For the checkerboard figure, $\rho_t(x) = \sum_i s_i \exp\left(-\frac{\|x-\mu_i\|^2}{2t}\right) + 10^{-2}$, where centers $\mu_i$ are selected from alternating cells of a 2D grid and the component near $(-0.25, 0.25)$ is assigned the largest weight. The Hessian eigenvalues are computed by automatic differentiation of $g = -\log \rho_t$; equivalently, the zero contour can be obtained from the sign-reversed Hessian of $\log \rho_t$. The reported local/global coverage regions are numerical certificates on the plotted domain, computed on a $2048 \times 2048$ grid by locating the nearest zero-level contour of $\lambda_{\min}(\nabla^2 g)$ around the reference optimizer.

In addition to the coverage analysis presented in Figure 3, we further investigate the geometric landscape of the probability distribution learned by the diffusion model. This analysis is performed across various noise

| Task/Env. | DIDA(ours) | MBD (Pan et al., 2024) | SA (Coleman et al., 1993) | MPPI (Williams et al., 2018) | CEM (Rubinstein & Kroese, 2004) | CMA-ES (Akimoto et al., 2012) |
|---|---|---|---|---|---|---|
| Ackley (d=200) | **3.1** ±0.1 | 6.8 ±0.3 | 14.0 ±0.1 | 14.2 ±0.1 | 14.3 ±0.1 | 14.2 ±0.1 |
| Ackley (d=400) | **4.4** ±0.2 | 8.8 ±0.2 | 14.4 ±0.0 | 14.6 ±0.0 | 14.7 ±0.1 | 14.6 ±0.0 |
| Ackley (d=800) | **6.0** ±0.1 | 9.9 ±0.1 | 14.6 ±0.1 | 14.8 ±0.0 | 14.9 ±0.0 | 14.8 ±0.0 |
| Levy (d=200) | **11.8** ±2.0 | 210.6 ±17.9 | 744.3 ±23.7 | 744.3 ±23.7 | 744.3 ±23.7 | 744.3 ±23.7 |
| Levy (d=400) | **53.6** ±5.0 | 628.4 ±29.1 | 1567.4 ±28.2 | 1567.4 ±28.2 | 1567.4 ±28.2 | 1567.4 ±28.2 |
| Levy (d=800) | **202.5** ±11.3 | 1508.4 ±43.7 | 3212.5 ±24.8 | 3212.5 ±24.8 | 3212.5 ±24.8 | 3212.5 ±24.8 |
| Rastrigin (d=200) | **1703.3** ±65.0 | 2823.3 ±74.3 | 3652.6 ±38.0 | 3648.8 ±43.4 | 3644.2 ±32.0 | 3648.8 ±43.4 |
| Rastrigin (d=400) | **3782.1** ±80.3 | 6224.6 ±101.5 | 7478.4 ±76.0 | 7478.4 ±76.0 | 7478.4 ±76.0 | 7478.4 ±76.0 |
| Rastrigin (d=800) | **8337.7** ±132.9 | 12947.9 ±48.3 | 15231.6 ±116.9 | 15231.6 ±116.9 | 15231.6 ±116.9 | 15231.6 ±116.9 |
| ant | 0.032 ±0.080 | **0.073** ±0.058 | 0.834 ±0.067 | 0.748 ±0.047 | 0.649 ±0.101 | 0.879 ±0.177 |
| halfcheetah | **0.414** ±0.042 | 0.906 ±0.008 | 0.997 ±0.006 | 0.924 ±0.024 | 0.998 ±0.008 | 0.995 ±0.006 |
| hopper | **0.623** ±0.007 | 0.749 ±0.010 | 0.924 ±0.008 | 0.855 ±0.030 | 0.861 ±0.006 | 0.929 ±0.010 |
| humanoid run | **0.298** ±0.059 | 0.356 ±0.031 | 0.998 ±0.009 | 0.928 ±0.083 | 0.973 ±0.008 | 0.989 ±0.017 |
| humanoid standup | **0.781** ±0.025 | 0.875 ±0.000 | 0.876 ±0.000 | 0.883 ±0.002 | 0.876 ±0.000 | 0.876 ±0.000 |
| humanoid track | **0.845** ±0.009 | 0.914 ±0.013 | 1.022 ±0.008 | 1.047 ±0.051 | 1.015 ±0.002 | 1.022 ±0.008 |
| pushT | **0.834** ±0.034 | 0.847 ±0.017 | 1.026 ±0.030 | 0.937 ±0.024 | 0.960 ±0.032 | 1.028 ±0.031 |
| walker2d | **0.352** ±0.062 | 0.756 ±0.011 | 0.849 ±0.001 | 0.745 ±0.016 | 0.850 ±0.001 | 0.848 ±0.001 |

Blackbox Optimization / Trajectory Optimization

Table 4: Optimized cost comparison of DIDA against sampling-based baselines on black-box optimization (Ackley/Levy/Rastrigin) and trajectory optimization tasks. Cost is the optimized objective value after the evaluation budget (lower is better); see the surrounding text for the precise definition.

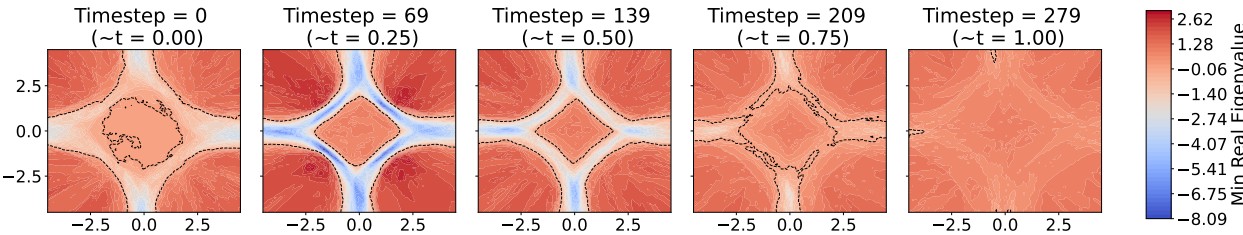

Figure 8: The smoothed landscape of diffusion model with different noise level $t$. When $t$ increases, the landscape becomes smoother.

levels (timesteps $t$) by examining the Hessian of the model's log-probability density function, $\nabla^2 \log p_t(x)$. The Hessian describes the local curvature of this landscape, offering insights into its structure.

The minimum eigenvalue of the Hessian is plotted in Figure 8 to illustrate the convexity of the landscape. The subtitles in the figure correspond to results at different timesteps $t$.

