# OpenReview forum: "Global Convergence of Sampling-Based Nonconvex Optimization through Diffusion-Style Smoothing"
_TMLR — Under review for TMLR_

### Review · Reviewer_RVf3 · 2026-04-21

**Summary Of Contributions:**

**Summary**

This paper establishes the nonasymptotic convergence rate of sampling-based optimization (SBO) for smooth nonconvex optimization with locally convex regions. The authors show that SBO can be interpreted as gradient descent on a smooth surrogate of the original loss function and characterize the convergence of SBO using this perspective. This perspective motivates new insights into the parameter selection and some new algorithm variants. Some experiments validate the claimed theoretical findings.

**Additional Comments:**

**Questions**

1. How would you choose the stepsize when you don't know $\alpha, \beta$?
2. Could you make your theorems more interpretable?
3. Could you conduct an ablation study to investigate how your proposed hyperparameter selection strategy compares with the optimal parameters obtained via grid search?

**Audience:**

Yes

**Audience Explanation:**

**Strength**

Overall, the paper is well written and easy to follow. The concepts are clearly presented.

**Broader Impact Concerns:**

N.A.

**Claims And Evidence:**

No

**Claims Explanation:**

**Weaknesses**

I find that some of the claims are overstated.

1. Global convergence

   The paper claims to derive global non-asymptotic convergence. However, the main results still rely on local initialization. This point should clearly be clarified.

2. Hyperparameter selection

   The paper focuses on a zeroth-order setting and derives results based on smoothness and strong convexity. Although the theoretical results provide some insights into parameter tuning, I don't think these parameters can be estimated efficiently, especially given that the only oracle access is function value evaluation.

Besides, some theorems make implicit restrictive assumptions on the problem parameters. I also notice a number of stylistic and presentation issues. See the comments below.

**Requested Changes:**

**Minor issues**

1. Figure 1

   $t=0$ is not italic.

2. Page 2

   - SBOC => SBO.

   - optimal $x^\*$ => optimum $x^\*$.

   - A recent discovery needs a reference.

   - The box for algorithm 1 looks strange. Why is it at the middle of literature review?
   - algorithm 1 starts from $m=1$, but initial guess is $x^0$.
   - gradient-free optimization uses => gradient-free optimization algorithms use.

3. Page 3

   - I didn't find the word "transport optimality" in the [Mobahi & Fisher, 2015]. Could you explain?
   - "bounds on the endpoint based on the function’s optimization complexity" is unclear.

4. Page 4

   - Why does the top-k update contain no k?
   - Please be consistent with zero-order and zeroth-order.
   - "let the Gaussian kernel ... be the zero-mean Gaussian kernel" is unclear.
   - The notation $p_{t|0}(y|x)$ looks undefined before. It looks like some density but actually denotes a distribution.

5. Page 6

   - The current statement of Theorem 4.3 is confusing. Some constants are used in the appendix but do not appear in the theorem body.
   - The theorem statement also makes implicit assumptions between problem parameters. For example, $D_\tau$, $\kappa_0$, and $d$ are your problem constants, but the relation $D_\tau^2 \geq bd$ requires $b \leq D^2_\tau / d$ and the requirement $b \geq 81e^{-1}\kappa_0^4$ imposes the restriction $81e^{-1}\kappa_0^4 \leq D^2_\tau / d$. This is not a natural assumption. Please make it clear in the revision.
   - It's inaccurate to say "a larger convex region provides a wider basin of attraction", since your strong convexity constant also vanishes, and the condition number of the smoothed problem can still scale badly with $t$.

6. Page 7

   - $\\|x - x^\*\\|$  is often known as distance to optimal set, optimality gap typically refers to the gap of between function value and optimal value. i.e., $f(x) - f(x^\*)$.
   - Theorem 4.4 only shows that two optimal solutions are close. It's insufficient to conclude graph convergence such as $g \rightarrow f$.

7. Page 8

   - "Given fixed noise $t$" is unclear.
   - Constant $M_{-1/2}, M_0, M_{1/2}$ (and $V$) appear undefined
   - It's not accurate to say Theorem 4.5 is a global convergence result, since it assumes that the initial iterate is in $R_{SC}(t)$. Please revise the claim unless you have a mechanism that guarantees finding a point in $R_{SC}$.
   - Where is $U_1$ and $U_0$ defined?
   - Again, Theorem 4.6 is too complicated to interpret. The theorem itself contains many new definitions, and quantities like $x_F^*$ look undefined.

8. Page 10

   - The table is again embedded in the text and looks strange.

**Requested changes**

See the comments above.

---

> ### Author Response · Authors · 2026-06-01
> **Global convergence wording, Hyperparameter selection and other minor fixes**
>
> # Reply to Reviewer RVf3
>
> We thank the reviewer for the careful and constructive comments. The two substantive concerns are the global-convergence wording and hyperparameter selection without first-order access; we address them first, then clarify the theorem assumptions and answer the closing question about ablations.
>
> ## "Global convergence" wording
>
> > The paper claims to derive ... should clearly be clarified.
>
> We agree that this wording should be more precise. In this paper, there are two convergence results. The first is the fixed-$t$ result, Theorem 4.5, which requires the initial iterate to lie inside the strongly convex region of the smoothed objective; we did **not** intend to present this result as **global convergence**.
>
> The second is the multi-stage result, Theorem 4.6. It starts from a large smoothing level so that the certified convex region is large enough to contain the initializer, and then gradually anneals the smoothing level while maintaining the invariant that the iterates remain inside the corresponding convex region. This gives a **conditional form of global convergence**, where by conditional we mean the global convergence holds only under the stated parameter conditions and for the class of functions covered by the theorem. Accordingly, we have revised the wording around "global convergence" to make these conditions explicit.
>
> ## Hyperparameter selection under zeroth-order access
>
> > Although the theoretical ... value evaluation.
>
> We agree that several constants appearing in the theorem, such as smoothness, strong convexity, and the switching threshold, are not directly available under zeroth-order access. Their role is to certify the scaling of the schedule rather than to provide quantities that must be estimated exactly in the implementation. In practice, as is common in derivative-free optimization, DIDA uses empirical surrogates: the smoothing level follows the geometric schedule $t_m=\gamma^m t_0$, and the temperature is set from the current batch variance of sampled objective values, $\lambda_m=\sqrt{\mathrm{Var}(f(y_i))}$. This empirical choice corresponds to estimating the local scale $\beta^2\approx \mathrm{Var}(f(y_i))/t_m$, and therefore follows the same annealing trend as the theoretical prescription $\lambda_m\propto \beta\sqrt{t_m}$. Thus, even when the exact constants needed for the proof are unavailable, the practical schedule preserves the scaling insight supplied by the theorem. We make this **theory--practice distinction** explicit in the revision.
>
> ## Theorem interpretability and hidden assumptions
>
> > Some theorems ... on the problem parameters.
>
> We agree with the reviewer. The previous statement hid real parameter restrictions behind Equation 20, making Theorem 4.3 harder to interpret than it should be. The revision explicitly states that the result is a **sufficient high-dimensional regime** and spells out the implied conditions, including $D_\tau^2 \ge \frac{1-C_\alpha}{18}\kappa_0^2 d$ and $D_\tau^2 \ge \frac{81}{8e}\kappa_0^4 d$, together with the restrictions on $P_{\mathrm{out}}$, $\tau$, $C_E$, and $\lambda$. We also clarify that these conditions are **not claimed to be necessary**.
>
> ## Minor issues
>
> We have applied the following and other minor fixes the reviewer noticed in the revised manuscript:
>
> - Clarified the Gaussian proposal notation: $P_{t|0}(\cdot\mid x)$ is the distribution, while $p_{t|0}(y\mid x)=k(y-x;t)$ is its density.
> - Clarified optimizer displacement versus function-value optimality gap.
> - Revised the Mobahi--Fisher paragraph to use "convex-envelope approximations" and "optimization-complexity measure" wording.
> - Reworded Theorem 4.5 to start from "For any fixed smoothing level $t>0$"
> - Added a coverage-vs-conditioning caveat after Theorem 4.3: the strong-convexity constant $\alpha_t$ shrinks as $t$ grows, so the basin enlarges while local contraction slows
> - Softened the diffusion-implications wording from "enables us to understand" to "offers a conceptual lens"
> - Defined $M_{-1/2}, M_0, M_{1/2}$ and the $V$ constants as dimension-independent constants from the SNIS gradient-estimator bounds.
> - Removed the undefined $U_0,U_1$ notation and replaced the optimal-temperature scaling with $\Theta(\cdot)$ notation.
> - We have revised the wording to clarify that Theorem 4.5 is a local convergence result, and that Theorem 4.6 provide wider initialization coverage under the stated conditions.
> - Make Theorem 4.6 interpretation more explicit: We have added extra explanations to clarify the assumptions and implications of Theorem 4.6, including the sufficient high-dimensional regime, the role of the switching threshold, and the practical hyperparameter choices under zeroth-order access. We also define $x_F$ as the user-specified swiching point for annealed stage and fixed noise stage.
> - Reformatted the main performance table as a regular full-width table and expanded it to include the component-ablation baselines.

---

> ### Author Response · Authors · 2026-06-01
> **Closing questions**
>
> ## Closing questions
>
> > How would you choose the stepsize when you don't know $\alpha, \beta$?
>
> We choose stepsize $\eta = \frac{1}{4} t$ as we stated in the algorithm design section, which is based on the assumption of $\alpha_t \approx \frac{1}{t}$ and $\beta_t \approx \frac{1}{t}$, which is a good approximation when $t$ is large.
> In practice, we find this choice to be robust across all tasks and smoothing levels, and it is the default stepsize used in the experiments.
>
> > Could you make your theorems more interpretable?
>
> Thank you for the suggestion. We have revised the theorem statements and added explanations to make both the assumptions and implications more explicit. In particular, for Theorem 4.3, we now spell out the sufficient parameter regime rather than hiding it behind a compact condition. In short, the explicit condition still only requires $D_\tau^2 \sim O(d)$ , which is practical in the high-dimensional setting considered in the paper.
>
> We have also clarified the role of each theorem. Theorem 4.3 shows that smoothing enlarges the certified strongly convex region, while Theorem 4.4 shows the accompanying tradeoff: the minimizer of the smoothed objective may be displaced from the original optimizer. Theorems 4.5 and 4.6 then explain how this tradeoff affects optimization. The fixed-$t$ result in Theorem 4.5 gives convergence once the initializer lies in the smoothed convex region, but using a single large $t$ can leave a large optimizer displacement $\|x_t^\star-x^\star\|$. The multi-stage result in Theorem 4.6 addresses this by starting with large smoothing for coverage, then decreasing $t_m$ geometrically so that the final bound depends on the small final smoothing level $t_F$. This gives the desired balance: the initializer can start farther away, while the final optimizer displacement remains controlled by $t_F$.
>
> > Could you conduct an ablation study comparing the proposed hyperparameter strategy with grid-searched parameters?
>
> We have added a sensitivity study to Section 5 (the sensitivity figure) sweeping the three hyperparameters of DIDA on the nine black-box tasks: temperature $\lambda$, sample size $N\in\{32,\dots,16384\}$, and global iteration count $M\in\{50,\dots,12800\}$. Three findings:
>
> 1. $\lambda=1$ is robust across all functions; it balances the SNIS variance and the optimizer-displacement bias predicted by Theorem 4.4.
> 2. Cost decreases near-monotonically with $N$ and $M$, matching the $1/\sqrt{N}$ SNIS rate from Theorem C.1 and the per-stage contraction from Theorem 4.6. Both curves plateau at $N\gtrsim 4096$ and $M\gtrsim 6400$; e.g., on Ackley $d=800$, increasing $N$ from 64 to 16384 reduces cost from $6.06$ to $1.34$ ($4.5\times$), while the increase from $M=6400$ to $M=12800$ gives only a small drop.
> 3. The **default budget** $(N,M)=(64,100)$ used in Table 1 already sits at the knee of these curves and beats every baseline at any budget reported. Increasing $N$ or $M$ further yields graceful but diminishing returns (mean residual gap-to-best $\approx526\%$ for $N$ and $\approx161\%$ for $M$, both at the plateau).
>
> We therefore recommend starting at $\lambda=1$ with the default $(N,M)$ and scaling the latter only when compute permits. Per-task tables for the full sweeps are included in Appendix F.

---

### Review · Reviewer_syCe · 2026-05-16

**Summary Of Contributions:**

The paper explains sampling-based optimization as zeroth-order gradient descent on a smoothed version of the objective. Its main contribution is a theory showing a coverage–optimality tradeoff: more smoothing makes the landscape easier to optimize by enlarging a convex region around the global minimizer, but it also introduces bias by shifting the smoothed optimum away from the true optimum.

Using this insight, the authors prove non-asymptotic convergence guarantees for sampling-based optimization and propose DIDA, a dual-annealing algorithm that jointly decreases the smoothing level and temperature to converge to the global optimum. Experiments on synthetic and trajectory-optimization tasks support the theory and show DIDA outperforming baselines such as CEM and CMA-ES.

Strengths: clear theoretical framing, useful algorithmic insight, and promising experiments. Weaknesses: assumptions are restrictive, multi-modal settings are not fully addressed, and the diffusion-model implications are mostly conceptual.

**Audience:**

Yes

**Audience Explanation:**

At least some TMLR readers would likely be interested, especially those working on zeroth-order optimization, sampling-based optimization, homotopy/annealing methods, control, and diffusion models.

The paper’s main appeal is its unifying view of SBO as optimization over a smoothed objective, together with theoretical convergence guarantees and a practically motivated dual-annealing algorithm. Even if the assumptions limit generality, the perspective and algorithmic implications are relevant to a broad subset of the TMLR audience.

**Claims And Evidence:**

Yes

**Claims Explanation:**

The theory is coherent and supports the main smoothing-based interpretation of SBO, and the experiments provide encouraging evidence that DIDA performs well on the tested benchmarks.

However, the strongest claims are only convincing under the paper’s fairly restrictive assumptions, and the empirical evidence would be stronger with more baselines, ablations, and sensitivity analysis. The diffusion-model implications are suggestive rather than fully demonstrated.

**Requested Changes:**

The paper should more clearly explain how restrictive the sub-Gaussian concentration and local convexity assumptions are, and which nonconvex problems are actually covered.

The authors should separate the effects of smoothing annealing, temperature annealing, sample size, and annealing rate to support the algorithmic claims.

The experiments would be more convincing with additional modern zeroth-order, annealing, and diffusion-inspired baselines, plus sensitivity analysis.

---

> ### Author Response · Authors · 2026-06-01
> **Assumption Scope and Ablations**
>
> # Reply to Reviewer syCe
>
> We thank the reviewer for the constructive summary and the three concrete asks: clarify the assumption scope, frame the diffusion implications, and strengthen the empirical evidence with ablations.
>
> ## Assumption scope
>
> > The paper should more clearly explain how restrictive the sub-Gaussian concentration ... actually covered.
>
> We agree. The intended sub-Gaussian assumption is on the concentration of the Boltzmann target distribution $p_0(x)\propto \exp(-f(x)/\lambda)$, hence on tail growth rather than global convexity. Nonconvex objectives with quadratic tail growth can satisfy it, and the revision now gives two concrete non-log-concave examples.
>
> **Example 1: certified Gaussian state-penalty construction.** We have added a 2D linear-control objective of the form $J(u)=Q_{\mathrm{LQR}}(u)+\sum_j \epsilon_j \Vert z(u)\Vert^2\exp(-\Vert z(u)-z_j\Vert^2/(2\sigma_j^2))$: a quadratic LQR core plus bounded localized Gaussian/RBF state penalties, a standard smooth soft-obstacle or keep-out cost in motion planning. The quadratic core gives **sub-Gaussian tails** for $p_0(u)\propto \exp(-J(u)/\lambda)$, while localized penalties create negative-curvature regions, so the density is **not log-concave**. This is exactly **confining quadratic tails** plus local regularity near the optimizer, but with localized nonconvexity. For quick visual reference, this is Figure 1 in `rebuttal.pdf`; full parameters and certification are in Appendix B.2 and manuscript Figure 5.
>
> **Example 2: nonlinear residual control landscape from the literature.** [Qu et al. (2020)](https://arxiv.org/abs/2006.07476) give a one-dimensional nonlinear-control example with residual dynamics $x_{t+1}=Ax_t+u_t+f(x_t)$, $f(x)=0.01x/(1+0.9\sin x)$, feedback $u_t=-Kx_t$, and finite-horizon cost $C_H(K)=\sum_t(Qx_t^2+Ru_t^2)$. The small residual creates many local minima, while the finite-horizon cost has quadratic growth in $K$; thus the induced Boltzmann density has **sub-Gaussian tails** but is **not log-concave**. For quick visual reference, this is Figure 2 in `rebuttal.pdf`; details are in Appendix B.2 and manuscript Figure 6.
>
> ## Baselines and ablations
>
> > The authors should separate the effects of smoothing annealing, ... to support the algorithmic claims.
>
> The five baselines we compare against in the main performance table (now moved to main text from the appendix) are themselves a **structural ablation** of the two annealing components, organized along the colored axes of the baseline-comparison table:
>
> - **MBD** (Pan et al., 2024) anneals over $t$ only and is recovered from DIDA by removing the temperature annealing.
> - **SA** (Coleman et al., 1993) anneals over $\lambda$ only.
> - **MPPI** (Williams et al., 2018) fixes both $t$ and $\lambda$.
> - **CEM** (Rubinstein and Kroese, 2004) fixes both and uses the top-$k$ update.
> - **CMA-ES** (Akimoto et al., 2012) adapts the proposal covariance from samples but does not anneal $t$.
>
> The clear ordering DIDA > MBD > SA across nearly all rows of the main performance table therefore isolates the **smoothing-annealing contribution** (DIDA vs. MBD: removing temperature annealing only) from the **temperature-annealing contribution** (MBD vs. SA: removing smoothing annealing only), matching the role each plays in Theorem 4.6.
>
> > The experiments would be more convincing with additional ... plus sensitivity analysis.
>
> For sensitivity, the sensitivity figure (new in the revision) sweeps the temperature $\lambda$, sample size $N\in\lbrace32,\dots,16384\rbrace$, and global iteration count $M\in\lbrace50,\dots,12800\rbrace$ on the nine black-box tasks. Three findings:
>
> 1. **$\lambda=1$ is robust** across all functions; it balances the SNIS variance and the optimizer-displacement bias predicted by Theorem 4.4.
> 2. Cost decreases near-monotonically with $N$ and $M$, matching the $1/\sqrt{N}$ SNIS rate from Theorem C.1 and the per-stage contraction from Theorem 4.6; both knobs plateau at $N\gtrsim 4096$ and $M\gtrsim 6400$.
> 3. The default $(N,M)=(64,100)$ used in the main performance table already sits at the knee of these curves and beats every baseline at any budget reported.
>
> The current set covers the methods most directly comparable to DIDA's smoothing/temperature-annealing structure, while the sensitivity study addresses the requested hyperparameter evidence. If the reviewer has suggestions on more concrete baselines we should compare, feel free to let us know.
>
> ## Diffusion-model implications
>
> > The diffusion-model implications are mostly conceptual.
>
> We agree and have softened the framing in the diffusion-implications section. The discussion is now presented as a **conceptual** optimization-perspective interpretation of noise-conditioned diffusion sampling, not as a complete empirical characterization of diffusion-model behavior. We have also removed the stronger claim wording.

---

### Review · Reviewer_PP9Q · 2026-05-18

**Summary Of Contributions:**

The paper tackles the question of finding global minima in non-convex landscape via a sampling-based optimization technique. Consider the optimization problem $\min_{x\in R^d}f(x)$ with global minimizer $x^\star$. The core contributions of this paper are the following --

1. **Algorithm**: The authors consider a sampling-based optimization (Algorithm 1) which samples $N$ points, $\{y_i \}$ from $\mathcal{N}(x_m, t I_d)$, where $x_m$ is the $m^{th}$ iterate. Then, the authors consider a soft-min update rule, to set the next iterate as $x_{m+1} = \sum_{i=1}^N \frac{\exp(-\frac{f(y_i)}{\lambda})}{\sum_{i'=1}^N \exp(-\frac{f(y_i)}{\lambda})} y_i$.
2. **Landscape after Smoothing:** Under Assumptions 4.1 and 4.2, which ensure that the density $p_{0}(x) \propto \exp(-\frac{f(x)}{\lambda})$ has a sub-Gaussian tail beyond a radius $D_{\tau}$ away from the minima , and inside the region given by $D_{\tau}$, it is strongly convex and smooth, the authors show that --
    - Theorem 4.3 (Coverage): The strongly-convex region $D_{\tau}$ after the gaussian smoothing is bounded with a change in the strong convexity parameter depending on $t$.
    - Theorem 4.4 (Optimality Gap): The gap between the optimal of the smoothed function, $x_t^\star$ and the true optimal $x^\star$ is bounded by terms depending on $t$.

3. **Convergence of Algorithm 1:**
    - Theorem 4.5 : The authors provide convergence of Algorithm 1 to the global minima $x^\star$ for a constant step-size. The convergence rate consists of $3$ terms -- a) the gap between $x_t^\star$ and $x^\star$, b) a linear convergence rate in number of iterations, c) an additional error term due to high probability guarantees, that scales as $\frac{1}{N}$.
    - Theorem 4.5 and Algorithm 2: The authors propose an algorithm that varies the smoothing parameter $t$ by decreasing it geometrically until it reaches an optimal that balances optimality gap and coverage in Theorems 4.4 and 4.3 respectively. For such a strategy, the convergence rate is again bounded by $3$ terms, which are similar to those in Theorem 4.5, however, with the best possible values of optimality gap and coverage.

4. **Experimental evidence:** For a wide-range of non-convex problems in black-box optimization, for instance the standard ackley and rastrigin functions, and in trajectory optimization, the authors show that their algorithm achieves state-of the art "cost" compared to existing SBO baselines like MBD, SA, MPPI, CEM and CMA-ES.





### Strengths --
- **Novel Loss-Landscape Analysis**: The analysis of the loss landscape under these assumptions seems novel, while several recent empirical work in SBO lacked these guarantees for instance the baselines MBD.
- **Experimentally superior:** The algorithm with varying smoothing outperforms all baselines even for highly non-convex functions which might not satisfy the assumptions, thus showing the benefit of this algorithmic innovation.
- **Connection to diffusion with varying temperature:** The authors establish a connection between their algorithm with varying step-sizes and diffusion with varying temperatures.

### Weaknesses  --
- **Feasibility of assumptions for theory:** The authors consider non-convex loss functions, but beyond the the loss is non-convex only beyond the region given by $D_\tau$, which is far from the optimal. The region beyond $D_{\tau}$, requires Assumption 4.3, which forces a sub-Gaussain tail. *A subGaussian tail is not possible if the function is not strongly-convex or close to strongly-convex in this region.*  If the function is strongly-convex in all of $R^d$, then this is no longer a convex function. The authors state that several functions in regularized ML or LQR control obey these assumptions and that these assumptions go beyond log-concavity. Can they  provide a concrete example of a non-convex function, that does not induce log-concavity, but satisfies their assumptions? In the current formulation of Assumption 4.3, I believe this should not be possible.

- **Interpretation of Theoretical Results:** The theoretical results have not been well-interpreted and as such are hard to digest. Several important implications have not been discussed. Firstly, in Theorem 4.3, to truly expand the convex region, we need the terms mutliplied by $D_{\tau}$ to be $>1$. Since $C_{E} = \sqrt{\frac{\log d}{d}}$, we are forced to use $t \geq \frac{d}{\log d}$. This would force a large optimality gap (Theorem 4.4), even for the smallest coverage that is an expansion. From Theorem 4.4, this gap is approximately minimum of $\frac{d}{\log (d)}$ and $D_{\tau}$, the radius of the initial convex region. Such a large optimality gap almost defeats the purpose of smoothing, as the smoothed algorithm can end at the boundary of the initial convex region. Secondly, in Theorems 4.5 and 4.6, how do we know that using a varying step-size actually benefited us? The authors only provide the expressions for convergence, but do not quantify, how large are the additional terms of $K_F, \sigma_F$, compared to the original terms in Theorem 4.5 for an arbitrary $t$. Further, they also show convergence only after $t_{M_0}$, the optimal has been smoothing parameter has been reached. This can be seen by the $(M-M_0)$ convergence rate. Also, the authors never formally define $M_0$ or $t_{M_0}$, but only motivate its existence. I'm not sure if they use the fact that it is optimal in the proofs. Consequently, $t_{M_0}$ can be replaced by any $t^\star$ and the whole proof of Theorem 4.6 would still hold. Also, they never mention what the terms of $V$ in Theorems 4.5 and 4.6 are, either in the main paper or in Appendix C. Are these absolute constants with respect to all problem parameters like $d$ and can they be ignored in the $\mathcal{O}(\cdot)$ notation? Further, the dependence on probability of error is $\frac{1}{\delta}$, which means that for an extremely small probability of error, we incur a large error in terms of optimality gap. Lastly, all the convergence results rely on all iterates lying inside the convex region after smoothing. While the first iterate might lie in this region, the authors don't prove that the subsequent iterates also lie in this region explicitly. They have assumed that this is true and used it in their proofs. Further, they have not quantified how the smoothness, and Lipschitz parameter of the loss changes with $t$. Note that a significant chunk of existing works deal with quantifying precisely this, and it is non-trivial.

- **Mistake in proofs of convergence results (Theorems ):** There is a mistake in the proof of Theorems 4.5 and 4.6 as both these results build on top of Theorem C.3. Theorem C.3 assumes that the gradient estimator has a fixed bias, bounded by $K$, i.e., $||g(x) - \nabla f(x)|| \leq K$, where $g(x)$ is the estimate of gradient. The authors plug-in $g(x)$ from Proposition 3.1, which shows that the soft-min update is a zeroth-order estimate of the gradient. The issue is that the *expected bias* of the gradient is bounded, i.e., $E[||g(x) - \nabla f(x)||] \leq K$, however, since $g(x)\in R^{d}$, the absolute value of the gradient estimate, i.e., $\max_{x\in R^d} || g(x) - \nabla f(x) ||$ can be be unbounded. I believe this can be fixed, by using Markov's inequality again, as they have done for variance in Page 52, however, this would further force sub-optimal terms of $\frac{1}{\delta}$.

- **Lack of comparison to existing works in smoothing-based optimization:** The authors have not provided a comparison to their theoretical results to existing convergence results for smoothing for convex/strongly-convex and non-convex functions in (Nesterov \& Spokoiny 2017, Duchi et al 2014, Hazan et al 2015, Mobahi \& Fisher 2015). Further, they have also missed certain references to smoothing-based methods for non-convex functions in (Vardhan \& Stich, 2021) and (Mishchenko \& Stich, 2023). Further, the varying smoothness parameters as an idea had already been explored in (Hazan et al 2015), with exact bounds for a certain class of non-convex functions. The authors have not explicitly shown how their assumptions can extend this. Finally, geometrically decreasing step-sizes have been well-studied in stochastic optimization, and closely resemble the case of geometrically decreasing smoothing parameters, with optimal step-size dependent on the smoothing parameter (Ge et al 2019).


- **Important experimental details are missing:** I'm not sure if any of the functions considered in their experiments in Section 5 and Appendix D actually satisfy either Assumption 4.1 or 4.2. Further, the authors never define what their main metric of "cost" in Tables 1 and 3. Additionally, they never explicitly state what the loss function of a GMM looks like, but they use it in Figure 1. The authors have also not provided a strategy to compute $t_{M_0}$. Their actual algorithm doesn't require it, but their theoretical guarantees do. Finally, in Figure 3, several quantities have not been defined in the legend --  what is the red star and the purple dot, what is the green dashed circle, do the blue and red shades in the left-most subfigures denote the function value, how did they compute the local coverage and global coverage regions for the right-most subfigure, how did they verify that these are indeed convex regions?



**References**--

- (Vardhan \& Stich 2021) Escaping Local Minima With Stochastic Noise. OPT.
- (Mischenko \& Stich 2023) Noise Injection Irons Out Local Minima and Saddle Points. OPT.
- (Ge et al 2019) The Step Decay Schedule: A Near Optimal, Geometrically Decaying Learning Rate Procedure For Least Squares. Neurips.

**Additional Comments:**

None.

**Audience:**

Yes

**Audience Explanation:**

The field of smoothing-based optimization is a relevant subfield of optimization, especially, for minimizing non-convex functions, where standard first-order gradient based methods don't have good theoretical convergence.

**Broader Impact Concerns:**

None.

**Claims And Evidence:**

No

**Claims Explanation:**

The assumptions do not hold for a non-convex objective, the theoretical results have mistakes in their proofs, and the experimental metrics have not been defined.

**Requested Changes:**

Please fix the main weaknesses.
There are several additional typos or missing explanations in the paper. Some examples are below --
1. Definition of $SG$ in Table 2, which doesn't have a caption, is incorrect in terms of both the sign as well as dividing a vector by a matrix.
2. Lemma B.2: Where is its proof or a reference for it?
3. What is $\pi(g)$ in Eq (136) and (137)? Also, both these equations correspond to a single equation but have been numbered as $2$ different equations.

Overall, I feel the experimental evidence suggests that geometrically decreasing smoothing is a viable approach for non-convex functions; however, the precise theoretical conditions for it have unfortunately not been properly explained by this draft.

---

> ### Author Response · Authors · 2026-06-01
> **Feasibility of the assumptions and examples**
>
> ## Feasibility of the assumptions and examples
>
> > A subGaussian tail is not possible if the function is not strongly-convex or close to strongly-convex in this region. If the function is strongly-convex in all of $\mathbb R^d$, then this is no longer a convex function. Can they provide a concrete example of a non-convex function that does not induce log-concavity, but satisfies their assumptions?
>
> Thank you for raising this point. The intended sub-Gaussian assumption is on the concentration of the Boltzmann target distribution $p_0(x)\propto \exp(-f(x)/\lambda)$. This is better understood as an assumption on the growth rate of $f$, not on global convexity of $f$. In particular, the assumption can hold for nonconvex objectives whose tails are dominated by a quadratic growth term, even when the objective is not globally convex. The revision now includes two concrete examples of this regime.
>
> **Example 1: certified Gaussian state-penalty construction.** We have added a 2D linear-control objective of the form $J(u)=Q_{\mathrm{LQR}}(u)+\sum_j \epsilon_j \Vert z(u)\Vert^2\exp(-\Vert z(u)-z_j\Vert^2/(2\sigma_j^2))$: a quadratic LQR core plus bounded localized Gaussian/RBF state penalties. This is a standard smooth soft-obstacle or keep-out cost used in artificial-potential-field and MPC-style motion planning. The quadratic core controls tail growth, so the induced Boltzmann density $p_0(u)\propto \exp(-J(u)/\lambda)$ has **sub-Gaussian tails**. The localized penalties create compact negative-curvature regions, so the same density is **not log-concave**. This is the regime our assumptions are meant to capture: **confining quadratic tails** plus local regularity near the optimizer, but localized nonconvexity that global log-concavity would exclude. For quick visual reference, this is Figure 1 in `rebuttal.pdf`; full parameters and certification are in Appendix B.2 and manuscript Figure 5.
>
> **Example 2: nonlinear residual control landscape from the literature.** Separately, [Qu et al. (2020)](https://arxiv.org/abs/2006.07476) give a one-dimensional nonlinear-control example with residual dynamics $x_{t+1}=Ax_t+u_t+f(x_t)$, nonlinear residual $f(x)=0.01x/(1+0.9\sin x)$, linear feedback $u_t=-Kx_t$, and finite-horizon objective $C_H(K)=\sum_t(Qx_t^2+Ru_t^2)$. Their example shows that a small nonlinear residual can create many local minima in the scalar policy-cost landscape. The same finite-horizon objective has quadratic growth in the scalar policy parameter, so the induced Boltzmann density has **sub-Gaussian tails**; the many local minima make it **not log-concave**. This gives a second control example of the same phenomenon: **confining tails plus localized nonconvexity**, beyond what global log-concavity can express. For quick visual reference, this is Figure 2 in `rebuttal.pdf`; details are in Appendix B.2 and manuscript Figure 6.

---

> ### Author Response · Authors · 2026-06-01
> **Interpretation of Theoretical Results Part 1**
>
> ## Interpretation of Theoretical Results Part 1
>
> > The theoretical results have not been well-interpreted and as such are hard to digest. ... Such a large optimality gap almost defeats the purpose of smoothing, as the smoothed algorithm can end at the boundary of the initial convex region.
>
> Thank you for pointing out the fixed-$t$ tradeoff. We agree that if one uses a single very large smoothing level $t$, Theorem 4.3 gives a larger smoothed strongly convex region $\mathcal{R}_{SC}(t)$, but Theorem 4.4 also implies that the smoothed minimizer $x_t^\star=\arg\min_x g(x;t)$ may be farther from the original optimizer $x^\star$. A large fixed $t$ can therefore improve coverage while leaving a large optimizer displacement $\|x_t^\star-x^\star\|$. A large fixed $t$ is therefore not the intended final algorithmic regime; it is exactly the **coverage--displacement tradeoff** that motivates the **multi-stage method**, as discussed in the response to your next comment.
>
> As a side note, "displacement" means distance between the minimizer of the smoothed objective and the original minimizer, not a function-value optimality gap. We intentionally changed the terminology in the rebuttal and in the revision to be clearer; previously, the original manuscript used "coverage-optimality tradeoff."
>
>
> > Secondly, in Theorems 4.5 and 4.6, how do we know that using a varying step-size actually benefited us?
>
> The benefit of annealing over $t$ is that it makes sure the algorithm can converge starting from a much wider range of initial points, as opposed to the restricted initializer assumption for the fixed $t$ regime (Theorem 4.5).
>
> Specifically, Theorem 4.5 requires the initializer to start from the convex region corresponding to the fixed $t$. This may be hard to satisfy if one uses a small $t$, as it will result in a small convex region; a fixed large $t$ can enlarge the convex region but leaves a large optimizer displacement $\|x_t^\star-x^\star\|$, as the reviewer also correctly identified.
>
> In contrast, the multi-stage theorem (Theorem 4.6) uses large initial smoothing for coverage, then decreases $t_m$ geometrically so that the final bound depends on the small final smoothing level $t_F$. This achieves the best of both worlds: it allows the initializer to start potentially far away (as one can always use a large enough initial $t$), while still ensuring that the final optimizer displacement depends on the small final smoothing level $t_F$.
>
> > The authors only provide the expressions for convergence, but do not quantify, how large are the additional terms of $K_F, \sigma_F$, compared to the original terms in Theorem 4.5 for an arbitrary $t$.
>
> The bound for $K_F, \sigma_F$ is specified in the gradient-estimator bound, Theorem C.1. As we mentioned in the previous response, the benefit of Theorem 4.6 compared to Theorem 4.5 is mainly the ability to allow the initializer to be much further away from $x^\star$, as opposed to the more restrictive initialization assumption in the fixed-$t$ theorem.
>
> Due to the above difference in initialization assumptions, we believe it is not an apples-to-apples comparison for all the constants in Theorems 4.5 and 4.6. We have updated the paper to clarify this point.
>
> > Further, they also show convergence only after $t_{M_0}$, the optimal smoothing parameter has been reached. This can be seen by the $(M-M_0)$ convergence rate.
>
> Yes, this is because our algorithm is composed of two stages:
> 1. annealing over $t$ for global convergence, in which case the proof mainly shows the iterate stays in the convex region as $t$ shrinks;
> 2. optimize at fixed $t_F$ for final convergence. Our linear convergence rate comes from the final-stage optimization, while the first stage is designed to make sure the final-stage initial point is in its local convex region.

---

> ### Author Response · Authors · 2026-06-01
> **Interpretation of Theoretical Results Part 2**
>
> ## Interpretation of Theoretical Results Part 2
>
> > Also, the authors never formally define $M_0$ or $t_{M_0}$, but only motivate its existence. I'm not sure if they use the fact that it is optimal in the proofs. Consequently, $t_{M_0}$ can be replaced by any $t^\star$ and the whole proof of Theorem 4.6 would still hold.
>
> We agree that the $t_{M_0}$ quantity in the original manuscript can be confusing, and the use of $t_{M_0}$ is actually not needed. In the revised paper, we have consolidated $t_{M_0}$ and $t_F$ to be the same quantity to streamline the algorithm. In the revised manuscript, $t_F$ is a user-chosen parameter: it is a switching threshold between the two optimization stages and we define $M_0 = \lceil \log_\gamma(\frac{t_0}{t_F})\rceil$ as the corresponding switching index: which is the index that the geometrically decreasing $t$ first drops below $t_F$.
>
> > Also, they never mention what the terms of $V$ in Theorems 4.5 and 4.6 are, either in the main paper or in Appendix C. Are these absolute constants with respect to all problem parameters like $d$ and can they be ignored in the $\mathcal{O}(\cdot)$ notation?
>
> Parameters like $V$ and $M$ are dimension-$d$ independent constants from Theorem C.1. We have updated the paper to clarify this point and pulled the dimensionality parameter to the front of the equation. They cannot be ignored, as they are coefficients for different powers of $t$ in the bias and variance bounds, which determine the relative weight of the $t^{-1/2}$, $t^0$, $t^{1/2}$ terms in the bias bound (and of $t^{-1}$, $t^0$, $t$ in the variance bound).
>
> > Further, the dependence on probability of error is $\frac{1}{\delta}$, which means that for an extremely small probability of error, we incur a large error in terms of optimality gap.
>
> We agree that the $\frac{1}{\delta}$ dependence is the conservative price paid for invoking **Markov's inequality** on the accumulated squared stochastic-error term, and we want to clarify exactly where it enters and why it is **not intrinsic to our framework**.
>
> The $1/\delta$ factor is inherited from the moment-only concentration bound for the SNIS softmax gradient estimator in Theorem C.1 which is from the literature: it controls only the first moment (bias) and the second moment of the estimator error. With only first and second moments available, the only route to a high-probability statement is Markov's (or Chebyshev's) inequality applied to the accumulated squared error, which forces a $1/\delta$ failure-probability dependence rather than the $\log(1/\delta)$ dependence obtainable from sub-Gaussian or sub-exponential tail bounds. The $1/\delta$ rate is therefore a property of the currently available concentration bound for the SNIS softmax gradient estimator, not of our optimization analysis.
>
> If a stronger concentration result for this estimator becomes available, for example a sub-Gaussian or Bernstein-type tail bound on $\hat g(x,\xi;t)-g(x;t)$ uniformly over the certified region $\mathcal R$, then the same proof of Theorems 4.5 and 4.6 immediately yields a $\log(1/\delta)$ dependence by replacing the Markov step with the sharper concentration inequality; no other part of the multi-stage tracking argument changes. Sharpening tail bounds for self-normalized importance-sampling estimators is itself an active research direction, and we view the present $1/\delta$ rate as a placeholder that can be tightened modularly as that line of work matures. We have added this discussion in the revised manuscript so the reader can see precisely which lemma controls the $\delta$-dependence and how an improved concentration bound would propagate through the convergence rate.
>
> > Lastly, all the convergence results rely on all iterates lying inside the convex region after smoothing. While the first iterate might lie in this region, the authors don't prove that the subsequent iterates also lie in this region explicitly. They have assumed that this is true and used it in their proofs. Further, they have not quantified how the smoothness, and Lipschitz parameter of the loss changes with $t$. Note that a significant chunk of existing works deal with quantifying precisely this, and it is non-trivial.
>
> This event is proved by induction in the appendix multi-stage theorem, Theorem C.4, **not assumed**. Under the sample-size condition, with probability $1-\delta$, the invariant $\|x_m-x^\star\|^2\le r_{\min}+k_gt_m$ holds for every tracking stage $m\le M_0$. This invariant implies $x_m\in\mathcal R_{\mathrm{SC}}(t_m)$, and the switching condition implies $x_{M_0}\in\mathcal R_{\mathrm{SC}}(t_F)$ for the final local-search phase.
>
> In the revised manuscript, we make this part of the proof more explicit to avoid any confusion.

---

> > ### Comment · Reviewer_PP9Q · 2026-06-01
> >
> > Theorem C.4 considers the multi-stage case. But, the requirement for all iterates to lie inside the convex region also holds for the single-stage case (Theorem C.3 and 4.5). For this case, the first iterate lying inside the convex region doesn't imply subsequent iterates lying inside the convex region.

---

> > > ### Author Response · Authors · 2026-06-03
> > >
> > > Thanks for pointing this out. The staying-in-convex-region for the fixed-t case (Theorem 4.5) can be obtained using the same argument as in the proof from Theorem 4.6 (as a simplified special case with $\gamma = 1$), so the staying-in-the-convex-region with high probability is guaranteed with the same union bound with the same $k/\delta$ for this fixed t case. We agree that the original manuscript did not explicitly prove this (but it is essentially a simple consequence of the same argument in Thm 4.6).  We have made this point clear in Theorem 4.5.

---

> ### Author Response · Authors · 2026-06-01
> **Proof issue in the convergence theorems**
>
> ## Proof issue in the convergence theorems
>
> > There is a mistake in the proof of Theorems 4.5 and 4.6 as both these results build on top of Theorem C.3. Theorem C.3 assumes that the gradient estimator has a fixed bias, bounded by $K$, i.e., $||g(x) - \nabla f(x)|| \leq K$, where $g(x)$ is the estimate of gradient. The authors plug-in $g(x)$ from Proposition 3.1, which shows that the soft-min update is a zeroth-order estimate of the gradient. The issue is that the *expected bias* of the gradient is bounded, i.e., $E[||g(x) - \nabla f(x)||] \leq K$, however, since $g(x)\in R^{d}$, the absolute value of the gradient estimate, i.e., $\max_{x\in R^d} || g(x) - \nabla f(x) ||$ can be unbounded. I believe this can be fixed, by using Markov's inequality again, as they have done for variance in Page 52, however, this would further force sub-optimal terms of $\frac{1}{\delta}$.
>
>
> We thank the reviewer for catching this. As suggested, we have updated the proof to apply **Markov's inequality** to the bias term:
> $$
> \mathbb{P}\!\left(\|b_m\|\ge \tfrac{2M}{\delta}K_m\right)\le \tfrac{\delta}{2M},\qquad
> \mathbb{P}\!\left(\|w_m\|^2\ge \tfrac{2M}{\delta}\sigma_m^2\right)\le \tfrac{\delta}{2M}.
> $$
> This way, the proof no longer requires an almost-sure bound on the gradient estimator, and the bias term is controlled with high probability rather than in expectation with Chebyshev's inequality. Note that this introduces an extra $\frac{1}{\delta^2}$ factor on $K_m^2$. As we discussed earlier, this can be improved with a sharper upstream concentration bound for the self-normalized gradient estimator.
>
> > They have not quantified how the smoothness, and Lipschitz parameter of the loss changes with $t$. Note that a significant chunk of existing works deal with quantifying precisely this, and it is non-trivial.
>
> We thank the reviewer for pointing this out, and we agree that quantifying the dependence of smoothness and Lipschitz constants on the smoothing scale is a central and nontrivial issue in the smoothing literature. Our setting differs from analyses based on global log-concavity or global convexity-type assumptions: Assumptions 4.1--4.2 allow localized nonconvex spikes outside the certified basin, so a useful global smoothness or Lipschitz bound may be loose or unavailable. We therefore state the quantities needed for the optimization argument in a **local rather than global** way: we analyze how the convex region evolves under smoothing and how it can be affected by the surrounding nonconvex landscape, as $x_m$, by design, always stays inside the certified smoothed-convex region $\mathcal R_{\mathrm{SC}}(t)$.
>
> The explicit local scalings are in the appendix Theorem B.7. Inside $\mathcal R_{\mathrm{SC}}(t)$, the local condition-number theorem bounds $\kappa_t=\beta_t/\alpha_t$. It uses the Hessian $\nabla_x^2 g(x;t)=\lambda(tI- \mathrm Cov_{0,t}(y\ |\ x))/t^2$ and the PSD covariance bound $0\preceq \mathrm Cov_{0,t}(y\ |\ x)\preceq tI$ to obtain $\beta_t\le \lambda/t$. Consequently, if $x_t^\star \in \mathcal R_{\mathrm{SC}}(t)$, a local function-Lipschitz constant over the certified ball can be taken as $L_t\le 2\beta_t R_{\mathrm{cert}}(t)$, where $R_{\mathrm{cert}}(t)$ is the certified convex radius. This smoothness/Lipschitz dependence is deliberately **local rather than global**, because the outer nonconvex region is not required to satisfy the same bounds.

---

> ### Author Response · Authors · 2026-06-01
> **Related work comparsion**
>
> ## Related-work comparisons
>
> > The authors have not provided a comparison to existing convergence results for smoothing for convex/strongly-convex and non-convex functions in (Nesterov & Spokoiny 2017, Duchi et al 2014, Hazan et al 2015, Mobahi & Fisher 2015). Further, they have also missed certain references to smoothing-based methods for non-convex functions in (Vardhan & Stich, 2021) and (Mishchenko & Stich, 2023). Finally, geometrically decreasing step-sizes have been well-studied in stochastic optimization and closely resemble geometrically decreasing smoothing parameters (Ge et al 2019).
>
> We agree and have expanded the related-work section. We read these papers as closely related but distinct in the following way:
>
> - **Nesterov and Spokoiny; Duchi et al.** establish complexity bounds for random gradient-free optimization. Nesterov and Spokoiny analyze Gaussian search directions, while Duchi et al. sharpen minimax rates for two-point zeroth-order convex optimization. Their focus is oracle complexity and direct optimization of the original objective, mostly in convex settings or stationarity-style nonconvex guarantees. Our focus is the landscape of the Boltzmann-smoothed objective $g(x;t)=-\lambda\log((p_0*k_t)(x))$, and how sampling-based softmax updates behave as approximate descent on that objective. The **coverage--displacement tradeoff** we analyze is not studied in those works.
> - **Hazan et al.** give a graduated-optimization algorithm for $\sigma$-nice nonconvex functions, including stochastic and zeroth-order variants. This is the closest continuation-style comparison. Our assumptions and object of analysis differ: rather than assuming a global niceness property of all smoothed objectives, we derive a local smoothed-convex region for the Boltzmann objective and explicitly track the **coverage--displacement tradeoff**. Our theorem can be viewed as deriving, for this Boltzmann-smoothed sequence, a local analogue of the niceness property that Hazan et al. assume globally.
> - **Mobahi and Fisher / Mobahi and Fisher III.** study Gaussian continuation from a homotopy viewpoint: one paper relates Gaussian smoothing to convex-envelope approximations, and the other bounds the endpoint of Gaussian continuation through an optimization-complexity measure. These works motivate Gaussian smoothing, but they do not analyze finite-sample sampling-based updates, the softmax estimator, the coverage--optimizer-displacement tradeoff, or the need to co-anneal $t$ and $\lambda$.
> - **Harshvardhan and Stich.** analyze first-order perturbed SGD for objectives $f=g+h$. Their assumptions control the smoothed stochastic-gradient noise and the bias between the smoothed gradient and $\nabla g$, where $g$ satisfies a PL or strong-convexity condition. Our setting does not assume access to gradients or a known regular component $g$; it studies zeroth-order sampling-based updates on the Boltzmann-smoothed objective.
> - **Mishchenko and Stich.** study Gaussian noise injection for first-order SGD when $f=\hat f+\omega$, with $\hat f$ smooth/PL and the perturbation $\omega$ bounded with bounded Clarke subgradients. Their guarantee is a gradient-based convergence-to-a-neighborhood result for the regular part, whereas our analysis tracks the smoothed convex region and the moving optimizer under annealed $t$ and $\lambda$.
> - **Ge et al.** study geometrically decaying learning rates for the final iterate of SGD in streaming least-squares regression under noise-covariance and fourth-moment assumptions. This supports the broader relevance of geometric schedules, but our geometric schedule is over the smoothing level $t_m$, coupled with $\lambda_m$, not over the learning rate.
>
> Compared to this literature, where smoothing-related ideas have been studied, the **coverage--displacement effect** on the optimization landscape has not been characterized for the sampling-based Boltzmann-smoothed setting. This is the core novelty of our work. We clarify that our contribution is therefore not simply "smoothing helps," but a more fine-grained analysis on how smoothing affects landscape, and the resulting convergence analysis for sampling-based updates that ties together the smoothed Boltzmann landscape, optimizer displacement, and annealing.

---

> ### Author Response · Authors · 2026-06-01
> **Experimental details**
>
> ## Experimental details
>
> > I'm not sure ... actually satisfy either Assumption 4.1 or 4.2.
>
> We agree the original experiments did not clearly distinguish certified examples from stress tests. We now state that the linear-control appendix construction is the **certified nonconvex example** satisfying Assumptions 4.1--4.2. Rastrigin and Levy are closer to the assumed setting because they combine quadratic growth with bounded/oscillatory terms and local regularity, but we do not claim full certification. Ackley, checkerboard, bounded-domain visualizations, and trajectory-optimization tasks are **empirical stress tests** beyond the strict sufficient assumptions.
>
> > The authors never define what their main metric of "cost" in Tables 1 and 3.
>
> We now define cost explicitly. For black-box optimization, cost is the best objective value found within the evaluation budget, $\mathrm{cost}=\min_{m,i}f(y_{m,i})$; equivalently, the implementation maximizes $J=-f$ and reports $-\max_{m,i}J(y_{m,i})$. Tables report mean $\pm$ standard deviation over seeds. For trajectory-optimization tasks, cost is the normalized benchmark trajectory objective, with lower values better. We have added this to the main text.
>
> > They never explicitly ..., but they use it in Figure 1.
>
> We now give the plotted GMM formula. For the 1D GMM visualization in Figure 1, we use the unnormalized 128-component density
> $$
> \rho_t(x)=\sum_{i=1}^{128} s_i\exp\!\left(-\frac{(x-\mu_i)^2}{2(t+3\cdot10^{-3})}\right)+10^{-2},
> $$
> and plot $g(x,t)=-\log\rho_t(x)$ on a bounded domain. Normalization would only add an additive constant to $g$ and does not affect the displayed minimizers or curvature, so we omit it.
> > The authors have also not provided a strategy to compute $t_{M_0}$.
>
> We agree that computing an optimal switching time is difficult in practice; we now treat $t_F$ as a user-chosen precision hyperparameter rather than an oracle quantity and state this as a limitation, not an algorithmic requirement. The practical DIDA algorithm does not compute $M_0$ or $t_F$; it follows the prescribed geometric annealing schedule. We clarified this in the paper.
>
> > In Figure 3, several quantities have not been defined in the legend.
>
> We expanded the caption and appendix description. The purple dot is the reference optimizer $x^\star=(-0.25,0.25)$, the red star is the grid minimizer $x_t^\star$ of the smoothed landscape, and the green dashed circle is the largest ball centered at $x^\star$ before the estimated zero-level contour of $\lambda_{\min}(\nabla^2g(x,t))$. The top row visualizes $g(x,t)=-\log\rho_t(x)$, and the bottom row visualizes the Hessian-based local convexity certificate. The coverage radius is computed on a $2048\times2048$ grid by evaluating Hessian eigenvalues and measuring the distance from $x^\star$ to the nearest zero-level contour. If no zero crossing is found inside the plotted domain, we label the plotted domain as globally covered; otherwise the finite-radius region is labeled local coverage.
>
> ## Other minor revision
>
> > The definition of $\mathcal{SG}$ ..., the table has no caption.
>
> We corrected the notation table. It now has a caption, and the Gaussian notation uses the standard inverse-quadratic form $\exp(-\frac12(x-\mu)^\top\Sigma^{-1}(x-\mu))$. The $\mathcal{SG}(\mu,\sigma^2)$ entry is now explicitly a scalar sub-Gaussian tail proxy: $\sigma^2>0$ is a scalar parameter in the bound $P(\|X-\mu\|\ge a)\le\exp(-a^2/(2\sigma^2))$, not a covariance matrix. This fixes both the missing negative sign and the invalid vector-by-matrix expression. We also corrected the smoothed-gradient entry to $\nabla_x g(x;t)=\frac{\lambda}{t}(x-\mathbb E_{p_{0|t}}[y\mid x])$, added the matching zeroth-order estimator $\nabla_x g^{(0)}(x;t)=\frac{\lambda}{t}\left(x-\frac{\sum_iw_iy_i}{\sum_iw_i}\right)$, and corrected the backward conditional density notation.
>
> > Lemma B.2: Where is its proof or a reference for it?
>
> Lemma B.2(NoW Lemma B.3) is proved immediately after its statement in Appendix B. We point this out in the response and keep the lemma self-contained, since the proof is a short tail-integral argument for one-sided sub-Gaussian moments.
>
> > What is the undefined notation ... numbered separately?
>
> We clarified this SNIS notation in Appendix C. Before the displayed SNIS bias/MSE bound, we now define $\pi$ as the law of the Gaussian proposal perturbation $Z$, define $\pi(F)=\mathbb E_\pi[F(Z)]$, define the recentered weight $h_x(Z)$, and define the self-normalized quantities $\mu_x(\phi)=\pi(\phi h_x)/\pi(h_x)$ and $\mu_x^N(\phi)=\frac{N^{-1}\sum_i\phi(Z_i)h_x(Z_i)}{N^{-1}\sum_i h_x(Z_i)}$ with $\phi(Z)=Z$. We also combine the two inequalities into a single numbered display, since they are the bias and MSE parts of the same SNIS bound.